# The basolateral amygdala-anterior cingulate pathway contributes to depression-like behaviors and comorbidity with chronic pain behaviors in male mice

Léa J. Becker[1,9], Clémentine Fillinger[1], Robin Waegaert[1], Sarah H. Journée[1], Pierre Hener[1], Beyza Ayazgok[1,2,10], Muris Humo[1,10], Meltem Karatas[1,3,10], Maxime Thouaye[1], Mithil Gaikwad[1,4], Laetitia Degiorgis[3], Marie des Neiges Santin[3], Mary Mondino[3], Michel Barrot[1], El Chérif Ibrahim[5], Gustavo Turecki[6], Raoul Belzeaux[5,7], Pierre Veinante[1], Laura A. Harsan[3], Sylvain Hugel[1], Pierre-Eric Lutz[1,8,11] & Ipek Yalcin[1,4,11] ✉

While depression and chronic pain are frequently comorbid, underlying neuronal circuits and their psychopathological relevance remain poorly defined. Here we show in mice that hyperactivity of the neuronal pathway linking the basolateral amygdala to the anterior cingulate cortex is essential for chronic pain-induced depression. Moreover, activation of this pathway in naive male mice, in the absence of on-going pain, is sufficient to trigger depressive-like behaviors, as well as transcriptomic alterations that recapitulate core molecular features of depression in the human brain. These alterations notably impact gene modules related to myelination and the oligodendrocyte lineage. Among these, we show that *Sema4a*, which was significantly upregulated in both male mice and humans in the context of altered mood, is necessary for the emergence of emotional dysfunction. Overall, these results place the amygdalo-cingulate pathway at the core of pain and depression comorbidity, and unravel the role of *Sema4a* and impaired myelination in mood control.

Major depressive disorder (MDD) and chronic pain are long-lasting detrimental conditions that significantly contribute to the worldwide burden of disease[1,2]. These two pathologies are highly comorbid, which results in increased disability and poorer prognosis compared to either condition alone[3,4]. Despite their high prevalence and co-occurrence, available treatments remain ineffective, urging for a better understanding of their pathophysiological convergence.

The frontal cortex, particularly the anterior cingulate cortex (ACC), is at the center of emotional and pain processing[5]. A large body of evidence documents major alterations in ACC neuronal activity in

[1]Centre National de la Recherche Scientifique, Université de Strasbourg, Institut des Neurosciences Cellulaires et Intégratives, Strasbourg, France. [2]Department of Biochemistry, Faculty of Pharmacy, University of Hacettepe, Ankara, Turkey. [3]Laboratory of Engineering, Informatics and Imaging (ICube), Integrative multimodal imaging in healthcare (IMIS), CNRS, UMR 7357, University of Strasbourg, Strasbourg, France. [4]Department of Psychiatry and Neuroscience, Université Laval, Québec, QC G1V 0A6, Canada. [5]Aix-Marseille Univ, CNRS, INT, Inst Neurosci Timone, Marseille, France. [6]Department of Psychiatry, McGill University and Douglas Mental Health University Institute, Montreal, QC, Canada. [7]Department of Psychiatry, CHU de Montpellier, Montpellier, France. [8]Douglas Mental Health University Institute, Montreal, QC, Canada. [9]Present address: Department of Anesthesiology, Center for Clinical Pharmacology Washington University in St. Louis, St. Louis, MO, USA. [10]These authors contributed equally: Beyza Ayazgok, Muris Humo, Meltem Karatas. [11]These authors jointly supervised this work: Pierre-Eric Lutz, Ipek Yalcin. ✉e-mail: yalcin@inci-cnrs.unistra.fr

patients with either chronic pain or MDD, as well as in rodent models of each condition[6–9]. Among other findings, our group previously showed that a lesion[10] or the optogenetic inhibition[11] of the ACC alleviates anxiodepressive-like consequences of neuropathic pain in mice, while activation of this structure is sufficient to trigger emotional dysfunction in naive animals. These results highlight a critical role of the ACC in the comorbidity between pain and MDD. However, the mechanisms priming ACC dysfunction remain poorly understood.

MDD originates from alterations affecting both the subcortical processing of external and internal stimuli, and their integration into perceived emotions by higher-level cortical structures[12]. It is, therefore, critical to understand how subcortical inputs to the ACC contribute to the emergence of emotional dysfunction. Among these, the anterior part of the basolateral nucleus of the amygdala (BLA) shows dense, direct and reciprocal connections with areas 24a/b of the ACC[13,14], which have been poorly studied in animal models of pain and mood disorders. The BLA plays a critical role in emotional processes, since its neurons encode stimulus with a positive valence, such as rewards[15,16], as well as those with a negative valence, including fear[16] or pain states[17]. In humans, neuroimaging studies have consistently found that depressed patients[7,18] and individuals suffering from chronic pain[6,19] exhibit pathological ACC and BLA hyperactivity.

In this context, we hypothesized that the neuronal pathway linking the BLA and the ACC might represent a core substrate underlying the comorbidity between pain and MDD. First, using retrograde tracing and neuronal activity markers, as well as rodent brain functional magnetic resonance imaging (MRI), we demonstrate that this pathway is hyperactive when chronic pain triggers depressive-like behaviors. Then, by optogenetically manipulating this circuit, we show that its hyperactivity is both necessary for neuropathic pain-induced depressive-like (NPID) behaviors, and sufficient to trigger similar deficits in naive animals, in the absence of chronic pain-like behavior. We next characterize transcriptomic changes occurring in the ACC when BLA−ACC hyperactivity triggers mood dysfunction, and find that they strikingly resemble the molecular blueprint of depression in humans. These results, which notably include alterations affecting oligodendrocytes and myelination, are coherent with our mouse imaging data showing altered tissue anisotropy along the BLA−ACC pathway, and establish the translational relevance of our BLA−ACC optogenetic model of MDD. Finally, we leverage gene-network approaches to prioritize *Semaphorin 4a* (*Sema4a*) as a gene significantly upregulated in both mice and humans in the context of altered mood. Using a gene knockdown approach, we demonstrate that, following optogenetic BLA−ACC stimulation, upregulation of *Sema4a* is necessary for the emergence of depressive-like behaviors. Overall, these results uncover the BLA−ACC pathway as a core substrate of pain and MDD comorbidity.

## Results

### Chronic neuropathic pain induces hyperactivity in the BLA−ACC pathway

To confirm the anatomical connection between the BLA and ACC, we first injected the retrograde tracer cholera toxin subunit B (CTB) into the areas 24a/b of the ACC (Fig. 1a). Consistent with previous reports, strongly labeled cell bodies were found in the BLA (Fig. 1b)[13,20,21]. Next, considering that we[11], and others[5,8,22], have found that the ACC is hyperactive when chronic pain triggers depressive-like behaviors, we wondered whether the BLA is similarly affected. To address this question, we quantified the immediate early gene c-Fos in our well-characterized NPID model[23,24]. In this model, right-sided peripheral nerve injury leads to immediate and long-lasting mechanical hypersensitivity, with delayed anxiodepressive-like behaviors (significant at 7 weeks post-operation, PO; Fig. 1c–f). In our previous study, increased c-Fos immunoreactivity was found in the ACC at 8 weeks PO[25]. Here, a similar c-Fos increase was found in the BLA (Supplementary Fig. 1a, b).

This indicates concurrent neuronal hyperactivity of the two structures when anxiodepressive-like behaviors are present. Next, we studied whether these c-Fos positive cells in the BLA directly innervate the ACC (Fig. 1g). The retrograde tracer fluorogold (FG) was injected into the ACC of neuropathic animals, and we quantified its co-localization with c-Fos in the BLA at 8 weeks PO (Fig. 1c). Similar numbers of BLA neurons projecting to the ACC (FG +) were found in both sham and NPID groups, indicating that neuropathic pain does not modify the number of neurons in this pathway (Fig. 1h, i). Compared to shams, an increase in the total number of c-Fos+ cells was observed in the ipsilateral, but not in the contralateral, BLA of NPID mice (Fig. 1j, k), confirming results from our previous cohort (Supplementary Fig. 1a, b). Importantly, this neuronal hyperactivity in the NPID group notably concerned neurons that project to the ACC, as shown by an increase in cells positive for both FG and c-Fos (Fig. 1m, n) in the BLA ipsilateral to nerve injury. Again, no change was observed in the contralateral BLA (Fig. 1l). While similar lateralization has already been reported in the central amygdala during chronic pain[26,27], especially at a later timepoint[28], our results extend these findings to the BLA in the context of NPID. Also, a recent systematic review of 70 human imaging studies further supports our observation, as pain concomitant with depression was found to strongly associate with enhanced connectivity of the right amygdala, particularly with the prefrontal cortex[29]. Finally, as a complementary strategy to assess the connectivity of the ACC and the BLA, we took advantage of imaging data recently generated in our NPID model using resting-state functional Magnetic Resonance Imaging (rs-fMRI). Consistent with above histological analyses, we observed that functional connectivity between the two structures was enhanced at 8, but not 2, weeks PO in NPID animals compared to sham controls (Fig. 1o and Supplementary Fig. 1c, d for behavioral characterization). Altogether, these results indicate that chronic pain induces hyperactivity of BLA neurons projecting to the ACC.

### Optogenetic inhibition of the BLA−ACC pathway prevents NPID

We next hypothesized that this hyperactivity may be responsible for emotional dysfunction. To address this, we inhibited the pathway using an AAV5-CamKIIa-ArchT3.0-EYFP vector injected bilaterally in the BLA (Fig. 2a, b). To characterize the effects of green light illumination on BLA neurons, we performed ex vivo electrophysiological recordings 6 weeks after viral injection (Fig. 2c). Patch-clamp recordings at the level of the BLA showed that light stimulation resulted in neuronal inhibition, proportional to light intensity, as indicated by outward currents recorded in voltage-clamp mode (Fig. 2d, e).

To assess behavioral effects of inhibiting the BLA−ACC pathway during NPID, the same viral vector was injected bilaterally in the BLA, followed by implantation of an optic fiber in the ACC, to specifically inhibit axon terminals coming from the BLA (Fig. 2f; for BLA viral injection localization, see Supplementary Fig. 2a). Acute inhibition did not impact mechanical thresholds, measured using von Frey filaments, in sham (Supplementary Fig. 2b, c) or nerve-injured animals at either 3 or 6 weeks PO (Fig. 2g, h). The BLA−ACC pathway is, therefore, not essential for mechanical hypersensitivity, consistent with what several groups reported when manipulating the whole ACC[10,30–32]. Likewise, the BLA−ACC pathway does not drive ongoing pain-like behavior, since we did not observe any significant conditioned place preference (CPP) during optogenetic inhibition (Fig. 2i). Inhibiting the whole ACC was however sufficient to induce CPP in our previous work[11]. Therefore, modulation of pain states by the ACC likely relies on other afferent structures than the BLA[33].

We next assessed the effect of inhibiting the BLA−ACC pathway on anxiodepressive-like behaviors. Optogenetic inhibition was applied just before (for light/dark, LD, and forced swim tests, FST) or during (splash test, ST) behavioral testing. As expected, NPID animals displayed significantly higher anxiety-like behaviors in the LD at 7 weeks PO (Fig. 2j), as well as higher depressive-like behaviors at 8 weeks PO

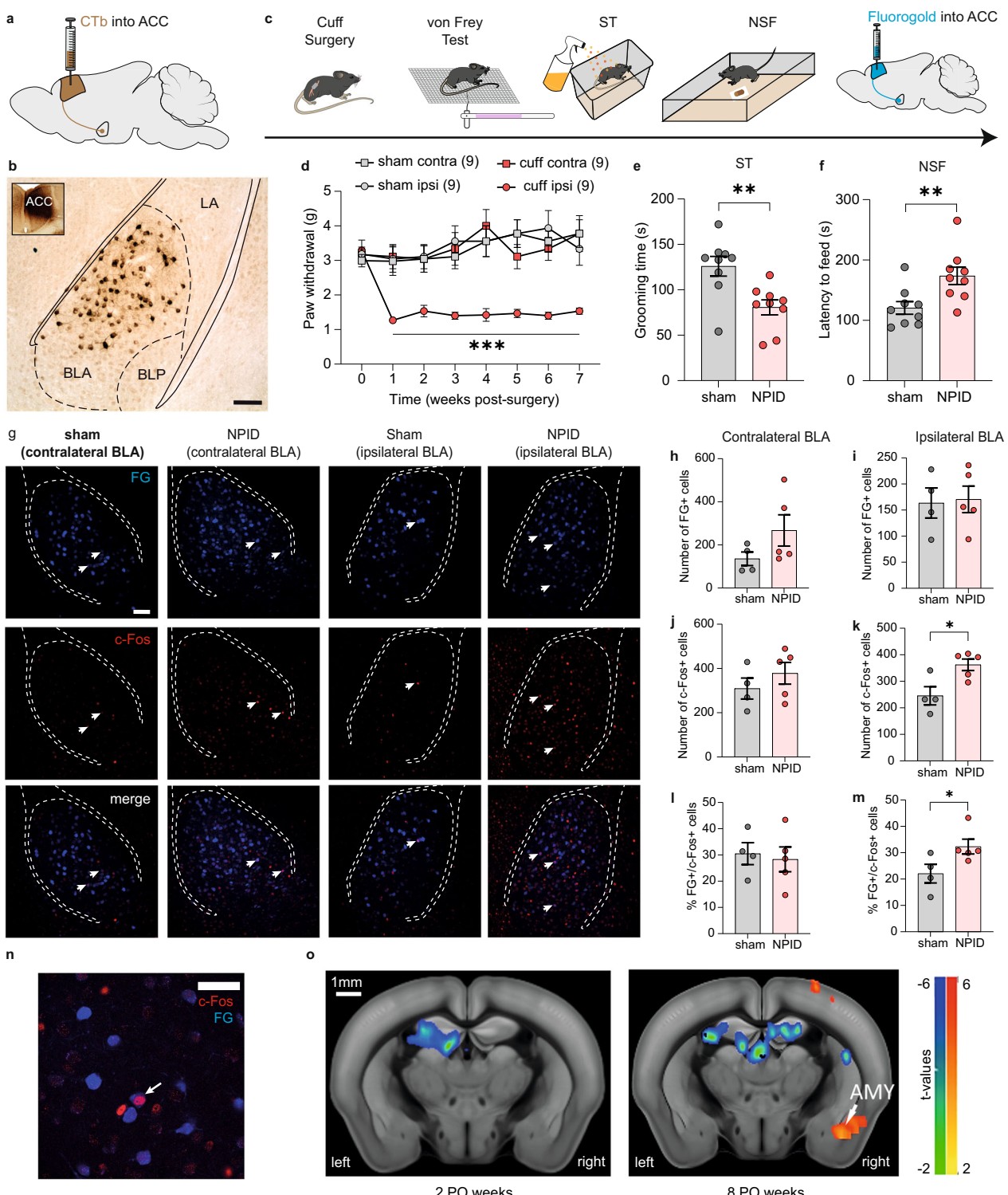

(Fig. 2k, l) in both the ST and FST[11,24]. Inhibiting the BLA–ACC pathway had no impact in the LD test, suggesting that other pathways may control anxiety-like consequences of chronic pain. In contrast, BLA–ACC inhibition completely reversed pain-induced decreased grooming in the ST (Fig. 2k), and significantly decreased immobility in the FST (Fig. 2l), revealing potent antidepressant-like effects. No effect of optogenetic inhibition was observed in sham animals, indicating that these antidepressant-like effects selectively manifest in the context of chronic pain. As a more general measure of emotionality[34], we also calculated z-scores for each animal across all three tests (LD, ST, FST). Results indicated that NPID mice showed global emotional deficit

compared to sham controls, as indicated by lower z-scores, an effect that was prevented by inhibition of the BLA–ACC pathway (Supplementary Fig. 2d). Overall, these results demonstrate that hyperactivity of BLA neurons targeting the ACC is necessary for the selective expression of chronic pain-induced depressive-like behaviors.

## Repeated activation of the BLA–ACC pathway triggers depressive-like behaviors in naive mice

We next determined whether BLA–ACC hyperactivity is sufficient to induce depressive-like behaviors in naive mice, in the absence of neuropathic pain. An AAV5-CamKIIa-ChR2-EYFP vector was injected

**Fig. 1 | Neuropathic pain-induced depression (NPID) triggers hyperactivity in BLA neurons projecting to the ACC and increases functional connectivity between the ACC and BLA. a** Retrograde tracing strategy, with the injection of the cholera toxin B subunit (CTB) into the anterior cingulate cortex (ACC). **b** Representative image of retrogradely labeled cell bodies in the anterior part of the basolateral nucleus of the amygdala (BLA). Scale bar = 100 μm. **c** Experimental design for quantifying the activity of BLA neurons projecting to the ACC, during NPID. **d**–**f** Peripheral nerve injury induced an ipsilateral mechanical hypersensitivity (**d**; $F_{(21,224)} = 2.710$; $P < 0.0001$; post hoc 1–7 weeks $P < 0.05$), decreased grooming in the splash test (ST) (**e**; sham: 125.90 ± 10.80; NPID: 80.67 ± 8.22; $P = 0.0042$) and increased latency to feed in novelty-suppressed feeding (NSF) (**f**; sham: 120.7 ± 10.49; NPID: 173.7 ± 14.38; $P = 0.0089$) ($n = 9$ mice/group). **g** Representative images showing positive cells for fluorogold (FG + , upper panel), c-Fos (c-Fos + , middle), or co-labeled (bottom), in the contralateral (sham: right column; NPID: middle right) or ipsilateral BLA (sham: middle left; NPID: left). Scale bar = 100 μm. **h**–**m** Quantification of FG + , c-Fos+ cells and their co-localization revealed that, at 8 weeks postoperative (PO), the number of FG + cells was not altered (**h**;

contralateral BLA: sham: 135.8 ± 31.52; NPID: 267.2 ± 72.58; $P = 0.21$; **i**; ipsilateral BLA: sham: 163.5 ± 28.85; NPID: 170.6 ± 25.43; $P = 0.3651$), as well as c-Fos + (**j**; sham: 309.3 ± 47.60; NPID: 378.2 ± 49.10; $P = 0.14$) and % of FG + /c-Fos + (**l**; sham 30.50 ± 4.20; NPID: 28.33 ± 4.72; $P = 0.45$) cells in the contralateral BLA. In contrast, the number of c-Fos + (**k**; sham: 245.5 ± 34.37; NPID: 362.4 ± 21.62; $P = 0.0238$) and % of FG + /c-Fos+ cells (**m** sham 22.06%±3.55; NPID: 32.32%±2.81; $P = 0.0159$) were increased in the ipsilateral BLA (sham: $n = 4$ mice; NPID: $n = 5$ mice). **n** Close-up image showing the co-localization of c-Fos+ and FG + cells. Scale bar = 20 μm. **o** Inter-group statistical comparisons showed increased functional connectivity between ACC and amygdala (AMY) at 8 (right image) but not 2 (left) PO weeks. FWER corrected at cluster level for $P < 0.05$. Data are mean ± SEM. *$P < 0.05$; **$P < 0.01$, ***$P < 0.001$. Two-way ANOVA repeated measures (Time × Surgery; VF); two-sided unpaired $t$ test (ST and NSF); one-sided Mann–Whitney test (FG, c-Fos quantification). Contra contralateral, ipsi ipsilateral, BLP posterior part of the BLA, LA lateral amygdala. Sagittal mouse brain cartoons (**a**, **c**) were created with Biorender.com. Source data are provided as a Source Data file.

bilaterally in the BLA, and patch-clamp recordings confirmed that blue light stimulation (wavelength: 475 nm; pulse duration: 10 ms; frequency: 10 Hz) evoked inward currents in eYFP-expressing BLA neurons, as well as evoked excitatory postsynaptic currents in pyramidal ACC neurons (Fig. 3a–h). In the BLA, optogenetic stimulation of cell bodies induced inward currents, with a plateau reached at 40% of maximal light intensity, while pulsed 10 Hz stimulation produced strong and stable currents (Fig. 3b–d). In the ACC, stimulating BLA axon terminals induced strong inward currents, with a plateau at 80% of maximal light (Fig. 3g, h), indicating activation of ACC neurons.

Having established this stimulation protocol, we explored its behavioral impact in naive mice. An optic fiber was implanted in the ACC, and blue light pulse stimulation applied for 20 min (with parameters validated ex vivo; see Supplementary Fig. 3a for BLA viral injection localization). We first found that a single stimulation session was not sufficient to trigger any alteration in spontaneous locomotor activity, real-time place avoidance or anxiodepressive-like assays (Supplementary Fig. 3b–e). Thus, we next tested the effects of repeated stimulations over 3 consecutive days each week, during 3 weeks (Fig. 4a, b). Behavioral tests were performed at 3 timepoints (i.e. at the end of week 1, 2, or 3), following each block of three activating sessions. A subset of mice was also used to document the impact of our optogenetic stimulation on neuronal activity in vivo. To do so, a 10th session of optogenetic stimulation was applied (5 days after the 9th stimulation), followed by quantification of c-Fos immunoreactivity (90 mn post-stimulation). Results showed robust increase in c-Fos+ cells throughout the ACC (Supplementary Fig. 3f), as expected. At the behavioral level, no effect of stimulations was found on locomotor activity (Supplementary Fig. 3g) or anxiety-like behaviors (Fig. 4b, LD/NSF) at any of the 3 timepoints. This is consistent with above NPID results indicating that optogenetic inhibition of the pathway did not rescue pain-induced anxiety-like behaviors. In contrast, depressive-like behaviors progressively emerged: (i) After the first 3 stimulations, no changes were observed in ST, Nest test, or FST; (ii) After 6 stimulations, immobility in the FST significantly increased, with a tendency for decreased nest building, without significant effects in the ST; (iii) After 9 stimulations, emotional deficits further strengthened, with increased immobility in the FST accompanied by a significantly poorer nest score, along with decreased grooming in the ST (Fig. 4b). Similar to previous NPID experiments, we also computed emotionality z-score at each timepoint (Fig. 4c). Although no effect of repeated BLA–ACC activation was observed after three stimulations, emotionality z-score showed a tendency for a decrease after 6 stimulations ($P = 0.094$), which became significant after 9 stimulations, confirming the progressive emergence of emotional dysfunction. Of note, none of these behavioral effects remained detectable 1 week after the last stimulation (Supplementary Fig. 3h, i), indicating a reversible phenotype.

Overall, it is possible to speculate that this kinetic bears an analogy with our results in the NPID model, in which emotional deficits also progressively emerged, and required tonic hyperactivity of the BLA–ACC pathway. Compared to the human disorder, this short-lasting phenotype might be considered to model proximal causes of depression (such as environmental stressors or recent life events), which trigger acute variations in thymic states, rather than underlying endophenotypes (constitutive biological traits thought to act as distal risk factor).

To determine which neuronal cell types are recruited in the ACC by the repeated optogenetic activation paradigm, we next quantified *c-fos* expression and its co-localization with markers of excitatory (glutamatergic, *Slc17a7*) and inhibitory (GABAergic, *Gad2*) neurons, this time 48 h after the 9th stimulation, matching behavioral deficits (in the absence of any new stimulation). Repeated optogenetic activation induced a strong increase in *c-fos* expression, predominantly in glutamatergic but also in GABAergic cells (Fig. 4d, e and Supplementary Fig. 4a, b, e) of the ACC (24a/24b), while global numbers of *Gad2* + or *Slc17a7* + cells remained unaltered (Supplementary Fig. 4c, d). Altogether, these results show that repeated activation of the BLA–ACC pathway is sufficient to activate major neuronal cell types in the ACC, and to trigger the progressive emergence of a depressive-like phenotype.

## Repeated activation of the BLA–ACC pathway produces transcriptional alterations similar to those observed in human depression

The need for repeated activations to produce behavioral effects suggests transcriptomic alterations. To understand the underlying molecular mechanisms, we used RNA Sequencing to identify gene expression changes occurring in the ACC (Supplementary Data File 1) after 9 stimulations (Fig. 5a), when behavioral deficits are maximal. We generated 2 animal cohorts ($n = 12$ controls and $n = 10$ stimulated mice in total) and, before harvesting ACC tissue, confirmed the development of depressive-like behaviors (Fig. 5b). Analysis of gene expression changes was conducted as described previously[35] ("Methods"). Using Principal Component Analysis, robust differences were observed across groups at the genome-wide level (Supplementary. Fig. 5a). At nominal $P$ value ($P < 0.05$), 2611 genes were significantly dysregulated in stimulated mice compared to controls, with 54 genes remaining significant after Benjamini–Hochberg correction for multiple testing (Fig. 5c). Over-Representation Analysis[36] (ORA) uncovered alterations in Gene Ontology (GO) terms related to myelination, dendritic transport, neurogenesis and cytoskeleton (Supplementary Data File 1 and Supplementary Fig. 5b).

To test the relevance of these data to human depression, we compared them to our recent postmortem study[37]. In the latter, similar

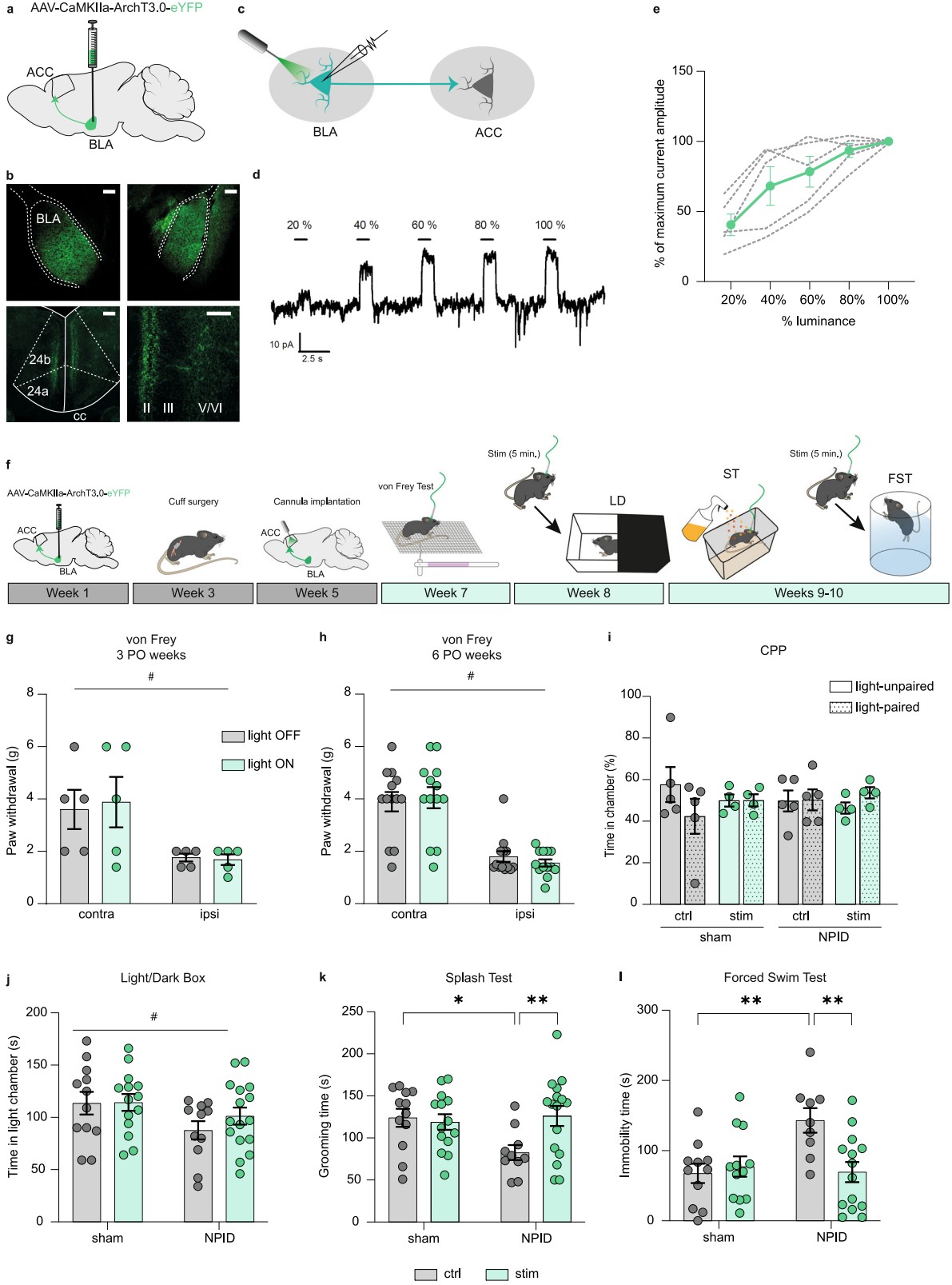

RNA-Seq was used to compare ACC tissue from individuals who died during a major depressive episode ($n = 26$; Supplementary Table 1) and healthy individuals without any psychiatric history ($n = 24$). Because only males were used in our mouse paradigm, we reprocessed human data to restrict differential expression analysis to men (for a similar analysis in both sexes pooled, see Supplementary. Fig. 6). We then used 3 strategies to compare men and mice, focusing on 13,572

orthologues (see "Methods" for detail). First, 398 genes were identified as differentially expressed (DEG, nominal $P$ value < 0.05) in both species (common DEGs; Supplementary. Fig. 5c). These represented 29.6% of all DEGs in mice (34.9% when considering both sex; Supplementary Fig. 6a, b), indicating robust overlap between species (Supplementary Fig. 5d). ORA performed on common DEGs revealed enrichments in GO terms related to neurogenesis, cytoskeleton or myelin sheath

**Fig. 2 | Optogenetic inhibition of the BLA−ACC pathway blocks NPID.**
**a** Graphical representation of virus delivery to the mouse BLA for voltage-clamp recordings. **b** Representative images of eYFP+ cell bodies in the BLA (upper panels) and eYFP+ axon terminals in the ACC (lower panels). Scale bars = 100 μm.
**c** Graphical representation of patch-clamp recording in the BLA. **d** Representative trace of outward currents. **e** Amplitude of currents induced by optogenetic stimulations of BLA neurons ($n = 5$ cells/2 mice, green trace = mean; gray traces = individual responses). **f** Graphical representation of the experimental design for in vivo optogenetic inhibition of the BLA−ACC pathway. **g**, **h** At 3 or 6 weeks PO, mechanical hypersensitivity was not affected by the inhibition of BLA−ACC pathway (ipsi vs contra; 3 PO weeks, cuff: $n = 5$ mice, $F_{(1,4)}=7.752$; $P = 0.0496$; 6 PO weeks, cuff: $n = 13$ mice $F_{(1,12)}=55.80$; $P < 0.0001$; light-off vs light-on; 3 PO weeks $F_{(1,4)}=0.669$; $P = 0.4592$; 6 PO weeks $F_{(1,12)}=2.971$; $P = 0.1104$). **i** Optogenetic inhibition of the BLA−ACC pathway did not induce a place preference at 6 weeks PO (sham-ctrl: $n = 5$ mice; sham-stim: $n = 4$ mice; NPID-ctrl: $n = 5$; NPID-stim: $n = 4$ mice; $F_{(3,13)}=0.153$; $P = 0.9998$). **j** At 7 weeks PO, BLA−ACC inhibition had no effect on the

decrease in time spent in the lit chamber observed in nerve-injured animals (sham-ctrl: $n = 12$ mice; sham-stim: $n = 14$ mice; NPID-ctrl: $n = 11$; NPID-stim: $n = 16$ mice; sham vs NPID: $F_{(1,49)} = 4.703$; $P = 0.035$; ctrl vs stim: $F_{(1,49)}=0.634$; $P = 0.43$). **k** At 8 weeks PO, BLA−ACC pathway inhibition reversed the decreased grooming observed in nerve-injured non-stimulated animals ($F_{(1,48)}=4.991$; $P = 0.03$; post hoc: sham-ctrl ($n = 12$ mice) >NPID-ctrl ($n = 10$ mice); $P < 0.05$; NPID-ctrl ($n = 10$ mice) <NPID-stim ($n = 16$ mice); $P < 0.05$ sham-ctrl ($n = 12$ mice) = sham-stim ($n = 14$ mice)). **l** At 8 weeks PO, BLA−ACC inhibition blocked the increased immobility observed in nerve-injured non-stimulated animals ($F_{(1,42)} = 7.539$; $P = 0.008$, post hoc: sham-ctrl ($n = 11$ mice) > NPID-ctrl ($n = 9$ mice), $P < 0.05$; NPID-ctrl ($n = 9$ mice) > NPID-stim ($n = 14$ mice), $P < 0.01$; sham-ctrl ($n = 11$ mice) = sham-stim ($n = 12$ mice)). Data are mean ± SEM. #, main effect; *$P < 0.05$; **$P < 0.01$. Two-way ANOVA repeated measures (von Frey); two-way ANOVA (Surgery × Stimulation; LD, ST and FST). ACC anterior cingulate cortex, PO postoperative, 24a/24b areas 24a/b of the ACC, II, III, V/VI ACC layers, cc corpus callosum. Sagittal mouse brain cartoons (**a**, **f**) were created with Biorender.com. Source data are provided as a Source Data file.

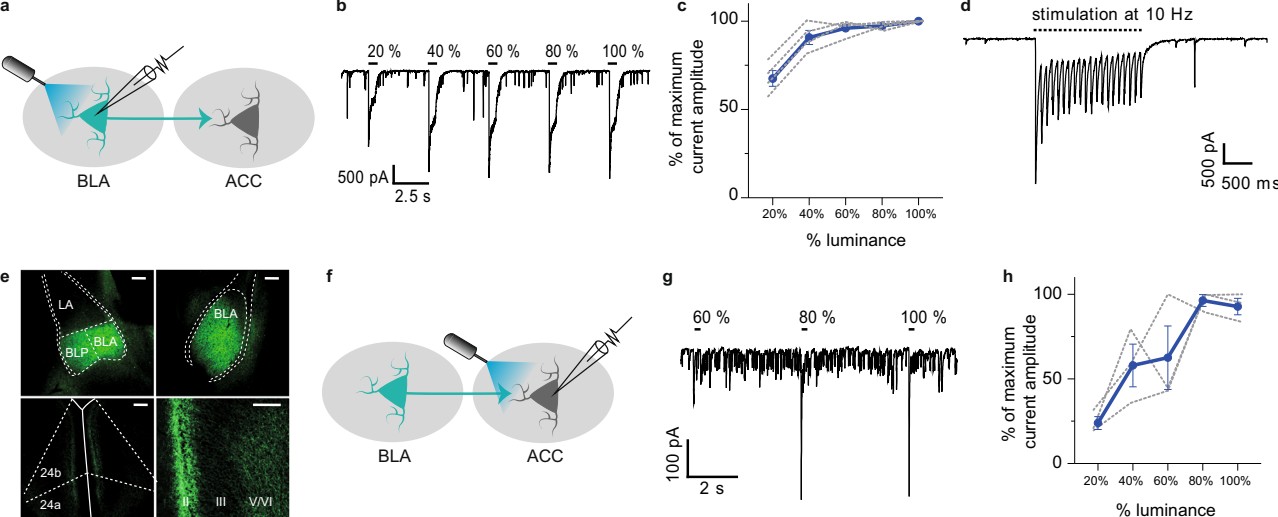

**Fig. 3 | ChR2 expression in BLA neurons drives robust light-induced activation of BLA and ACC neurons. a** Graphical representation of ex vivo voltage-clamp recordings in the BLA. **b** Representative trace of the outward currents induced by optogenetic stimulation with increased luminance in a BLA neuron. **c** Amplitude of currents evoked by optogenetic stimulation of BLA neurons as a function of light stimulation intensity ($n = 4$ cells/3 mice, blue trace = mean; gray traces = individual responses). **d** Representative trace of response of BLA neurons to 10 Hz optogenetic activation showing that after an initial decrease in the amplitude of light-induced currents, a plateau is reached. **e** Representative images of eYFP+ cell bodies in the BLA (upper panels) and eYFP+ axon terminals in the ACC (lower panels). Scale bars = 100 μm. **f** Graphical representation of the configuration for

ex vivo voltage-clamp recordings in the ACC. **g** Representative trace of the inward currents induced by optogenetic activation of BLA terminals within the ACC with increased luminance. **h** Amplitude of currents evoked by optogenetic stimulations of BLA terminals recorded in ACC pyramidal neurons as a function of light stimulation intensities ($n = 3$ cells/3 mice, blue trace = mean; gray traces = individual responses). Data are represented as mean ± SEM. 24a, 24b: areas 24a and 24b of the ACC, II, III, V/VI cortical layers of the ACC, BLA anterior part of the basolateral nucleus of the amygdala, BLP posterior part of the basolateral nucleus of the amygdala, cc corpus callosum, LA lateral amygdala. Source data are provided as a Source Data file.

(Supplementary Fig. 5e). Second, for a more systematic threshold-free comparison, we used RRHO2, (Rank-Rank Hypergeometric Overlap[38]). This analysis uncovered large patterns of transcriptional dysregulation in similar directions as a function of MDD in men, and of optogenetic activation in mice. Indeed, strong overlaps were observed between species for both upregulated and downregulated genes (Fig. 5d), notably affecting pathways related to oligodendrocyte cell fate and myelination (Supplementary Data Files 2 and 3). Third, because results repeatedly pointed toward myelin, we used Gene Set Enrichment Analysis (GSEA, Fig. 5e−g) to interrogate a well-characterized list of 76 genes primarily expressed by oligodendrocytes, encompassing their major biological functions[39]. Among these, 48 (63%; Fig. 5e) were downregulated in stimulated mice, and 57 in men with MDD (75%; Fig. 5f). This pattern of myelin downregulation is strikingly similar across species (Fig. 5g). Altogether, these 3 approaches (common DEG, RRHO2 and GSEA) converged to

reveal significant dysregulation of the myelination gene program (see Supplementary Figs. 5e−g and 6c−e).

## Gene ontology pathways affected in mice and men point towards altered oligodendrocyte function

To better capture the modular disorganization of gene expression in MDD, we then used weighted gene co-expression network analysis, WGCNA[40], similar to our recent work[41]. Gene networks, constructed independently in each species, were composed of 25 and 29 gene modules in mice and humans, respectively. Using Fischer's exact $t$ test, we found that the gene composition of 14 mice modules (56%) was significantly enriched in human modules, indicating their conservation (Benjamini−Hochberg, $P < 0.05$; Fig. 6a, Supplementary Data File 4, see Supplementary Fig. 6f, and Supplementary Data File 5 when including both sex). To identify which modules are most significantly impacted, we computed correlations between each module's eigengene

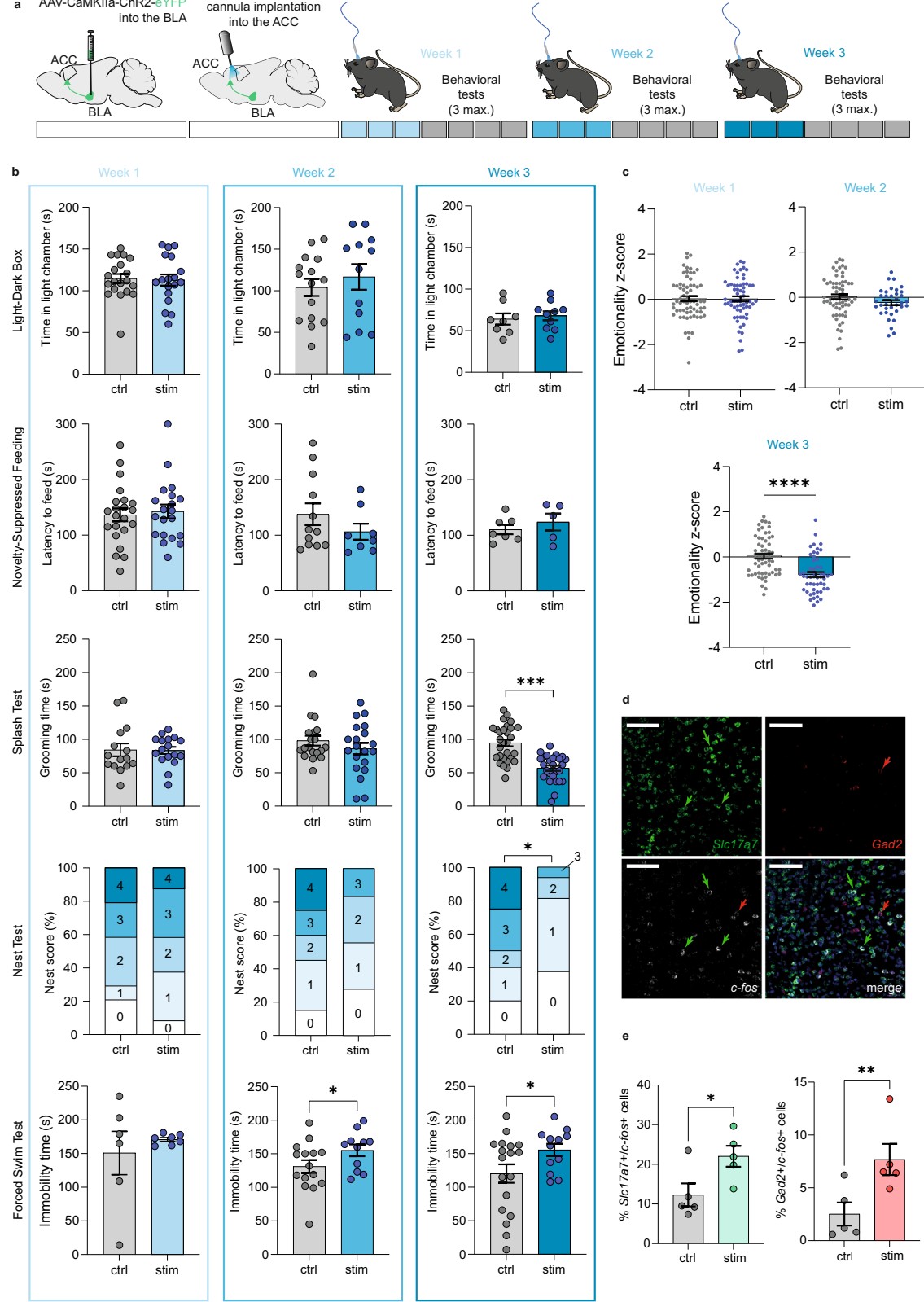

(a measure that summarizes co-expression) and optogenetic activation or MDD. In humans, 12 module eigengenes significantly correlated with MDD; in mice, 6 modules associated with optogenetic stimulations (Fig. 6b and Supplementary Fig. 6g). Importantly, among these, 8/12 and 5/6 modules were also conserved across species. Therefore, human MDD and repeated optogenetic activation of the BLA–ACC pathway impact conserved modules in the ACC.

We then conducted ORA of conserved but disorganized modules. Enriched GO terms were related to synaptic activity, mitochondria or RNA processing (Supplementary Table 2), consistent with previous studies of MDD[42–44]. Importantly, the 2 most strongly conserved modules (Men/Yellow and Mouse/Brown) again implicated myelin, with enrichments related to oligodendrocytes and myelination (Fig. 6c). To assess where myelin genes are located

**Fig. 4 | Repeated activation of the BLA–ACC pathway triggers depressive-like behaviors in naive mice. a, b** Repeated optogenetic activation of the BLA–ACC pathway did not induce anxiety-like behaviors in the LD (3 stim: ctrl: $n = 20$ mice; 114.8 ± 5.34; stim: $n = 18$ mice; 112.9 ± 6.75; $P = 082.$; 6 stim: ctrl: $n = 15$ mice; 104.0 ± 10.31; stim: $n = 12$ mice; 116.7 ± 15.37; $P = 0.49$; 9 stim: ctrl: $n = 8$ mice; 64.0 ± 6.73; stim: $n = 10$ mice; 68.0 ± 5.34; $P = 0.64$) nor NSF (3 stim: ctrl: $n = 22$ mice; 136.5 ± 11.65; stim: $n = 20$ mice; 142.6 ± 12.44; $P = 0.72$; 6 stim: ctrl: $n = 12$ mice; 137.8 ± 19.52; stim: $n = 8$ mice; 106.4 ± 14.51; $P = 0.26$; 9 stim: ctrl: $n = 7$ mice; 110.3 ± 8.36; stim: $n = 5$ mice; 123.8 ± 15.24; $P = 0.42$) tests. Three stimulations did not change grooming in the splash test (ST, ctrl: $n = 15$ mice; 84.20 ± 9.37; stim: $n = 17$ mice; 83.41 ± 5.33; $P = 0.94$), nest building (ctrl: $n = 24$ mice; stim: $n = 24$ mice; Chi-square=0.012; $P = 0.91$) nor immobility in the FST (ctrl: $n = 6$ mice; 150.8 ± 32.29; stim: $n = 7$ mice; 171.0 ± 2.89; $P = 0.26$). Six stimulations did not alter grooming (ctrl: $n = 19$ mice; 98.16 ± 7.37; stim: $n = 20$ mice; 86.10 ± 8.7; $P = 0.51$) nor nesting (ctrl: $n = 20$ mice; stim: $n = 18$ mice; Chi-square=2.81; $P = 0.094$), but increased immobility in the FST (ctrl: $n = 15$ mice; 131.1 ± 9.47; stim: $n = 11$ mice; 155.1 ± 8.71; $P = 0.042$). The latter increase was still present after 9 stimulations (ctrl: $n = 18$ mice; 120.3 ± 13.69; stim: $n = 12$ mice; 155.5 ± 9.15; $P = 0.033$), along with decreased grooming (ctrl: $n = 29$ mice; 94.79 ± 5.02; stim: $n = 26$ mice; 56.81 ± 3.96; $P < 0.0001$) and nest quality (ctrl: $n = 20$ mice; stim: n = 16 mice; Chi-square=7.35; $P = 0.0067$). **c** Emotionality z-scores across tests and timepoints: three stimulations had no effect (ctrl: $n = 67$ mice; 0.033 ± 0.11; stim: $n = 64$ mice; 0.023 ± 0.11; $P = 0.97$), while a tendency for a decrease emerged after 6 stimulations (ctrl: $n = 40$ mice; −0.004 ± 0.082; stim: $n = 35$ mice; −0.232 ± 0.109; $P = 0.095$) and became significant after 9 (ctrl: $n = 59$ mice; 0.042 ± 0.11; stim: $n = 52$ mice; −0.78 ± 0.12; $P < 0.0001$). **d** Representative RNAscope images of *Slc17a7* (upper-left panel), *Gad2* (upper-right), *c-fos* (lower-left) mRNAs and their co-localization (lower-right) in the ACC. Scales = 100 μm. **e** Proportions of *Slc17a7* + /*c-fos* + (green) and *Gad2* + /*c-fos* + (red) cells increased in stimulated animals (ctrl: $n = 5$ mice; stimulated: $n = 5$ mice; *Gad2* + /*c-fos* +: ctrl: 2.52 ± 1.09; stim: 7.68 ± 1.48, $P = 0.008$; *Slc17a7* + /*c-fos* +: ctrl: 12.28 ± 2.89; stim: 22.04 ± 2.64, $P = 0.028$). Data are mean ± SEM. *$P < 0.05$; **$P < 0.01$; ***$P < 0.001$; ****$P < 0.0001$. Two-sided unpaired $t$ test (LD, NSF, ST, z-score); chi-square test for trend (Nest); one-sided Mann–Whitney test (mRNA quantification). 24a/b: areas 24a/b of the ACC, II, III, V/VI ACC layers. Sagittal mouse brain cartoons (**a**) were created with Biorender.com. Source data are provided as a Source Data file.

within these modules, we analyzed their module membership (MM), a measure of module centrality. Myelin-related genes displayed higher absolute values (i.e., higher centrality) compared to means among their host modules, in both species (Fig. 6d). Among the 235 genes belonging to the intersection between the 2 Men/Yellow and Mouse/Brown modules, 36 were myelin-related (Fig. 6e, red dots), and a strong correlation between mouse and human MM was found, indicating that the same set of genes is centrally located among the two modules. Finally, we also observed among these 2 modules a strong correlation in the directionality of gene expression changes across species, with a majority of downregulated genes (Fig. 6f). Altogether, this suggests that myelination transcriptional deficiency, a feature of MDD pathophysiology in the ACC, is recapitulated in our optogenetic paradigm.

We next validated RNA-sequencing results using microfluidic qPCR and a new mouse cohort generated with the same optogenetic protocol ($n = 8$ control and 7 stimulated mice). Behavioral effects of stimulations were first confirmed (Fig. 6g), followed by dissection of the ACC tissue and analysis of the expression of most abundant myelin sheath proteins (*Plp1, Mal, Mog, Mag, Mbp*), enzymes involved in the synthesis of myelin lipids (*Aspa, Ugt8*), as well as positive (*Ermn*) and negative (*Sema4a, Lingo1*) regulators of myelination (Fig. 6h, i). Results significantly correlated with RNA-sequencing data, with similar downregulation of myelin proteins, or synthesis enzymes, as well as upregulation of 2 well-known negative regulators of myelination, *Sema4a* and *Lingo1*.

Finally, as complementary approaches to document how these myelin gene expression changes translate at cellular and network levels, we used immunohistochemistry (Fig. 7a) and brain imaging (Fig. 7e). We first assessed the number of ACC cells expressing Olig2, a transcription factor essential for proliferation and differentiation in the oligodendrocyte lineage, as well as PDGFRA, a marker of oligodendrocyte progenitor cells (OPC; Fig. 7b). Our results showed that the number of Olig2 + (Fig. 7b, c), but not PDGFRA + (Fig. 7b, d) cells decreased after nine stimulations, when depressive-like consequences are maximal (Supplementary Fig. 7a). This suggests that loss of mature oligodendrocytes underlies the decreased expression of myelin genes observed in bulk tissue. Of note, this decrease in Olig2+ cells was not observed one week after the 9th stimulation, when behavioral deficits are no longer present (Supplementary Fig. 7b, c), indicating a rapid recovery matching the behavioral kinetic. Finally, in a different cohort (Supplementary Fig. 7d for behavioral validation), we also performed MRI with DTI acquisition sequences, to analyze microstructural changes induced by optogenetic activation. Interestingly, stimulated animals displayed lower fractional anisotropy (FA) compared to controls in the ACC, amygdala and along the pathway connecting the two regions (Fig. 7f). This effect significantly correlated with increased depressive-like behavior (Supplementary Fig. 7e). Since myelination is an important determinant of the structural connectivity assessed by DTI[45,46], these results reinforce the notion that repeated activation of the BLA–ACC pathway disrupts the transcriptional program and the survival of mature oligodendrocytes within the ACC, leading to altered connectivity between the two structures.

Prompted by these consistent transcriptomic, histological and imaging findings, we then asked whether myelin dysregulation might also occur during chronic pain. To address this possibility, we took advantage of recent work by Dai and collaborators in a chronic neuropathic pain model (spared nerve injury), which provided transcriptomic analysis in both the PFC and ACC[47]. For consistency, these data were reprocessed in our differential expression analysis pipeline, and compared to those obtained after optogenetic stimulations (using RRHO2). Results uncovered strong similarity across the 2 datasets, including the downregulation of myelin- and oligodendrocyte-related genes and GO terms (Supplementary Fig. 8). Therefore, it is possible that impaired transcriptional activity of the myelination program may also contribute to mood dysregulation during chronic neuropathic pain.

### Sema4a is necessary for the emergence of optogenetically induced depressive-like states

While the relationship between myelination and the expression of depressive-like behaviors has already been documented[48–52], little is known about underlying molecular substrates. A few rodent studies have investigated the effect of depleting myelin sheath protein[53] or positive regulator of myelination[54] on depressive-like behaviors. In contrast, upstream factors that prime myelin deficits and lead to behavioral dysregulation are unknown. Hence, we decided to focus on *Sema4a*, which was upregulated in the ACC of animals showing depressive-like behaviors (Fig. 6i). *Sema4a* has indeed known cytotoxic effects on oligodendrocytes[55,56] and has been associated with white matter defects[57,58].

To block the over-expression of *Sema4a* in our model, we first established a knockdown (KD) approach. Three different shRNAs (#108, 576, 791; Fig. 8a) targeting exon 15 of *Sema4a* were designed, packed into AAV plasmids with mCherry reporter, and transfected in HEK cells overexpressing the targeted *Sema4a* exon, in fusion with eGFP. Among these, we prioritized shRNA-791 because it yielded the most profound KD, as shown by a near complete loss of eGFP signal at 2 days post-transfection (Fig. 8b). To characterize its in vivo efficiency, shRNA-791, was then packaged into an AAV vector and injected in the ACC of adult mice, followed 6 weeks later by qPCR quantification of *Sema4a* (Fig. 8c, d). Compared with a control vector expressing the

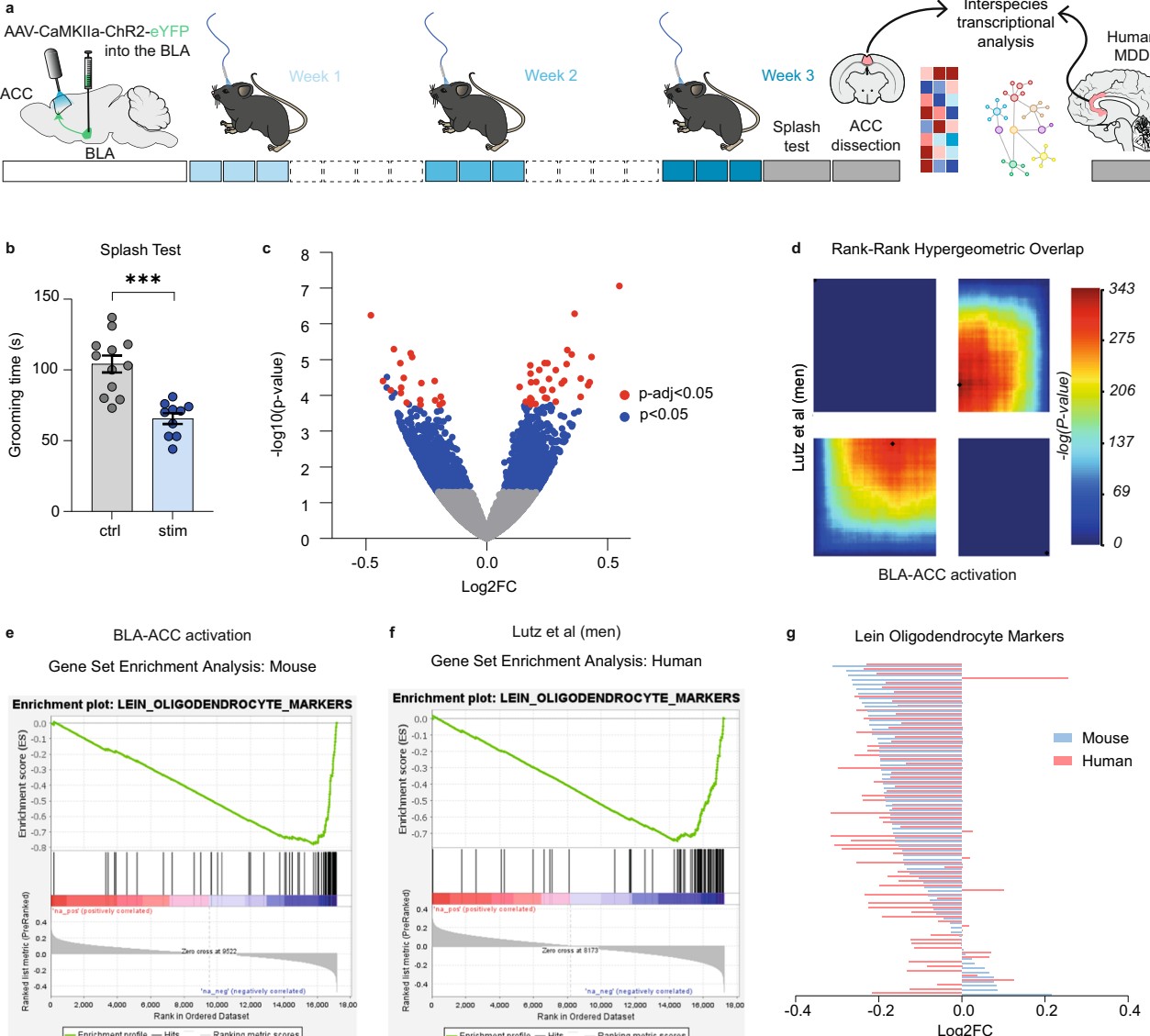

**Fig. 5 | Repeated activation of the BLA–ACC pathway induces transcriptional alterations similar to those observed in human-depressed patients. a** Graphical representation of experimental design, including virus delivery into the BLA, cannula implantation into the ACC, 9 sessions of optogenetic activation, ACC extraction in mice, and transcriptomic analysis in mice and humans. **b** Nine sessions of optogenetic activation of the BLA–ACC pathway decreased grooming behaviors in stimulated animals used for RNA Sequencing (ctrl: $n = 12$ mice; 104.2 ± 6.0; stim: $n = 10$ mice; 65.60 ± 3.80; $P < 0.0001$). **c** Volcano plot showing the 2611 (blue dot, 6.9% of all genes) genes differentially expressed (nominal $P$ values<0.05; Wald test) between stimulated ($n = 10$) and control animals ($n = 12$). Red circles depict the 54 genes that showed a significant dysregulation after multiple testing correction ($P$adj<0.05, Benjamini and Hochberg correction). **d** Rank-Rank Hypergeometric

Overlap (RRHO2) identified shared transcriptomic changes in the ACC across mice and men as a function of optogenetic stimulation (mouse) or a diagnosis of major depressive disorder (MDD). Levels of significance for the rank overlap between men and mice are color-coded, with a maximal one-sided Fisher's Exact Test (FET) $P < 1.0E-343$ for upregulated genes (lower-left panel), and a maximal FET $P < 1.0E-315$ for downregulated genes (upper-right panel). **e**–**g** Gene set enrichment analysis (GSEA) revealed an enrichment for genes specifically expressed by oligodendrocytes and showing evidence of downregulation as a function of optogenetic stimulation (**e**) or MDD diagnosis (**f**). The direction of the changes correlated across mice and humans (**g**; $r^2 = 0.15$, $P = 0.0005$). Behavioral data are mean ± SEM, ***$P < 0.001$, two-sided unpaired $t$ test. Brain cartoons (**a**) were created with Biorender.com. Source data are provided as a Source Data file.

mCherry and a scrambled shRNA, the AAV-shRNA-791 achieved a 62% reduction of *Sema4a* expression, demonstrating its efficacy.

Finally, we hypothesized that knocking down *Sema4a* prior to optogenetic stimulations may prevent the emergence of depressive-like behaviors. Cohorts of mice (Fig. 8e) went through bilateral injections of the ChR2-expressing virus in the BLA (see Supplementary Fig. 9 for injection placement), bilateral injections of the AAV-shRNA-791 vector (or the Scrambled control) in the ACC (Fig. 8f), followed, 2 weeks later, by optogenetic cannula implantation in the ACC. Behavioral testing was performed after 9 stimulations over 3 weeks, corresponding to the 6-week timepoint at which we documented

shRNA-791 in vivo efficiency (Fig. 8d). In the splash test, knocking down *Sema4a* did not affect the decreased grooming observed in stimulated animals (Fig. 8g). However, in the FST (Fig. 8h), we detected a significant interaction between optogenetic stimulations and *Sema4a* KD, with a potent increase in depressive-like behaviors of stimulated mice, which did not occur when *Sema4a* was knocked-down. Analysis of emotional reactivity across ST and FST tests further strengthened these results, as global emotional dysfunction induced by BLA–ACC activation was reversed by *Sema4a* KD (Fig. 8i). Of note, *Sema4a* KD had no effect in unstimulated mice across any tests, indicating that it is not sufficient to trigger emotional dysfunction in naive animals.

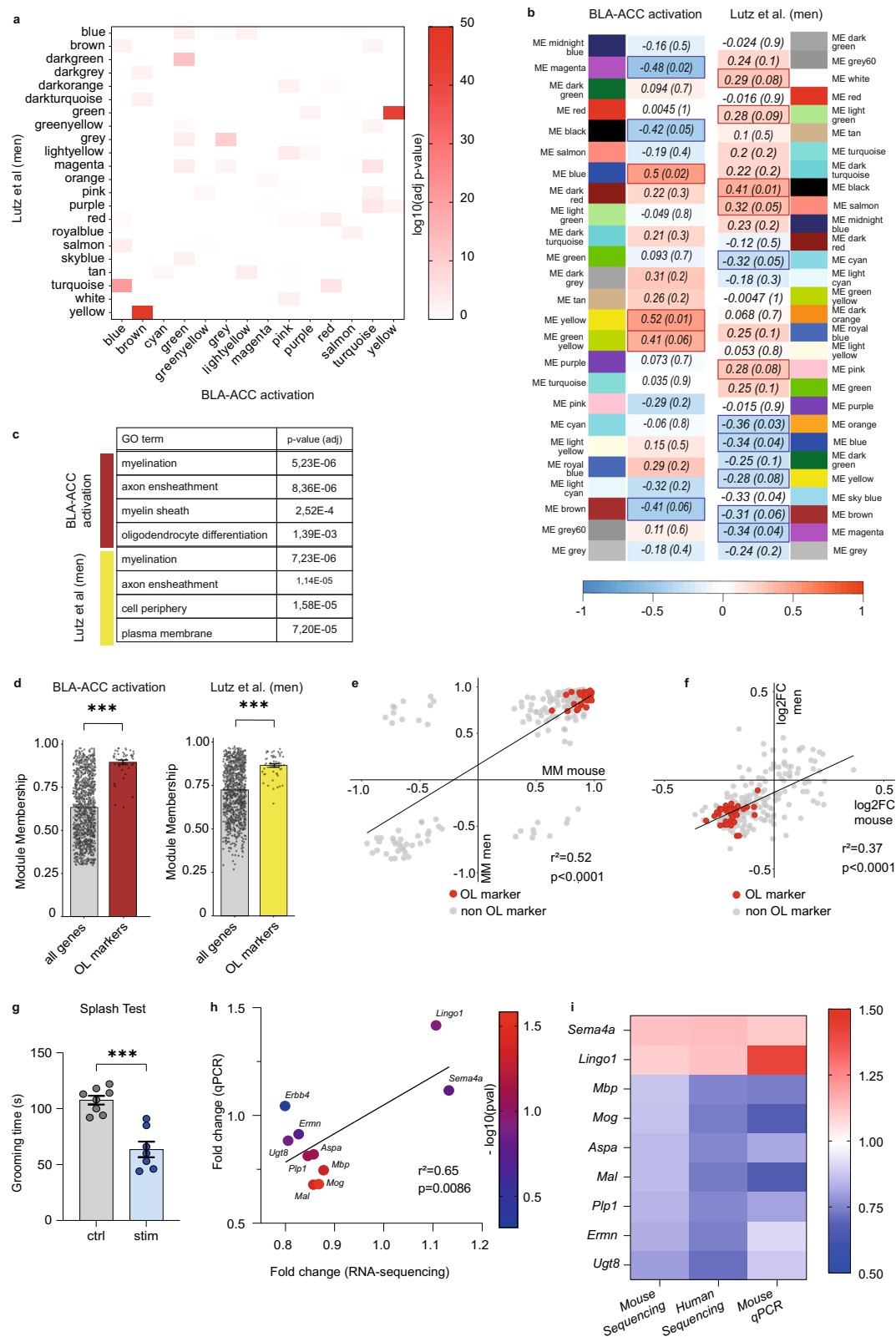

Altogether, these results indicate that silencing *Sema4a* in the ACC prevents the emergence of emotional deficits driven by activation of the BLA–ACC pathway.

## Discussion

Given the complexity of the emotional consequences of chronic pain, disentangling the circuitries involved in its different components is crucial for uncovering new therapeutic leads and strategies. The ACC is considered to play a pivotal role in these processes[10,32,59]. However, while its connectome has been robustly established using neuroanatomical[13,14] and imaging approaches, how it integrates in polysynaptic neuronal circuits that may differentially regulate mood and nociception is poorly understood[60]. To address this gap, here we focused on the BLA–ACC pathway, based on their reciprocal

**Fig. 6 | Gene-network analysis points toward alterations of myelination and oligodendrocyte in mice and men. a** Heatmap representing the level of significance of overlaps between mice and men gene modules (two-sided Fisher Exact Test). The highest overlap (P = 8.36E-49) was obtained for the man/yellow and mouse/brown modules. **b** WGCNA gene modules in the mouse and men ACC. The tables depict associations between each module's eigengene and optogenetic stimulation in mice (left panel), or MDD diagnosis in men (right), and show both r correlation coefficients and P values (in brackets). **c** Gene Ontology analysis for man/yellow and mouse/brown modules, with most significant findings for myelin-related terms in both species (two-sided Fisher Exact Test). **d** The absolute value of the module membership (MM) of oligodendrocytes (OL) markers (OL genes) was significantly higher than the MM of all genes in each module, for both mouse/brown (n = 1088 genes total, n = 43 OL genes; left panel; P value < 2.2E-16) or man/yellow (n = 1049 genes total, n = 52 OL genes, right panel; P value < 2.2E-16) modules (two-sided paired t test). **e** Significant Pearson correlation between mouse/brown and man/yellow MM rankings (r² = 0.52, P = 0.0001). Red dots indicate myelin- and oligodendrocyte-related genes/genes also present in the Lein-Oligodendrocytes-Markers database. **f** Pearson linear regression of fold changes measured by RNA sequencing for men and mouse, showing a significant positive correlation in the direction of the change in expression of the genes in mouse/brown and man/yellow modules (r² = 0.37, P < 0.0001). Red dots indicate myelin and oligodendrocyte-related genes. **g** Repeated activation of the BLA–ACC decreased grooming time in stimulated animals in the splash test (ctrl n = 8 mice; 107.6 ± 3.65; stim n = 7 mice; 63.57 ± 6.96; P < 0.0001). **h** Linear regression of fold changes measured by RNA sequencing and qPCR, showing a significant positive correlation between the two methods (r² = 0.42, P = 0.0025). **i** Downregulation of the myelin-related genes (*Mbp*, P = 0.044; *Mog*, P = 0.026; *Aspa*, P = 0.078; *Mal*, P = 0.038; *Plp1*, P = 0.080; *Ermn*, P = 0.180; *Ugt8*, P = 0.133) and upregulation of inhibitors of the myelination process (*Lingo1*, P = 0.114; *Sema4a*, P = 0.154) were consistent across mice and men in RNA-Sequencing data, and validated in mice by qPCR, after nine stimulations. Data are mean ± SEM, ***P < 0.0001, two-tailed (Splash Test) and one-tailed (qPCR) unpaired t test. Source data are provided as a Source Data file.

anatomical connections and well-known functions. By leveraging 2 optogenetic strategies for neuronal activation or inhibition in the mouse, we manipulated ACC inputs coming from the BLA. Our translational results uncover a critical role of this discrete pathway in mood, both in the context of chronic pain or in the absence of neuropathy.

First, we found that inhibiting the BLA–ACC pathway blocks the expression of depressive-like consequences of chronic pain, without affecting anxiety-like behaviors, aversiveness, or mechanical hypersensitivity. Because acute inhibition is sufficient to produce this effect, depressive-like features seem to be mediated by ongoing hyperactivity of the pathway. In contrast, our previous work had shown that inhibition of CaMKIIa+ cells in the ACC, regardless of their connectivity features, attenuated both anxiety and depressive-like consequences of chronic pain[11]. This suggests that distinct ACC inputs differentially contribute to various aspects of the pain experience, consistent with data from other groups[33,61]. Accordingly, Gao et al showed that, in the sciatic nerve chronic constriction injury model, inhibiting projections from the ACC to the mesolimbic pathway (nucleus accumbens and ventral tegmental area) induced CPP, without affecting evoked pain[33]. In parallel, Hirschberg et al. found that ACC inputs coming from the locus coeruleus (LC) are involved in anxiety-like and aversive consequences of pain[61]. Combined with ours, these results suggest functional segregation, with a LC-ACC-mesolimbic circuit preferentially involved in pain-induced aversion, while BLA and LC inputs targeting the ACC may predominantly mediate depressive- and anxiety-like consequences of chronic pain, respectively.

Second, we show that, in the absence of neuropathy, in naïve mice, repeated but not acute BLA–ACC activation is sufficient to trigger depressive-like effects. This is in line with our previous study showing that chronic but not acute activation of the whole ACC induces emotional deficits[10,25]. The lack of detectable impact of this optogenetic manipulation on anxiety-like responses also strengthens the aforementioned notion that ACC inputs coming from the BLA selectively modulate mood states. Interestingly, activation of BLA terminals in other subparts of the PFC (i.e., areas 25/32, located more rostrally), different from those recruited in the present work (ACC areas 24a/24b), induced anxiety-like behavior[62–65]. These results therefore suggest differential control of emotional responses by multiple pathways that originate in the BLA but target separate cortical regions. Finally, because it is likely that the depressive-like phenotype induced by chronic BLA–ACC activation requires molecular and circuit plasticity, we next conducted open-ended transcriptomic analysis, along with direct comparison with the human MDD signature.

Based on convergent bioinformatic analyses (common DEG, RRHO2, GSEA, WGCNA), we found that transcriptomic changes occurring in our optogenetic paradigm recapitulated a series of adaptations previously associated with human MDD, notably affecting the mitochondria[44,66], chromatin remodeling factors[67], synaptic function, translational regulation[42] or myelination[37,68]. These results document the translational relevance of our paradigm. They also indicate that selective manipulation of a restricted neuronal pathway may represent a valuable strategy to model MDD, at both behavioral and molecular levels. Because recent studies suggest that the various mouse models available for this disorder may capture distinct aspects of its molecular pathophysiology[44], we argue that our optogenetic model provides a complementary approach. While it is based on an artificially induced neuronal hyperactivation, it opens the possibility of modeling some of the effects of internal insults or states, such as chronic pain. Finally, this paradigm, and its potential extension to other neuronal circuits, also enables deciphering what is sufficient for the emergence of mood dysfunction.

Most widespread alterations affected oligodendrocytes and the myelination process, which were consistently identified across human and mouse data. Global downregulation of myelin sheath proteins (*Plp1, Mal, Mog, Mbp*) and enzymes involved in the synthesis of myelin lipids (*Aspa, Ugt8*), as well as upregulation of myelination inhibitors (*Lingo1, Sema4a*) were observed in the ACC of animals displaying depressive-like behaviors. These findings further translated at the cellular level, since a decrease in the number of mature oligodendrocytes was detected after repeated BLA–ACC activation. Of note, one week after the cessation of optogenetic stimulations, when behavioral deficits were no longer present, the loss of Olig2+ cells was reversed. Similarly rapid kinetics were previously characterized in the context of motor learning, with an increase in oligodendrocyte proliferation 4–11 days after mice exercised on a complex wheel[69]. By analogy, quick changes in oligodendroglia might also underlie the emotional fluctuation seen in our optogenetic model, a hypothesis that will need to be further investigated. These results are also congruent with previous studies reporting deficits in myelination[37,70], white matter tract organization[71–73] or oligodendrocytes integrity[50,74], in the ACC of MDD patients (in cross-sectional comparisons of symptomatic patients and euthymic controls). The crucial role of myelination in MDD pathophysiology is further supported by preclinical studies[52,75,76]. Depletion of the myelin sheath component CNP[53], or of the positive regulator of myelination ErbB4[54], as well as cellular depletion of oligodendrocyte progenitors[77], have all been shown to induce emotional dysfunction in rodents. Conversely, the pharmacological compound Clemastine, which enhances oligodendrocyte differentiation and myelination, exerts antidepressant-like effects in socially defeated[76] or isolated[74] mice. Molecular mechanisms mediating such effects, however, remain poorly characterized. Here, using gene-network theory, we prioritized *Sema4a* as one of the most prominently upregulated genes in a myelin-enriched gene module strongly affected by BLA–ACC activation. Previous reports had shown that increased *Sema4a* function contributes to the broad loss of mature oligodendrocytes and demyelination observed in neurological

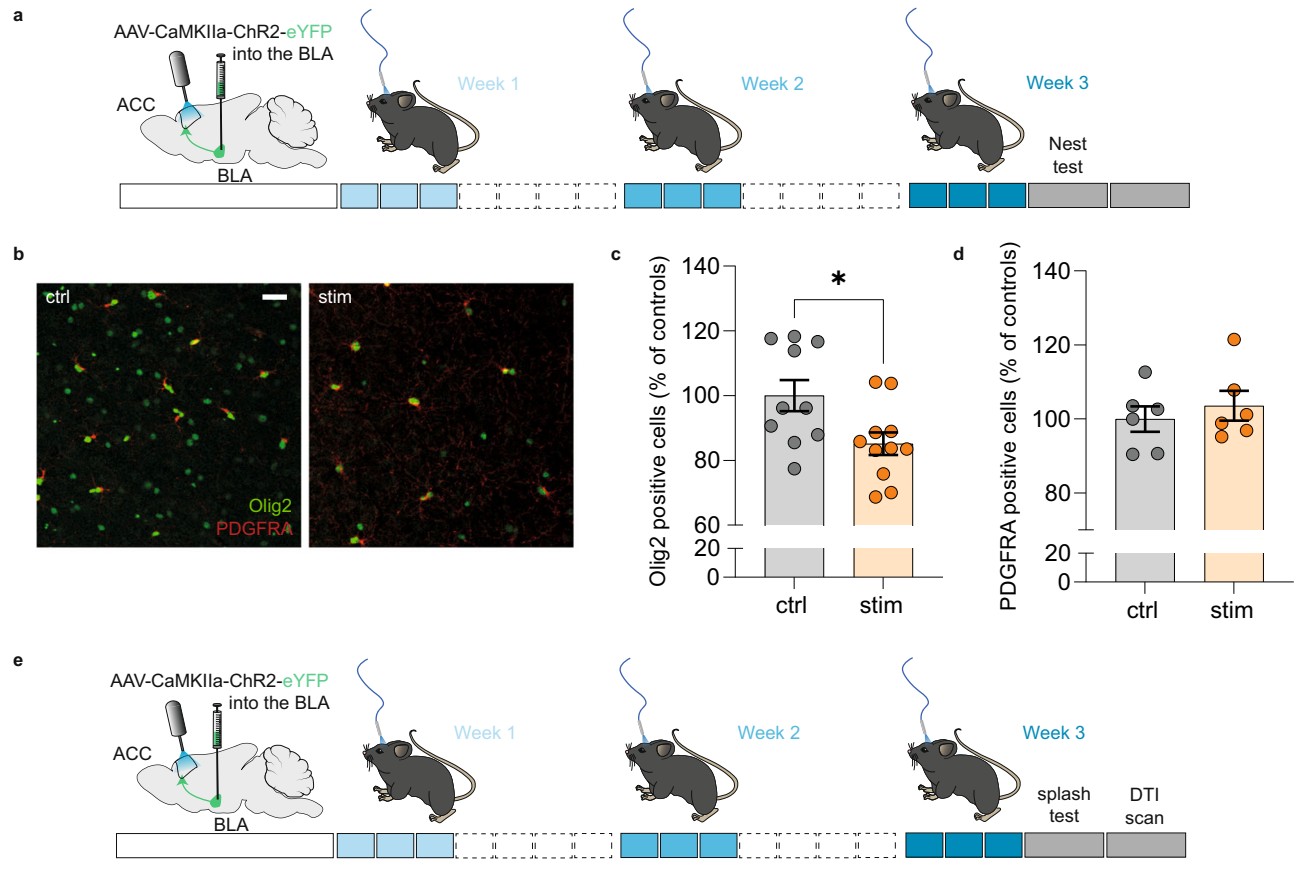

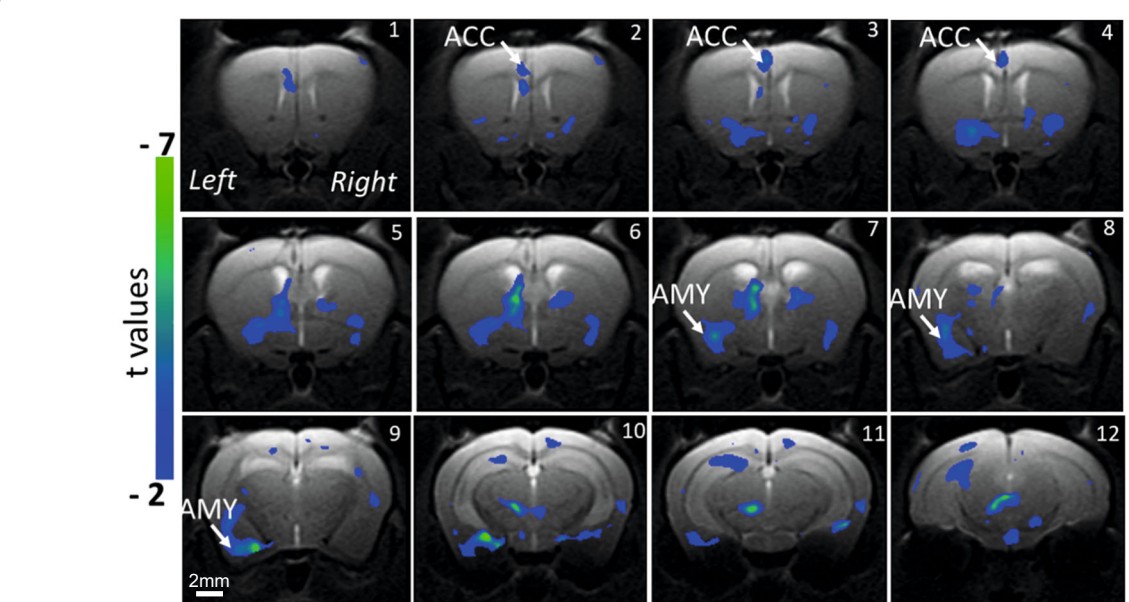

**Fig. 7 | Repeated activation of the BLA–ACC pathway impairs myelination.**
**a** Graphical representation of the experimental design, including virus delivery into the BLA, cannula implantation into the ACC and the nine sessions of optogenetic activation. **b** Representative fluorescence images showing cells that are Olig2 + (green) or PDGFRA + (red) in non-stimulated (left panel) and stimulated mice (right panel). Scale bar = 50 μm. **c**, **d** Quantification of Olig2 and PDGFRA-positive cells showed that nine optogenetic stimulations of the BLA–ACC pathway decreased the number of Olig2+ cells without affecting the number of PDGFRA + cells (**e**: ctrl: $n = 6$ mice; 99.97 ± 3.45; stim: $n = 6$ mice; 103.6 ± 4.00; $P = 0.41$) in the ACC (**d**: ctrl: $n = 10$

mice; 100.00 ± 4.83; stim: $n = 11$ mice; 85.11 ± 3.48; $P = 0.01$). **e** Graphical representation of the experimental design for the DTI experiment, including virus delivery into the BLA, cannula implantation into the ACC and the nine sessions of optogenetic activation. **f** Representative coronal MRI images showing in blue the areas with a significant decrease of FA along the left BLA–ACC pathway in stimulated animals compared to the control group (ctrl: $n = 7$; stim: $n = 6$; GLM $P < 0.001$ uncorrected). Data are mean ± SEM. *$P < 0.05$; **$P < 0.01$. one-sided Mann–Whitney test (Olig2 and PDGFRA quantification). Sagittal mouse brain cartoons (**a**, **e**) were created with Biorender.com. Source data are provided as a Source Data file.

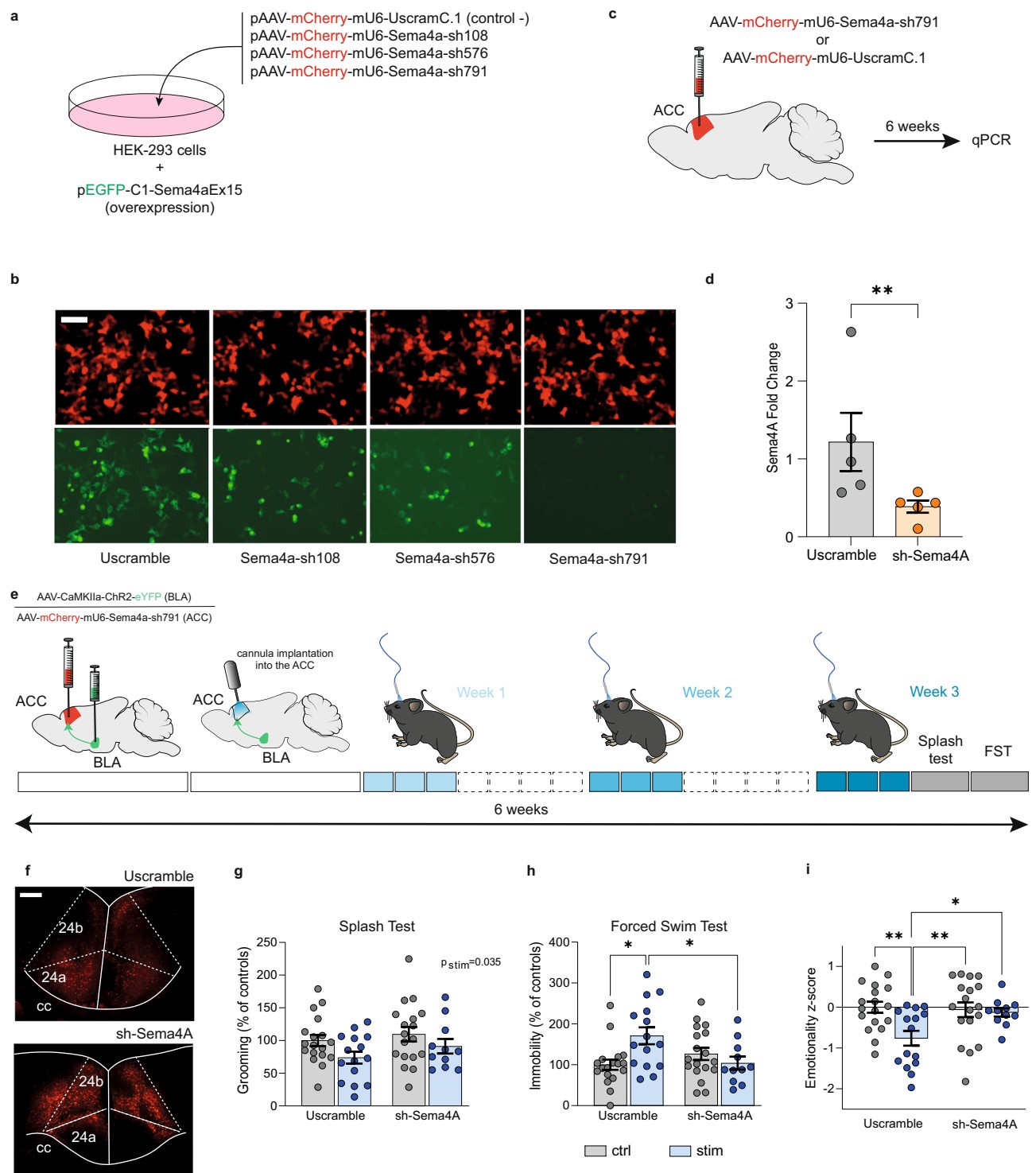

and autoimmune disorders, such as multiple sclerosis[55–58,78]. Our results extend these findings in the context of mood regulation, and involve a milder and possibly more localized dysregulation of *Sema4a* within the ACC. Its knockdown in this region was sufficient to prevent the development of depressive-like behaviors induced by repeated activation of incoming BLA fibers. Therefore, we document those changes in *Sema4a* expression may play a pivotal role in mood regulation, and represent a putative therapeutic target. Among others, avenues for future work will include characterizing the mechanisms by which *Sema4a* modulates oligodendrocytes, identifying in which ACC cell types such processes are recruited by BLA afferents, and

describing putative sex differences in these aspects of pain and emotional processing.

Accordingly, a limitation of the present study is that we focused on males only. Since sexual differences have been described in relation to chronic pain and associated emotional dysregulation[79–81], future studies will be necessary to investigate sex as a biological variable in behavioral and transcriptomic consequences of various models (including NPID), as well as in interactions among structures such as the BLA and the ACC (with no concrete evidence available to date).

In conclusion, our results demonstrate that the BLA–ACC pathway critically mediates the interplay between chronic pain and depression.

**Fig. 8 | Semaphorin-4A is essential for the depressive-like behaviors induced by the activation of the BLA–ACC pathway. a** Graphical representation of the experimental design for shRNA validation in vitro. HEK-293 cells were transfected with plasmids expressing the mouse *Sema4a* exon 15 (green), and 1 out of 3 shRNAs tested for knockdown efficiency (red; with 1 scrambled shRNA as control). **b** Representative fluorescence images showing that the pAAV-mCherry-mU6-Sema4A-sh791 plasmid was the most efficient. Scale bar =50 μm. **c** Graphical representation of the bilateral virus injection in the ACC for in vivo validation of the selected sh791, inserted into an AAV vector. **d** qPCR analysis showing a down-regulation of *Sema4a* expression level in the ACC of mice injected with Sema4A-sh791 (Uscramble: $n = 5$ mice; 1.22 ± 0.37; sh-Sema4A: $n = 5$ mice; 0.39 ± 0.07; $P = 0.0079$). **e** Graphical representation of the experimental design, including bilateral virus delivery in the BLA (AAV5-CaMKIIa-ChR2(H134R)-EYFP) and the ACC (rAAV-mCherry-scrambleUsh or rAAV-mCherry-Sema4A-sh791), cannula implantation, optogenetic stimulation and behavioral testing. **f** Representative images of mCherry+ cells in the ACC after the injection of UScramble (upper panel) or

sh-Sema4A (lower panel). Scale bars =100 μm. **g**–**i** Effect of *Sema4a* knockdown on optogenetically induced emotional deficits (UScramble-Ctrl: $n = 18$ mice; UScramble-Stim: $n = 15$ mice; sh-Sema4A-Ctrl: $n = 18$ mice; sh-Sema4A-Stim: $n = 11$ mice). Grooming time in the ST was decreased by repeated activation of the BLA–ACC pathway (**g**: $F_{(1,58)} = 4.623$; $P = 0.036$) but was not affected by knocking down *Sema4a* in the ACC (**g**: $F_{(1,58)} = 1.694$; $P = 0.20$). Knocking down *Sema4a* in the ACC counteracted the increased immobility time observed in stimulated animals in the FST (**h**: $F_{(1,58)} = 8.032$; $P = 0.006$; post hoc: Uscramble-Ctrl<Uscramble-Stim; $P < 0.05$; sh-Sema4A-Stim<Uscramble-Stim; $P = 0.05$). Knocking down *Sema4a in the ACC* normalized the emotionality z-score (**i**: $F_{(1,58)} = 4.39$; $P = 0.041$; post hoc: Uscramble-Ctrl>Uscramble-Stim; $p < 0.01$; UScramble-Stim<sh-Sema4A-Ctrl; $P < 0.01$ UScramble-Stim<sh-Sema4A-Stim; $P < 0.05$; sh-Sema4A-Ctrl=sh-Sema4A-Stim; $p > 0.05$). Data are mean ± SEM. *$P < 0.05$; **$P < 0.01$; one-tailed Mann–Whitney test (*Sema4A* quantification), two-way ANOVA (Stimulation × KnockDown; ST, FST and z-score). Sagittal mouse brain cartoons (**c**, **e**) were created with Biorender.com. Source data are provided as a Source Data file.

By combining animal and human studies, we define the behavioral relevance of this pathway, and uncover the essential role of impaired myelination and *Sema4a* signaling in depression.

## Methods

### Animals
Experiments were conducted using male adult C57BL/6J (RRID: IMSR JAX: 000664) mice (Charles River, France), 8 weeks old at the beginning of experimental procedures, group-housed with a maximum of five animals per cage and kept under a reversed 12 h light/dark cycle (room temperature 24 ± 1 °C, humidity around 50%). After the optic fiber implantation, animals were single housed to avoid possible damage to the implant. We conducted all the behavioral tests during the dark phase, under red light. Our animal facility (Chronobiotron) is registered for animal experimentation (Agreement A67-2018-38), and protocols were approved by the local ethical committee of the University of Strasbourg (CREMEAS, APAFIS8183-2016121317103584).

### Surgical procedures
Surgical procedures were performed under zoletil/xylazine anesthesia (zoletil 50 mg/ml, xylazine 2.5 mg/ml; i.p., 4 ml/kg, Centravet). For stereotaxic surgery, a local anesthetic was delivered subcutaneously at the incision site (bupivacaine, 2 mg/kg).

### Neuropathic pain induction: cuff surgery
For the BLA–ACC inhibition study, we used a well-characterized chronic pain-induced depression model[11,23]. Before surgery, mice were assigned to experimental groups so that these groups did not initially differ in mechanical nociceptive threshold or body weight. Chronic neuropathic pain was induced by placing a 2 mm polyethylene tubing (Cuff, Harvard Apparatus, Les Ulis, France) around the right common branch of the sciatic nerve[24]. The Sham group underwent the same procedure without cuff implantation.

### Virus injection
After general anesthesia, mice were placed in a stereotaxic frame (Kopf Instruments). In total, 0.5 μl of AAV5-CaMKIIa-ChR2(H134R)-EYFP or AAV5-CaMKIIa-eArchT3.0-EYFP (UNC Vector core) were injected bilaterally into the BLA using a 5-μl Hamilton syringe (0.05 μl/min, coordinates for the BLA, anteroposterior (AP): −1.4 mm from bregma, lateral (L): +/−3.2 mm, dorsoventral (DV): 4 mm from the brain surface). The same method was used to bilaterally inject the rAAV-mCherry-scrambleUsh or the rAAV-mCherry-Sema4A-sh791 into the ACC (coordinates: AP: + 0.7 mm, L: +/−0.2 mm, DV: −2 mm, from the bregma). After injection, the 32-gauge needle remained in place for 10 min before being removed and then the skin was sutured. Following surgery, animals were left undisturbed for at least two weeks before cannula implantation. To check viral injection localization at the end of

the experiment, animals were anesthetized with Euthasol (182 mg/kg) and perfused with 30 mL of 0.1 M phosphate buffer (PB, pH 7.4) followed by 100 mL of 4% paraformaldehyde solution (PFA) in 0.1 M PB. Brains were extracted and postfixed overnight and kept at 4 °C in 0.1 M PB saline (PBS) until cutting. Coronal sections (40 μm) were obtained using a vibratome (VT 1000 S, Leica, Deerfield, IL) and were serially collected in PBS. Sections were serially mounted with Vectashield medium (Vector laboratories) and localization of the fluorescence was checked using an epifluorescence microscope (Nikon 80i, FITC filter). Only animals well-injected bilaterally in the BLA were kept for further analyses.

### Tracer injections
Analysis of afferent neurons from the BLA to the ACC was performed by injecting the retrograde tracer hydroxystilbamidine methanesulfonate (FluoroGold®, Molecular Probes, 0.5 μl) bilaterally in the ACC, using a microsyringe pump controller (UMC4, World precision instruments) and a 5-μl Hamilton syringe (100 nl/min, coordinates for the ACC, AP: + 0.7 mm from bregma, L: +/−0.2 mm, DV: −2mm from the bregma). 7 days after the tracer injection, mice were anesthetized with Euthasol (182 mg/kg) and perfused with 30 ml of 0.1 M phosphate buffer (PB, pH 7.4) followed by 150 ml of 4% paraformaldehyde solution (PFA) in 0.1 M PB. Brains were removed, postfixed overnight in PFA at 4 °C, and then kept at 4 °C in 0.1 M PB saline (PBS, pH 7.4) until cutting. Coronal sections (40 μm) were obtained with a Vibratome (VT 1000 S, Leica, Deerfield, IL) and serially collected in PBS.

### Optic fiber cannula implantation
At least two weeks after virus injection, both control and stimulated animals underwent a unilateral optic fiber cannula implantation into the ACC. The optic fiber cannula was 1.7-mm long and 220 μm in diameter. The cannula was inserted 1.5-mm deep in the brain at the following coordinates: AP: + 0.7 mm L: +/− 0.2 mm (MFC 220/250–0.66 1.7 mm RM3 FLT, Doric Lenses)[10]. For behavioral experiments, cannulas were implanted in the left hemisphere in half of each experimental group, whereas the other half received the implant in the right hemisphere. For DTI protocol, all mice were implanted in the left hemisphere. Animals were then left undisturbed for 3 to 7 days before undergoing optogenetic stimulation and behavioral testing.

### Optogenetic stimulation procedures
**BLA–ACC pathway inhibition.** The BLA–ACC pathway was inhibited using a green light-emitting laser with a peak wavelength of 520 nm (Miniature Fiber Coupled Laser Diode Module, Doric Lenses). From the light source, the light passes through the fiber optic patch cable (MFP 240/250/900-0.63 0.75 m FC CM3, Doric Lenses) to the implant cannula. Green light was delivered in a continuous manner during 5 min prior (Forced Swim Test and Dark/light test) or during behavioral

testing (Splash test, Novelty-Suppressed Feeding Test and von Frey test). The onset and end of stimulation were manually controlled. Light intensity was measured before implantation at the fiber tip using a photodetector (UNO, Gentec, Quebec, Canada) and was set at 16 mW. All control animals underwent the same procedure but the light remained switched off.

**BLA–ACC pathway activation.** Activation of the BLA–ACC pathway was achieved using a blue light-emitting diode (LED) with a peak wavelength of 463 nm (LEDFRJ-B FC, Doric Lenses). From the light source, the light passes through the fiber optic patch cable (MFP 240/250/900-0.63 0.75 m FC CM3, Doric Lenses) to the implant cannula. Blue light was delivered by pulses generated through a universal serial bus connected transistor-transistor logic pulse generator (OPTG 4, Doric Lenses) connected to a LED driver (LEDRV 2CH v.2, Doric Lenses). Transistor-transistor logic pulses were generated by an open-source software developed by Doric Lenses (USBTTL V1.9). For acute activation of the BLA–ACC pathway, the stimulation was delivered during behavioral testing. For repeated activation of the pathway, stimulated animals received repetitive stimulation sequences of 3 s consisting of 2 s at 10 Hz with 10 ms pulses and 1 s without stimulation. The whole sequence was repeated for 20 min each day for 3 consecutive days for 3 weeks. Each stimulation session was performed in the animals' home cage. Light intensity was measured as described above and set between 3 mW and 5 mW. All control animals underwent the same procedure, but the light remained switched off.

### Behavioral assessment
Behavioral testing was performed during the dark phase, under red light. While each mouse went through different tests, they were never submitted to more than 3 tests per week. The forced swimming test (FST) was always performed as a final test. Body weights were measured weekly. Experimenters were always blind to the pain conditions and optogenetic stimulation, except during the von Frey and Splash test concerning the inhibition paradigm condition and during the RTA, NSF and Splash test concerning the acute activation paradigm. A summary of the tests performed by each animal cohort for repeated activation paradigm is provided in Supplementary Data File 6.

**Nociception-related behavior.** The mechanical threshold of hind paw withdrawal was evaluated using von Frey hairs (Bioseb, Chaville, France)[23]. Mice were placed in clear Plexiglas® boxes (7 × 9 × 7 cm) on an elevated mesh screen[24]. Filaments were applied to the plantar surface of each hind paw in a series of ascending forces (0.4–8 g). Each filament was tested five times per paw, being applied until it just bent, and the threshold was defined as 3 or more withdrawals observed out of the five trials. All animals were tested before the cuff surgery to determine the basal threshold every week after cuff surgery to ensure the development of mechanical allodynia and during optogenetic stimulation to assess the effect of the inhibition of the BLA–ACC on mechanical hypersensitivity.

**Locomotor activity.** Spontaneous locomotor activity was monitored for each experimental group. Mice were individually placed in activity cages (32 × 20 cm floor area, 15 cm high) with seven photocell beams. The number of beam breaks was recorded over 30 min using Polyplace software (Imetronic, Pessac, France).

**Real-time place avoidance (RTA) and conditioned place preference (CPP).** The apparatus consists of two connected Plexiglas chambers (size 20 × 20 ×30 cm) distinguished by the wall patterns. On the first day (pre-test), animals are free to explore the apparatus for 5 min (CPP) or 10 min (RTA), and the time spent in each chamber is measured to control for the lack of spontaneous preference for one compartment. Animals spending more than 75% or less than 25% of the total time in

one chamber were excluded from the study. For RTA, the second day (test), animals are plugged to the light source placed between the two chambers and let free to explore for 10 min. Light is turned on when the mouse enters its head and forepaws in the stimulation-paired chamber and turned off when it quits the compartment. The total time spent in the stimulation-paired chamber is measured. For CPP, on the second and third days (conditioning), animals are maintained for 5 min in one chamber, where optogenetic stimulation occurs, and 4 h later placed for 5 min in the other chamber, without optogenetic stimulation. On the 4th day (test), the time spent in each chamber is recorded for 5 min.

**Dark-light box test.** The apparatus consists of connected light and dark boxes (18 × 18 × 14.5 cm each). The lit compartment was brightly illuminated (1000 lux). This test evaluates the conflict between the exploratory behavior of the rodent and the aversion created by bright light. Mice were placed in the dark compartment in the beginning of the test, and the time spent in the lit compartment was recorded during 5 min[10]. For inhibition experiment, the test was performed immediately after the light stimulation.

**Novelty-suppressed feeding test.** The apparatus consisted of a 40 × 40 × 30 cm plastic box with the floor covered with 2 cm of sawdust. Twenty-four hours prior to the test, food was removed from the home cage. At the time of testing, a single pellet of food was placed on a paper in the center of the box. The animal was then placed in the corner of the box and the latency to eat the pellet was recorded within a 5-min period. This test induces a conflict between the drive to eat the pellet and the fear of venturing in the center of the box[82]. For inhibition experiments, optogenetic stimulation was conducted during the test.

**Splash test.** This test, based on grooming behavior, was performed as previously described[24,82]. Grooming duration was measured for 5 min after spraying a 20% sucrose solution on the dorsal coat of the mouse. Grooming is an important aspect of rodent behavior and decreased grooming in this test is considered related to the loss of interest in performing self-oriented minor tasks[83]. For inhibition experiments, optogenetic stimulation was conducted during the test.

**Nest test.** This test, based on a rodent innate behavior, was performed in cages identical to the home cages of animals. Each mouse was placed in a new cage with cotton square in the center. Water and food were provided ad libitum. After 5 h, mice were placed back in their original cages and pictures of the constructed nest were taken. A score was given blindly to each nest as follows: 0 corresponds to an untouched cotton square, 1 to a cotton square partially shredded, 2 if the cotton is totally shredded but not organized, 3 if cotton is totally shredded and organized in the center of the cage, 4 if the cotton is totally shredded and shows a well-organized shape in the corner of the cage, like a nest[84,85].

**Forced swim test.** FST[86] was conducted by gently lowering the mouse into a glass cylinder (height 17.5 cm, diameter 12.5 cm) containing 11.5 cm of water (23–25 °C). The test duration was 6 min. The mouse was considered immobile when it floated in the water, in an upright position, and made only small movements to keep its head above water. Since little immobility was observed during the first 2 min, the duration of immobility was quantified over the last 4 min of the 6 min test. Concerning inhibition experiments, the test was performed just after the stimulation.

### Ex vivo electrophysiological recordings
We performed whole-cell patch-clamp recordings of BLA neurons or ACC pyramidal neurons. In the BLA, we recorded from eYFP-expressing neurons of mice bilaterally injected with an AAV driving the

expression of either the archaerhodopsin ArchT3.0 or the channelr-hodopsin 2 under control of the CaMKIIa promoter (with AAV5-CamKIIa-ArchT3.0-EYFP and AAV5-CaMKIIa-ChR2(H134R)-EYFP, respectively). In the ACC, we recorded from pyramidal neurons surrounded by eYFP-positive fibers. For these experiments, mice were anesthetized with urethane (1.9 g/kg) and killed by decapitation, their brain was removed and immediately immersed in cold (0–4 °C) sucrose-based ACSF containing the following (in mM): 2 kynurenic acid, 248 sucrose, 11 glucose, 26 $NaHCO_3$, 2 KCl, 1.25 $KH_2PO_4$, 2 $CaCl_2$, and 1.3 $MgSO_4$ (bubbled with 95% $O_2$ and 5% $CO_2$). Transverse slices (300-µm thick) were cut with a vibratome (VT1000S, Leica). Slices were maintained at room temperature in a chamber filled with ACSF containing the following (in mM): 126 NaCl, 26 $NaHCO_3$, 2.5 KCl, 1.25 $NaH_2PO_4$, 2 $CaCl_2$, 2 $MgCl_2$, and 10 glucose (bubbled with 95% $O_2$ and 5% $CO_2$; pH 7.3; 310 mOsm measured). Slices were transferred to a recording chamber and continuously superfused with ACSF saturated with 95% $O_2$ and 5% $CO_2$. BLA neurons expressing eYFP were recorded in the whole-cell patch-clamp configuration. Recording electrodes (3.5–4.5 MΩ) were pulled from borosilicate glass capillaries (1.2 mm inner diameter, 1.69 mm outer diameter, Warner Instruments, Harvard Apparatus) using a P1000 electrode puller (Sutter Instruments). Recording electrodes were filled with, in mM: 140 KCl, 2 $MgCl_2$, 10 HEPES, 2 MgATP; pH 7.3. The pH of intrapipette solutions was adjusted to 7.3 with KOH, and osmolarity to 310 mOsm with sucrose. BLA or ACC were illuminated with the same system used for the in vivo experiments (see above) triggered with WinWCP 4.3.5, the optic fiber being localized in the recording chamber at 3 mm from the recorded neuron. The holding potential was fixed at −60 mV. Recordings were acquired with WinWCP 4.3.5 (courtesy of Dr. J. Dempster, University of Strathclyde, Glasgow, UK). All recordings were performed at 34 °C.

## MRI data acquisition

Animals were scanned 48 h after the 9th stimulation. Mouse brain resting-state functional MRI scans were performed with a 7 T Bruker BioSpec 70/30 USR animal scanner, a mouse head adapted room temperature surface coil combined with a volume transmission coil for the acquisition of the MRI signal and ParaVision software version 6.0.1 (Bruker, Ettlingen, Germany). Imaging was performed at baseline, 2 weeks, and 8 weeks after peripheral nerve injury in the cuff model (Cuff $n = 7$ and Sham $n = 7$). For rs-fMRI the animals were briefly anesthetized with isoflurane for initial animal handling. The anesthesia was further switched to medetomidine sedation (MD, Domitor, Pfizer, Karlsruhe, Germany), initially induced by a subcutaneous (sc) bolus injection (0.15 mg MD per kg body weight (kg bw) in 100 µl 0.9% NaCl-solution). 10 min later, the animals received a continuous sc infusion of MD through an MRI-compatible catheter (0.3 mg/kg bw/h) inserted at the mouse shoulder level. During the whole acquisition a 2-mm thick agar gel (2% in NaCl) was applied on the mouse head to reduce any susceptibility artifacts arising at the coil/tissue interface. Respiration and body temperature were monitored throughout the imaging session. Acquisition parameters for rs-fMRI were: single shot GE-EPI sequence, 31 axial slices of 0.5 mm thickness, FOV = 2.12 × 1.8 cm, matrix=147 × 59, TE/TR = 15 ms/2000ms, 500 image volumes, $0.14 \times 0.23 \times 0.5 \, mm^3$ resolution. Acquisition time was 16 min. Morphological T2-weighted brain images (resolution of $0.08 \times 0.08 \times 0.4 \, mm^3$) were acquired with a RARE sequence using the following parameters: TE/TR = 40 ms/4591 ms; 48 slices, 0.4-mm slice thickness, interlaced sampling, RARE factor of 8, 4 averages; an acquisition matrix of 256 × 256 and FOV of $2.12 \times 2 \, cm^2$. Brain Diffusion Tensor MRI (DT-MRI) acquisition in the BLA−ACC optogenetically stimulated animals was performed with the 7 T animal scanner, but using a combination of a transmit−receive volume coil (86 mm) and a mouse brain adapted loop surface coil allowing the passage of the optogenetic cannulas (MRI, Bruker, Germany). Stimulated animals ($n = 6$) and their controls ($n = 7$) were brain imaged under isoflurane anesthesia (1.5% for maintenance) using a 4-shot DTI-EPI

sequence (TE/TR = 24 ms/3000 ms), 8 averages; with diffusion gradients applied along 45 nonlinear directions, gradient duration [δ] =5.6 ms and gradient separation [Δ] = 11.3 ms and a b-factor of 1000 s/mm². Images with a b-factor = 0 s/mm² were also acquired. In total, 30 axial slices with 0.5 mm thickness were acquired, covering the whole brain with a FOV of $1.9 \times 1.6 \, cm^2$ and an acquisition matrix of 190 × 160 resulting in an image resolution of $0.1 \times 0.1 \times 0.5 \, mm^3$. The total acquisition time was 1 h and 20 min.

## MRI data processing

Rs-fMRI images were spatially normalized into a template using Advanced Normalization Tools (ANTs) software[87] using SyN algorithm and smoothed (FWHM = $0.28 \times 0.46 \times 1 \, mm^3$) with SPM8. Seed-based functional connectivity analysis was performed with a MATLAB tool developed in-house. Regions of interest (ROI) were extracted from Allen Mouse Brain Atlas[39] which were later normalized into the template space. Resting-state time series were de-trended, band-pass filtered (0.01–0.1 Hz) and regressed for the cerebrospinal fluid signal from the ventricles. Principal component analysis (PCA) of the BOLD time courses across voxels within a given ROI was performed, and first principal component accounting for the largest variability was selected as the representative time course for further analysis. Spearman correlations between the PCA time course of single ROIs and each voxel of the brain was computed at the group and individual levels and r values were converted to z using Fisher's r-to-z transformation. Individual connectivity maps for baseline rs-fMRI acquisitions were subtracted from 2 and 8 PO weeks counterparts for each subject. Baseline subtracted connectivity maps were subsequently used for two sample $t$ test with SPM8 to perform group comparison. Family-wise error rate (FWER) correction was applied at the cluster level ($P < 0.05$) for each statistical image. Preprocessing of diffusion-weighted images included denoising[88], removal of Gibbs ringing artifacts[89], motion correction[90], and bias field inhomogeneity correction[91]. Diffusion tensor was estimated[71] using weighted least-squares (WLS) approach and the following tensor-derived parameters were computed: fractional anisotropy (FA), axial diffusivity (AD), mean diffusivity (MD) and radial diffusivity (RD). All these processing steps were done using MRtrix3 (https://www.mrtrix.org); except the motion correction step, done using Advanced Normalisation Tools (ANTs, http://stnava.github.io/ANTs/). Based on b = 0 s/mm² images (i.e., the volume without diffusion weighting), these images were then spatially registered in a common space using the SyN registration method of ANTs to build a study-specific template, which was then registered onto the Allen Brain Atlas template. Each mouse tensor-derived maps was warped in this common space. These registered images were finally smoothed by a 0.5 mm full-width half maximum (FWHM) Gaussian kernel. Inter-group differences for all the DTI-derived parameters were assessed using the SPM12 General Linear Model (GLM). The results were analyzed according to a level of statistical significance, $P < 0.001$ without correction. Further, correlational analyses were performed between the DTI metrics (voxel level) and the results from splash tests (statistical significance was $P < 0.001$, uncorrected).

## Immunohistochemistry

**c-Fos immunoperoxidase.** On the 5th day after the last stimulation, animals were stimulated once with the same procedure as described before (for BLA−ACC activation). 90 min later, animals were anesthetized with Euthasol (182 mg/kg) and perfused with 30 ml of 0.1 M PB (pH 7.4) followed by 100 ml of 4% PFA in 0.1 M PB. Brains were removed, postfixed overnight and kept at 4 °C in 0.1 M PBS (pH 7.4) until cutting. Coronal sections (40 µm) were obtained using a vibratome (VT 1000 S, Leica, Deerfield, IL) and were serially collected in PBS. Sections were incubated 15 min in a 1% $H_2O_2$/50% ethanol solution and washed in PBS (3 × 10 min). Sections were then pre-incubated in PBS containing Triton X-100 (0.3%) and donkey serum (5%) for 45 min.

Sections were then incubated overnight at room temperature in PBS containing Triton X-100 (0.3%), donkey serum (1%) and rabbit anti-c-Fos (1:10,000, Santa Cruz Biotechnology, E1008). Sections were then washed in PBS (3x10min), incubated with biotinylated donkey anti-rabbit secondary antibody (1:300) in PBS containing Triton X-100 (0.3%), donkey serum (1%) for 2 h and washed in PBS (3 × 10 min). Sections were incubated with PBS containing the avidin-biotin-peroxidase complex (ABC kit; 0.2% A and 0.2% B; Vector laboratories) for 90 min. After being washed in Tris-HCl buffer, sections were incubated in 3,3'diaminobenzidine tetrahydrochloride (DAB) and $H_2O_2$ in Tris-HCl for approximately 4 min and washed again. Sections were serially mounted on gelatin-coated slides, air dried, dehydrated in graded alcohols, cleared in Roti-Histol (Carl Roth, Karlsruhe, Germany) and coverslipped with Eukitt. c-Fos immunohistochemistry then allowed controlling for both the implant location and the activation of the ACC by the optogenetic procedure. Animals having c-Fos induction outside of the ACC, for instance, in the motor cortex, were excluded from analysis.

**c-Fos immunofluorescence.** One week after the Fluorogold injection in the ACC, animals were anesthetized with Euthasol (182 mg/kg) and perfused with 30 ml of 0.1 M PB (pH 7.4) followed by 100 ml of 4% PFA in 0.1 M PB. Brains were removed, postfixed overnight, and kept at 4 °C in 0.1 M PBS (pH 7.4) until cutting. Coronal sections (40 μm) were obtained using a vibratome (VT 1000 S) and were serially collected in PBS. Sections were washed in PBS (3 × 10min) and pre-incubated in PBS containing Triton X-100 (0.3%) and donkey serum (5%) for 45 min. Sections were then incubated overnight at room temperature in PBS containing Triton X-100 (0.3%), donkey serum (1%) and rabbit anti-c-Fos (1:1000, Synaptic System, 226-003). Sections were then washed in PBS (3 x 10min), incubated with Alexa fluor 594 donkey anti-rabbit secondary antibody (1:400) in PBS containing Triton X-100 (0.3%), donkey serum (1%) for 2 h and washed in PBS (3 × 10 min). Sections were finally serially mounted with vectashield medium (Vector laboratories).

**Olig2 immunofluorescence.** Twenty-four hours after the last behavioral test, animals were anesthetized with Euthasol (182 mg/kg) and perfused with 30 ml of 0.1 M PB (pH 7.4) followed by 100 ml of 4% PFA in 0.1 M PB. Brains were removed, postfixed overnight and kept at 4 °C in 0.1 M PBS (pH 7.4) until cutting. Coronal sections (40 μm) were obtained using a vibratome (VT 1000 S) and were serially collected in PBS. Sections were washed in PBS (3 × 10 min) and pre-incubated in PBS containing Triton X-100 (0.3%) and donkey serum (5%) for 1 h. Sections were then incubated overnight at +4 °C in PBS containing Triton X-100 (0.3%) and rabbit anti-Olig2 (1:200, Merck-Millipore, AB9610). Sections were then washed in PBS (3x10min), incubated with Cy3 donkey anti-rabbit secondary antibody (1:400, Jackson Immu-noResearch, 711-165-152) in PBS containing Triton X-100 (0.3%) for 2 h and washed in PBS (3x10min). Sections were finally serially mounted with Fluoromount-G (Electron Microscopy Sciences, EM-17984-25).

**Olig2 and PDGFRa co-staining.** Twenty-four hours after the last behavioral test, animals were anesthetized with Euthasol (182 mg/kg) and perfused with 30 ml of 0.1 M PB (pH 7.4) followed by 100 ml of 4% PFA in 0.1 M PB. Brains were removed, postfixed overnight and kept at 4 °C in 0.1 M PBS (pH 7.4) until cutting. Coronal sections (40 μm) were obtained using a vibratome (VT 1000 S) and were serially collected in PBS. Sections were mounted on a Superfrost slide (Epredia reference J1800AMNZ) and dried for 30 min at room temperature. Slides were dipped 2 times in H2O and incubated for 15 min at 98 °C in RNAscope target retrieval buffer (ACDBio ref. 322001). Slides were quickly rinsed in H2O, washed in PBS (2×10 min) and pre-incubated in PBS containing Triton X-100 (0.3%) and donkey serum (5%) for 1 h and 30 min. Sections were then incubated overnight at +4 °C in PBS containing Triton

X-100 (0.3%), goat anti-PDGFRa (1:200, RandD systems, AF1062) and rabbit anti-Olig2 (1:200, Merck-Millipore, AB9610). Sections were then washed in PBS (3 × 10 min), incubated with Cy3 donkey anti-goat secondary antibody (1:400, Jackson ImmunoResearch, 705-165-147) and Alexa Fluor 647 donkey anti-rabbit (1:400, Invitrogen, A31573) in PBS containing Triton X-100 (0.3%) for 2 h and washed in PBS (3 × 10 min). Sections were finally serially mounted with Fluoromount-G (Electron Microscopy Sciences, EM-17984-25).

## Fluorogold and c-Fos quantification

Single-layer images were acquired using a laser-scanning microscope (confocal Leica SP5 Leica Microsystems CMS GmbH) equipped with a ×20 objective. Excitation wavelengths were sequentially diode 405 nm, argon laser 488 nm, and diode 561 nm. Emission bandwidths are 550-665 nm for Fluorogold fluorescence and 710–760 nm for Alexa594 signal. Segmentation and classification of c-Fos+ cells were performed from three sections for each animal using a deep learning model. The model was trained from scratch for 400 epochs on 10 paired image patches (image dimensions: (160,160), patch size: (160,160)) with a batch size of 2 and a mae loss function, using the StarDist 2D ZeroCostDL4Micnotebook (v1.11)[92]. Key python packages used include tensorflow (v0.1.12), Keras (v2.3.1), csbdeep (v0.6.2), numpy (v1.19.5), cuda (v11.0.221). The training was accelerated using a Tesla K80 GPU and the dataset was augmented by a factor of 4. Segmentation and classification of Fluorogold signal was done using Stardist 2D_versatile_fluo[93,94] pre-trained model. Fluorogold masks and c-Fos masks were then overlaid to count the double-positive cells using Fiji[95].

## PDGFRa and Olig2 quantification

Three field of view by three single-layer mosaics were acquired using a laser-scanning microscope (confocal Leica Stellaris 8 Leica Micro-systems CMS GmbH) equipped with ×10 objective. Excitation wave-lengths were sequentially white light laser 554 nm and 653 nm. Emission bandwidths are 559–632 nm for Cy3 signal and 663–839 nm for Alexa Fluor 647 signal. Positive cells were counted in the whole ACC using Fiji[95].

## RNAscope

Forty-eight hours after the last stimulation, brain samples were immersed in isopentane and immediately placed at −80 °C. Frozen samples were embedded in OCT compound and 14-μm thick sections were performed on cryostat, mounted on slides and put back in −80 °C freezer. Sections were fixed, dehydrated and pre-treated using the "RNAscope Sample Preparation and Pre-treatment Guide for Fresh Frozen Tissue using RNAscope Fluorescent Multiplex Assay" protocol (Advanced Cell Diagnostics). Hybridation of *Slc17a7* (ACD, 416631), *Gad2* (ACD, 415071-C2) and *c-fos* (ACD, 316921) probes and develop-ment of the different signals with Opal 520, 590, and 690 fluorophores were performed in accordance with the "RNAscope Multiplex Fluor-escent Reagent Kit v2 Assay" instructions (Advanced Cell Diagnostics). Single-layer images were acquired using a laser-scanning nanozoomer (S60; Hamamatsu Photonics) at ×40 magnification. Quantifications were performed from two sections for each animal on QuPath 0.3.0 software[96]. First, the region of interest was delimited using the polygon annotation tool. Then nuclei were detected within regions of interest using the cell detection module on the Dapi staining. To determine the *c-fos*, *Gad2*, and *Slc17a7* positive cells on our regions of interest, object classifiers were trained in Qupath using Random trees classifiers. We selected all the features by output class (Nucleus mean, Nucleus sum, Nucleus standard deviation, Nucleus maximum, Nucleus minimum, Nucleus range, Cell mean, Cell standard deviation, Cell maximum, Cell minimum, Cytoplasm mean, Cytoplasm standard deviation, Cytoplasm maximum, Cytoplasm minimum), and annotated manually a minimum of 20 points for positive cells and negative cells.

Classifiers were then applied sequentially on the whole region of interest to determine the *c-fos*, *Gad2*, and *Slc17a7* positive cells.

## RNA extraction

Two different batches of animals were generated for RNA sequencing, with a third one for Fluidigm validation of RNA-sequencing results. Forty-eight hours after the last stimulation, bilateral ACC was freshly dissected from animals killed by cervical dislocation and tissues were stored at −80 °C. Total RNA was extracted from ACC tissue with the Qiagen RNeasy Mini Kit (Hilden Germany). Around 20 mg of ACC tissue was disrupted and homogenized with a Kinematica Polytron 1600E in 1.2 ml QIAzol Lysis reagent, for 30 s, and then left at room temperature for 5 min. Next, 240 μl of chloroform was added and mixed before centrifugation for 15 min at $17.000 \times g$ at 4 °C. The aqueous phase (600 μl) was transferred to a new collection tube and mixed with 600 μl of 70% ethanol. The mix was transferred into a RNeasy spin column in a 2-ml collection tube, and centrifuged at $12.000 \times g$ for 15 s. Next, 350 μl of RW1 buffer was added and centrifuged at $12.000 \times g$, for 15 s, before adding 10 μl of DNAse and 70 μl of RDD buffer. The mix was left at room temperature for 15 min and 350 μl of RW1 buffer was added and centrifuged at $12.000 \times g$ for 15 s. The column was then transferred to a new 2 ml collection tube and washed with 500 μl of RPE buffer, before being centrifuged at $12.000 \times g$. Finally, the column was dry centrifuged at $12.000 \times g$ for 5 min, and transferred to a new 1.5 ml collection tube to which 18 μl of RNase-free water was added. Finally, the RNA was eluted by centrifugation for 1 min at $12.000 \times g$. Samples were kept at −80 °C until use.

## Mouse RNA sequencing

RNA sequencing was performed by the Genomeast platform at IGBMC. Full length cDNAs were generated from 5 ng of total RNA using the Clontech SMARTSeq v4 Ultra Low Input RNA kit for Sequencing (PN 091817, Takara Bio Europe, Saint-Germain-en-Laye, France) according to the manufacturer's instructions, with 10 cycles of PCR for cDNA amplification by Seq-Amp polymerase. Six hundred pg of pre-amplified cDNA were then used as input for Tn5 transposon tagmentation by the Nextera XT DNA Library Preparation Kit (PN 15031942, Illumina, San Diego, CA), followed by 12 PCR cycles of library amplification. Following purification with Agencourt AMPure XP beads (BeckmanCoulter, Villepinte, France), the size and concentration of libraries were assessed by capillary electrophoresis. Libraries were then sequenced using an Illumina HiSeq 4000 system using single-end 50 bp reads. Reads were trimmed using cutadapt v1.10, mapped onto the mm10 assembly of the Mus musculus genome, using STAR version 2.5.3a[97]. Gene expression quantification was performed from uniquely aligned reads using htseq-count[98] version 0.6.1p1, with annotations from Ensembl version 95. Read counts were then normalized across samples with the median-of-ratios method proposed by Anders and Huber[99], to make these counts comparable between samples. Principal Component Analysis was computed on regularized logarithm-transformed data calculated with the method proposed by Love and collaborators[100]. Differential expression analysis was performed using R and the Bioconductor package DESeq2 version 1.22.1[100], using RIN values and batches as covariates. Because we generated two batches of mice, the lfcShrink function was used instead of betaPrior in order to calculate *P* values from the log2 Fold changes unshrinked and to perform the shrinkage afterward. RIN and sample batches can be found in Supplementary Table 1.

## Human RNA-sequencing data

Human gene expression data, obtained from our previous publication[37] (archived on GEO Datasets under the reference series: "GSE151827" samples: GSM5026548-97), were generated initially using postmortem ACC tissue from the Douglas-Bell Canada Brain Bank. This cohort was composed of 26 subjects who died by suicide during a major depressive episode, and 24 psychiatrically healthy controls. Groups were matched for age, postmortem interval and brain pH, and included both man (19 in control group and 19 in MDD group) and woman (5 in control group, 7 in MDD group) subjects. All control subjects corresponded to healthy individuals with no psychiatric history. Depressed individuals had a history of early-life adversity (ELA), but did not suffer from comorbid psychosis or bipolar disorder (exclusion criteria). Demographics for the cohort can be found in Supplementary Table 1. While differential expression analysis for the whole cohort (both males and females) was reported previously[37], during the present work we also reprocessed raw gene counts from male individuals only, and conducted a new differential expression analysis to compare men with MDD and healthy control men, taking into account RIN, and age, as in ref. 37. Matching of self-reported gender and biological sex was validated by inspecting relative expression levels of well-known sex-specific genes (e.g., *XIST*, *KDM5D*, *ZFY*, *UTY*, *EIF1AY*).

## Rank-rank hypergeometric overlap (RRHO) analysis

In order to compare mouse and human RNA-Sequencing data, we used the RRHO2 procedure, as described previously[101], using the R package available at https://github.com/Caleb-Huo/RRHO2. Mice-human orthologous genes were first obtained using the R package BioMart, leaving a total of 13572 genes. Genes in each dataset were ranked based on the following metric: −log10(*P* value) x sign(log2 Fold Change). Then, the RRHO2 function was applied to the two gene lists at default parameters (with step size equal to the square root of the list length). The significance of hypergeometric overlaps between human and mouse gene expression changes are reported as log10 *P* values, corrected using the Benjamini–Yekutieli procedure.

## Gene set enrichment analysis (GSEA)

Mouse and human genes were ranked independently based on the fold changes obtained from their respective differential expression analysis. GSEA was performed as previously described[102] using the GSEA-Preranked tool and the Lein Oligodendrocyte markers gene set.

## Weighted gene co-expression network analysis (WGCNA)

WGCNA[40] was used to construct gene networks in mice and humans using RNA-seq expression data and then identify conserved gene modules between the two species. The RNA-sequencing expression data were normalized for batch and RIN in mice; and for age and RIN in humans (as well as sex included when analyzing the whole cohort). First, a soft-threshold power was defined (mouse: 4, human: 8) to reach a degree of independence superior to 0.8 and thus ensure the scale-free topology of the network. To construct the network and detect modules, the blockwiseModules function of the WGCNA algorithm was used, with the minimum size of modules set at 30 genes. Then, the eigengene of each module was correlated with our traits of interest (optogenetic stimulation of the BLA−ACC pathway in mice, or MDD in humans) and gene significance (GS), defined as the correlation between each individual gene and trait, was calculated. Inside each module a measure of the correlation between the module eigengene and the gene expression profile, or module membership (MM), was also assessed. Conservation of WGCNA modules across mice and humans was assessed by Fisher's exact test. Modules were considered as significantly overlapping, and therefore conserved, when *P*adj < 0.05. Among the modules displaying a significant overlap between human and mice, only those with a significant (*P*adj < 0.1) association between the module eigengene and trait, in both species, were kept for further analysis.

## Gene ontology

Enrichments for functional terms in differentially expressed genes (DEGs) in humans and mice were performed using WEBGSTALT for biological process, cellular component, and molecular function[36]. Analysis was restricted to DEG with nominal *P* values<0.05. The same

procedure was applied to the list of genes obtained by RRHO, dysregulated in similar directions in mice and humans, and corresponding to the best one-sided Fisher exact *t* test *P* value.

## Fluidigm

cDNA was generated by subjecting 50 ng of RNA from each sample to reverse transcriptase reaction (Reverse Transcription Master Mix Kit Fluidigm P/N-100-6297). Then, 1.25 μL of each cDNA solution was used to generate a preamp mix containing a pool of the 26 primers pairs and the PreAmp Master Mix Kit (Fluidigm P/N-100-5744). Preamp mixes were run for 14 cycles and the remaining primers were digested with Exonuclease I (New England BIOLAB. P/N M0293l. LOT 0191410). Preamp samples were analyzed for the expression of 22 genes of interest (for primer sequences, see Supplementary Table 3) using the BioMark qPCR platform (Fluidigm, San Francisco, CA, USA). Data were normalized to Gadph, B2m, Actb and Gusb (from the same animal) and fold changes were calculated using the 2-ΔΔCt method[94].

## Plasmid construction

For the expression of shRNAs, long oligonucleotide linkers were designed containing the HindIII and BglII restriction sites at the 5' and 3' extremities, respectively. Each linker contained the loop TTCAA GAGA separating a forward and a reverse copy of the following shRNA sequences: GGAAGAGCCAGACAGGTTTCT for mouse *Sema4a* exon 15, GCCACAACGTCTATATCATGG for eGFP and GCGCTTAGCTGTAGG ATTC for a universal scramble. Linkers were cloned by restriction/ligation downstream the mouse U6 promoter into the pAAV-CMV-mCherry-mU6 construct derived from pAAV-MCS (Agilent). For the construction of the plasmid for the sh efficiency testing in cells, the Sema4a exon 15 was amplified from C57Bl/6 embryonic stem cell genomic DNA and cloned at the C-terminal end of eGFP into pEGFP-C1 by SLIC using the following primers:

GTACAAGTCCGGACTCAGATCTCGAGCTATTAAAGAAGTCCTGA CAGTCCC and GATCAGTTATCTAGATCCGGTGGATCCTTAAGCCACT TCGGCGCC.

## AAV production

Recombinant adenoassociated virus AAV serotype 5 (AAV5) were generated by a triple transfection of HEK293T-derived cell line using polyethylenimine (PEI) transfection reagent and the three following plasmids: pAAV-CMV-mCherry-mU6-shRNA, pXR5 (deposited by Dr Samulski, UNC Vector Core) encoding the AAV serotype 5 capsid and pHelper (Agilent) encoding the adenovirus helper functions. 48 h after transfection, AAV5 vectors were harvested from cell lysate treated with Benzonase (Merck) at 120 U/ml. They were further purified by gradient ultracentrifugation with Iodixanol (OptiprepTM density gradient medium) followed by dialysis and concentration against Dulbecco's PBS using centrifugal filters (Amicon Ultra-15 Centrifugal Filter Devices 100 K, Millipore). Viral titers were quantified by Real-Time PCR using the LightCycler480 SYBR Green I Master (Roche) and primers targeting mCherry sequence. Titers are expressed as genome copy per milliliter (GC/ml).

## Validation of *Sema4a* shRNAs in cell culture

The efficiency of the different shRNA sequences in inhibiting the expression of Sema4a was evaluated in a cell culture test. For this purpose, $5 \times 10^5$ HEK293T cells were seeded in each well of a six-well plate and transfected (6 h after seeding) with 1 mg of each of the pAAV-mCherry-U6-sh-Sema4a plasmids, combined with 1 mg of the peGFP-Sema4a-ex15 plasmid. The peGFP plasmid used in this test was constructed by cloning, into the peGFP-C1 backbone, the Sema4a exon 15 sequence downstream the eGFP gene. The inhibition of the Sema4a mRNA by a specific shRNA induces a decrease in the GFP expression and, therefore, a decrease of the green fluorescence intensity. The fluorescence intensity of the wells transfected with peGFP-Sema4a-

ex15 and pAAV-U6-sh-Sema4a was measured 48 h after transfection and compared with the fluorescence intensity of cells transfected with the same concentration of peGFP-Sema4a-ex15 and pAAV-U6-shScramble (negative control) or pAAV-U6-shGFP (positive control).

## Statistics and reproducibility

Statistical analyses were performed in GraphPad Prism v9.0 software. Data are expressed as mean ± SEM, with statistical significance set as *$P < 0.05$, **$P < 0.01$, ***$P < 0.001$. Student's *t* test (paired and unpaired), One-way analysis of variance (ANOVA), one-way repeated measures ANOVA, and two-way ANOVA followed by Newman–Keuls post hoc test were used when appropriate. If data failed the Shapiro–Wilk normality test, Mann–Whitney non-parametric (one- or two-tailed) analysis was used when comparing two independent groups, and the Chi-Square test for trend was used to compare the distribution of nest scores between groups. Experiments for brain morphological analysis (immunohistochemistry, RNAscope) were repeated independently with similar results in at least three animals using a minimum of three sections per side and per animal. Electrophysiological recordings were performed in at least two mice.

## Emotionality z-scores

Emotionality z-scores were computed as previously described by[34]. First, individual z-score values were calculated for each mouse in each test using Eq. (1):

$$z = \frac{X - \mu}{\sigma} \tag{1}$$

where X represents the individual data for the observed parameter while μ and σ represent the mean and standard deviation of the control group. The directionality of scores was adjusted so that decreased score values reflect increased dimensionality (anxiety- or depressive-like behaviors). For instance, increased immobility time in the FST was converted into negative standard deviation changes compared to group means indicating increased behavioral deficits. Finally, z values obtained for each test were averaged to obtain a single emotionality score using Eq. (2):

$$\text{Emotionality z} - \text{score}: \frac{Z_{test1} + Z_{test2} + \ldots + Z_{testn}}{\text{Number of tests}} \tag{2}$$

## Reporting summary

Further information on research design is available in the Nature Portfolio Reporting Summary linked to this article.

# Data availability

Raw and processed human data reported in this study using postmortem brain tissue from the ACC are publicly available on NCBI's GEO Datasets website, via the Gene Expression Omnibus accession number "GSE151827" (samples: GSM5026548-97). Raw and processed mouse RNA-Sequencing data related to transcriptional signature of neuropathic pain, published by Dai et al. and used in this study for comparison with optogenetic activation of the BLA–ACC pathway (Supplementary Fig. 8) are publicly available via the Gene Expression Omnibus accession number "GSE197233". Finally, raw and processed RNA-sequencing data regarding transcriptional effects, in the ACC, of optogenetic activation of the BLA–ACC pathway, are publicly available via the Gene Expression Omnibus accession number "GSE227159". Source data are provided with this paper.

# Code availability

The source code generated during this study is provided in Supplementary information (Supplementary Note 1).

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

## Acknowledgements

This work was supported by the Centre National de la Recherche Scientifique (contract UPR3212), the University of Strasbourg, the Fondation pour la Recherche Médicale (FRM, FDT202012010622; FDT201805005527) (L.J.B. and M.H.), the Fondation de France (FdF N° Engt:00081244; I.Y., P.E.L., R.B., and E.C.I.), a NARSAD Young Investigator Grant from the Brain & Behavior Research Foundation (24736; I.Y.), the French National Research Agency (ANR) through the Programme d'Investissement d'Avenir EURIDOL graduate school of pain ANR-17- EURE-0022 (L.J.B.), ANR-18-CE37-0004 (I.Y.), ANR-18-CE19-0006-03 (I.Y.) and ANR-19-CE37-0010 (P.E.L.), Hacettepe University Scientific Research Projects Coordination Unit (HUBAB), International Cooperation Project TBI-2018-17569 (B.A.), the Scientific and Technological Research Council of Turkey (TUBITAK) through international post-doctoral research fellowship program (B.A.), IdEx Young Investigator award of University Strasbourg (I.Y.), IdEx postdoctoral fellow of University of Strasbourg (M.T.) and EU Erasmus Mundus Neurotime program (M.K., I.Y., and L.H.). This work received support from the European Union's Horizon 2020 research and innovation program under the Marie Sklodowska-Curie grant agreement N°955684 (M.G., P.E.L., and I.Y.). The authors would also like to acknowledge the CAIUS High-Performance Computing Center of the University of Strasbourg for providing scientific support and access to computing resources. Part of the computing resources was funded by the Equipex Equip@Meso project (Program Investissements d'Avenir) and the CPER Alsacalcul/Big Data. We would like to thank the UMS3415 Chronobiotron for animal care, the In Vitro UAR 3156 imaging platform, Pascale Koebel and Paola Rossolillo from IGBMC for virus preparations, Jennifer Kaufling, Khaled Abdallah, Noémie Willem and Quentin Leboulleux for technical support, Violaine Alunni from IGBMC for the Fluidigm experiment and Daniel Almeida for English editing. Behavioral and microscopy platforms were supported by the Région Grand-Est (Fonds Régional de la Coopération pour la Recherche, CLueDol project). Sequencing was performed by the GenomEast platform, a member of the 'France Génomique' consortium (ANR-10-INBS-0009). We would like to acknowledge that images were adapted from "Mouse brain (sagittal cut)", "Mouse brain (coronal cut)", and "Brain (sagittal cut)" by BioRender.com (2023). Retrieved from https://app.biorender.com/biorender-templates.

## Author contributions

Behavioral experiments: L.J.B., R.W., C.F., S.H.J., M.H., M.T., and I.Y.; molecular experiments: L.J.B., P.E.L., and I.Y.; electrophysiological recordings: S.H.; human data acquisition: G.T. and P.E.L.; imaging: L.J.B., M.K., L.D., M.M., M.N.S., and L.A.H.; immunohistochemistry: R.W., B.A., and M.B.; neuroanatomy: R.W., C.F., and P.V.; experimental design: P.E.L. and I.Y.; data analyses: L.J.B., R.W., P.H., M.G., B.A., P.E.L., and I.Y.; fundings: I.Y., P.E.L., E.C.I., and R.B.; manuscript preparation: L.J.B., R.W., S.H., L.A.H., P.E.L., and I.Y.

## Competing interests

The authors declare no competing interests.
