## [Peer Review File · Nature Communications]

The basolateral amygdala-anterior cingulate pathway contributes to depression-like behaviors and comorbidity with chronic pain behaviors in male miceREVIEWER COMMENTS

Reviewer #1 (Remarks to the Author):

Summary: In this manuscript, the authors explore connections between the BLA in the amygdala and the ACC in a mouse model of chronic pain concomitant with despair-like symptoms. First the authors characterize BLA activation through cFos 8 weeks after nerve injury and find some interesting changes ipsilateral to the injury. The connection between BLA and ACC is corroborated with MRI. Next, the authors utilize an inhibitory opto vector to evaluate the potential of the BLA to ACC projection to modify behavior in the model. The authors bother to actually determine if the Arch is inhibitory in slices, which is appreciated. Inhibition of BLA projections in the ACC did not modify von frey or CPP behavior or LD behavior but did reduce FST and splash test responses (in nerve injured animals). Z score analysis substantiated this effect although an argument could be made based on the original Guilloux paper that the correct analysis here should include LD pref (see below). Next, optogenetic excitation in naïve animals is used to actually induce depressive behaviors across successive weeks. This is an exciting result demonstrating that the pathway here is not just responsible for acute modification of a mouse-specific behavior (which may or may not be “depression”) but leads to more persistent changes with repeated stimulation. There are a couple of issue. Issues here are a lack of emotionality scoring of the LD, ST, NSF, and FST and missing data for FST at the 3 stimulation time point. In addition, the interpretation of this phenotype seems incomplete (see below comment). Next, RNA sequencing was used to evaluate differential gene expression in the ACC of the (male) mice and cohorts from human subjects. This is one of the most powerful parts of this manuscript especially the demonstration of correlations between the induced “depressed” mice and human suicides. The results of these data highlight oligodendrocytes and myelin states in the findings. Although there seem to be some issues between the mouse finding between sequencing/cell counting (ie decreased myelin/oligodendrocytes) and the behavior (quick return to normal 1 week after stimulation paradigm ended), these are still interesting results. Finally, one of the targets identified in the sequencing (Sema4a) was evaluated using a knockdown approach although low n’s for the experimental group reduce the confidence of the data. Overall, the paper uses a nice logical process to go from a functional anatomical connection to a specific gene identified across species to evaluate the connection between pain and major depression. The results have the potential to generate a significant number of new questions to be explored in this important clinical situation. Remaining major issues below reduce enthusiasm but many of these can be addressed.

Major:

- Masking/Blinding – It appears from the methods that only one assay was completed with blinding/masking (Nest test – Line 597). If this is a mistake, please carefully clarify where masking and blinding was used and where it was not used. This is critical for transparency and reproducibility. There are circumstances for behavior assessment where it seems to have been difficult/impossible to do blinded to group. In the optogenetic repeat stimulation experiment, experimental animals are stimulated throughout the splash, NSF, and von Frey test but for controls “the light remained switched off”. It seems difficult to remain blinded if one group has green light running through a fiber optic but the controls did not. For the CPP/RTA, the same issue exists where the light is turned on for the

experimental groups but not for the controls. Although there are no behavioral differences seen in the CPP experiments, this is still a concern.

- **Methods – Sample size justifications.** There are a number of circumstances in which the sample size seemed to change over the course of an experiment. A striking example is in Figure 3. Here, the implication of the text and Fig 3i is that a single mouse when through all stimulations and all behavioral tests across the 4 weeks. The methods line 571-572 partially clarify how many tests mice went through but looking at the n, there are still concerns. For example, ST testing, the n seems to get larger across week 1, 2 and 3 (n=9 then 19 then 28 counting the control mice only). And those n's are significantly bigger than say the NSF or LD tests (n=7 at 9 stimulations). In other words there are two issues here. (1) Minor - The illustration is misleading (there is some clarification in the methods) and (2) Major - there is a serious question related to power here and the potential for type II errors. In other words, if an n=28 was used for all experiments might behavioral effects have been seen earlier in testing. If they were seen earlier in testing then the interpretation that this is not an acute effect would be off. This then impacts the remaining interpretation of the results.
- Another example where the n is concerning is in the final figure showing normalization of the depressive phenotype. The true experimental group here (sema4a knockdown plus stimulation) has an n of 4. Given the very large n's used in the earlier FST experiments with the 9 stimulation paradigm (fig 3k), it is surprising that such a low n would be planned for a such a challenging knockdown experiment. Please justify.
- Please justify focusing on ipsilateral changes (e.g. Fig 1l) for the primary figures of the paper rather than contralateral which would be more typical for a nerve injury model. Although the contralateral data is included in the study, it is odd that it is in the supplemental. Likely the practical reason was the lack of some significant effects on the left contralateral side perhaps due to potential for right amygdala lateralization as you cited. For transparency to readers of the main document, at a minimum, please rephrase Lines 120-122 to indicate that significant changes in FG/cfos was not seen in the contralateral left side (Supp Fig 1e-g). Older data from Goncalves (DOI: 10.1111/j.1460-9568.2012.08235.x) may also be informative in the interpretation of your data. Finally, these differences have even more importance given the random selection of the left or right ACC during cannula implantation (Lines 518-520).
- Please re-do z-score analysis from Figure S2c to include the LD preference test to truly get at “emotionality”. As is, it's kind of unnecessary to do the analysis when the two assays you include are already significant on their own. As argued in Guilloux, the power of the technique is to evaluate overall emotionality when not all tests agree. A similar argument can be made for figure 7h if other tests were completed (if only splash and FST were done then it is fine in figure 7, as is). Note – Z-score method is not in the method section. Please add.
- Given the inclusion of the emotionality z score analysis in figure 2, there is a missed opportunity for this analysis in figure 3. Here, the authors are arguing about a gradual development of a despair phenotype. This is an idea situation to use a z-score to determine if there is an overall change in behavior given that some assays changed and some did not. Likely it is impossible to complete to the large variation in n and different animals completing different tests BUT if it is possible to evaluate a cohort that received all of the despair tests in the same order, it would be interesting to see what a z-score analysis shows.
- Lines 373-381 Discussion – This paragraph discusses the repeat stim effect on depression behavior etc,

but the interpretation feels incomplete. Although the gradual change in behavior is demonstrated from 1 to 2 to 3 weeks in Figure 3 suggesting plasticity, the fact that it immediately reverses at 4 weeks suggests the potential of a cumulative acute effect (analogous to wind-up) especially since it appears that stimulation occurred immediately prior to or during the actual behavior (Methods lines 535-537). The return to baseline at week 4 is particularly curious given the connection in this paradigm to oligodendrocytes and myelin (sequencing data). One might expect such changes to be more long term if they indeed contribute to the depression seen in the repeated stimulation paradigm. This idea is highlighted even more by the reduction in the number of Olig2 positive cells seen after 9 stimulation sessions (line 296).

- Please include sample images of expression in *uscrumble* and *sh-Sema4a* ACC. Ideally this would show knockdown of *Sema4a* (with RNAScope or IHC) but at a minimum targeting with *mCherry* would be good to see.
- Given the differences in *cFos* expression seen along the Anterior to posterior axis (fig S1), please include targeting illustrations for all animals used in these experiments that received stereotaxic injections to the BLA. In other words, overlay Ant to Post illustrations from an atlas with each individual animal for the different experiments indicating where the center of the injection was for each animal (bilaterally). This is fine as extended data.
- Lines 104/105 – Please clarify the details of the well-characterized CPID model that you are referring to here. In other words, there are multiple CPID models that exist. The sentence implies that your cuff model is the model of CPID. While it is a strong model, it is suggested that you switch out the acronym here and throughout for an acronym specific to your model (or just say “cuff model”) rather than use a general acronym to describe the model.

Minor:

- Line 69 – Phrase “In extension to other afferences” is confusing without additional context. Please re-word for clarification. Are you referring to other afferent projections from higher level cortical structures, from the ACC, from the BLA? The part that is particularly confounding is the “in extension” part of the phrase.
- Line 105 – Indicate side of injury in sentence “...peripheral nerve injury on the right side...”
- Figures – Inclusion of individual data points is appreciated but it is suggested that the error bars be placed in front of data points as there are a couple of circumstances in which the error bars are occluded by the data points.
- Figure 3J – Add weeks above the three horizontal columns for easy interpretation.
- Supp Table 2 – Please list sex for mouse cohort too, not just human.
- Extended data figures – In the main text sometimes these figure numbers have an ‘s’ for supplemental and sometimes they do not.
- The authors may consider reviewing and citing a recent meta-analysis Zheng et al (<https://doi.org/10.1038/s41398-022-01949-3>) which shows that pain then depression is most commonly associated with amygdala activation further supporting the translation of the cuff model (e.g. data in Fig 1K).
- Figure 3 – Please include FST data for 3 week stimulation. The 6 and 9 week data are included in 3K but the 3 week data is absent for some reason.

- Methods – Optogenetic stimulation procedure (lines 522-540). This should be separated into two subsections, one for inhibition and one for excitation. As is, the two are confounded as the paragraph implies the 3 day stimulation paradigm was used for both but it seems from the data that no experiments involved 3 days of optogenetic inhibition. Related to excitation, please indicate when behavior was completed after the three days of stimulation during a week. Was it completed on day 4 or immediately after the 3 day of stimulation? Where did stimulation occur (in home cage?)? Finally, please describe and explain the use of green light stimulation right before behavior. Is this just an issue of confounding the excitation and inhibition experiments? Was the light truly continuous (not pulsed)? Why was green light used instead of blue OR was blue light used for the excitation experiments? Was the goal to show that generic light stimulation did not have an effect? Was Did control animals have fiber optics in their cannula even though the light was off? There is no mention in this section for the timing done for the cFos IHC or RNAScope experiments. Please add to section so that the timeline is very clear. Same issue for RNA seq experiments and MRI/DTI experiments. For all of these, when after the 9 stimulations were the tissues collected and/or animals scanned?
- Results Lines 199-207/Extended Fig s3 – In results paragraph, indicate time point for cFos quantification. Was this done at the 4 week time point when the behavior was no longer significant? In Figure S3 legend, please add info for I-L making it clear that this is mRNA from in situ hybridization not IHC.
- Line 229 – May want to rephrase “men” as “biological males” to account for the possibility that some of your human samples may have included biological (XY) males who identified as women or otherwise.
- Please speculate on change in FST behavior in control animals. Earlier in the paper you are seeing immobility of 100+ sec for controls but later (fig 7) the controls in the unscramble group are at 50.
- Methods – HEK-293 knockdown culture validation methods (other than the plasmid) are not described in methods. Please add.
- Stats – Please double check and add all stats used in paper to this section. As an example, chi-square is not included but is used for nest behavior. This might be a good place for the z-score analysis. Consider the use of non-parametric tests for von Frey. Based on the method used, this will not result in truly parametric data (although most people use parametric stats).

Reviewer #2 (Remarks to the Author):

Depression and chronic pain are frequently comorbid. Yet, our understanding of the mechanisms leading to the emotional consequences of chronic pain is poor and limits the design of rationale-based and effective therapeutic strategies. The study by Becker and colleagues aims at disentangling the neuronal circuitries and the molecular/cellular substrates underlying the emergence of depression as a consequence of chronic pain.

First, by using retrograde tracing and neuronal activity markers (cFos), the authors show that basolateral amygdala (BLA) neurons projecting to the anterior cingulate cortex (ACC) are hyperactive in a mouse model of chronic pain-induced depression (CPID). Increased functional connectivity between the ACC

and the amygdala is also detected by fMRI in this CPID model. Then, by acute optogenetic inhibition of BLA inputs to ACC, they show that the activation of the BLA-ACC circuit is required to induce CPID. In a complementary and opposite way, the repeated (3days x consecutive 3weeks) optogenetic activation of this circuit is sufficient to trigger a reversible depressive (but not anxiety) -like phenotype similar to CPID in absence of chronic pain. Mechanistically, such depression-inducing BLA-ACC circuit activation paradigm is accompanied by ACC transcriptomic changes that partly overlap with the transcriptomic signature of ACC in major depression patients. An extensive RNAseq data analysis shows that some of the most conserved and dysregulated gene modules are related to myelination and oligodendrocyte cell biology. qRT-PCR validation confirms the reduction of mature oligodendrocyte genes, and the upregulation of Sema4a, which is an interesting target as formerly shown to be cytotoxic to oligodendrocytes. Finally, the authors show that the in vivo knock-down of Sema4a before BLA-ACC circuit optogenetic activation reduces the emergence of depressive-like behaviors.

The study is well conducted, and the results are important. Yet, in the present form, the mechanistic part of the study is relatively weak, and there are concerns that the authors should address to provide more compelling evidence for their conclusions:

1) In Sema4a knock-down experiment (Figure 7e-h), the sample size of the stimulated sh-Sema4a group is too small (n=4) compared to the standard sample size for behavioral analysis (as also testified by other experiments in this present study). Since this experiment should validate the role of Sema4a in the emergence of depressive behaviors following BLA-ACC circuit activation, the sample size of this experimental group (and perhaps also of the other groups, given the general high variability of the data) should be increased. Moreover, in the forced swim test, UScramble ctrl mice seem to behave differently compared to ctrl mice at the same stage (i.e. most UScramble ctrl mice show about/less than 50s of immobility, while 2/3 of ctrl mice show more than 100s of immobility; compare Fig.7g with Fig. 2k right panel). How this discrepancy can be explained? This is again critical to assess the effect of Sema4a manipulation.

2) The hypothesis that myelin/oligodendroglia loss or dysfunction mediate - or at least accompany - the emergence of depressive behaviors in the BLA-ACC circuit optogenetic activation model is not directly supported by any data, beyond reduced numbers of Olig2+ cells (RNAseq data and reduced fractional anisotropy are not direct evidence). Histological analyses of the density/distribution of mature oligodendrocytes, OPCs and myelin should be done in the ACC of stimulated vs ctrl animals at week 3 and 1 week after the ending of the stimulation paradigm (to assess plastic changes/rescue). Sema4a immunostainings (or protein quantification) in ACC should be also performed to corroborate its upregulation and involvement in the process.

3) In line with this, since Sema4a is proposed as a possible oligodendro-toxic factor following repeated BLA-ACC circuit stimulation, histological analyses of the density/distribution of mature oligodendrocytes, OPCs and myelin should be done also in the ACC of stimulated vs ctrl animals upon Sema4a manipulation.

4) Are Sema4a and oligodendrocyte genes/markers dysregulated also in the ACC of CPID mice? This would strengthen the link with chronic pain.

Minor:

5) In the abstract and in the main text, Sema4a is indicated as “a hub gene”. What do you mean?

6) BLA and ACC abbreviations appear in the abstract without being previously introduced.

7) Line 87: what do you mean with “synergistic”?

8) Line 187: “at the end of week 1, 2 or 3”. Could you specify at which exact day?

9) Line 248: “reveal significant dysregulation of myelination”. Since, these are transcript data, I would tone down this statement in “dysregulation of myelination gene program”.

10) Line 279: “myelination deficiency,..., was indeed modelled by our optogenetic paradigm”. In absence of a more direct evidence (see above), myelination deficiency is only suggested by the presented data.

11) Line 305: “repeated activation of the BLA-ACC pathway disrupts... cell proliferation within the ACC oligodendrocyte lineage”. In the present manuscript, no data are included about OPC proliferation.

12) Line 314: “a potent inhibitor of myelination” and line 343: “inhibitor of myelin function”. Indeed, available literature is more in line with a cytotoxic role of Sema4a on oligodendrocytes (at difference with Lingo, that is a classical inhibitor of myelination). I would change these definitions, as a cytotoxic activity is also more relevant for the proposed role of Sema4a in this study in adult mice (where myelination is already set up).

13) Line 379-380: “requires long-term molecular and neuronal plasticity”. I would replace “neuronal plasticity” with “circuit plasticity”, to also include myelin changes.

14) Figure 1g: Please, include a high magnification inset to better show FG/cFos colocalization.

15) Suppl. Fig3 i-m and text: at which time point has this analysis been performed? In which layers of the ACC?

Reviewer #3 (Remarks to the Author):

In this manuscript Becker and colleagues, tested the role of the BLA-ACC projections in mediating the depressive like behaviors in a model of chronic pain. They first validated their model of pain induced depressive-like behaviors by showing that peripheral nerve injury, through cuff surgeries, induces depressive-like behaviors 7 weeks after surgeries. They showed that this was associated with an

increase in the activity of BLA neurons projecting to the mPFC as measured by c-fos staining combined with retrograde fluorogold beads. The hyperactivity of this pathway was further confirmed with rsfMRI showing elevated connectivity between the ACC and BLA regions 8 weeks after the cuff surgeries. Next, they used optogenetics to test whether light-induced inhibition of this pathway could reverse pain-induced depressive-like behaviors. They first showed that the acute optogenetic inhibition of the BLA-ACC pathway has no effect on pain sensitivity. However, optogenetic inhibition of the BLA-ACC pathway before or during behavioral assessment reversed pain induced behavioral deficits in the splash test (ST) and forced swim test (FST) with no effect on light/dark box (LD) test. Conversely, the optogenetic activation of this pathway gradually triggered the expression of depressive but not anxiety-like behaviors with a stronger effect 3 weeks after the beginning of light stimulations. Importantly, this was associated with an increase in the expression of c-fos in pyramidal cells most importantly but also in GABAergic interneurons in the ACC.

They then used RNAseq to map changes in gene expression and gene networks in the ACC after the repeated optogenetic stimulation of the BLA to ACC pathway for 3 weeks. Interestingly, they took advantage of human data from a cohort of MDD samples for an equivalent brain region to overlap and compare transcriptional signatures in both species. Their results show this optogenetic stimulation protocol is sufficient to induce a large proportion of transcriptional changes equivalent to the signatures observed in humans with MDD. From this analysis, they identified a significant enrichment for genes related to the function of oligodendrocytes in both human MDD and optogenetically stimulated mice. Next, they showed that repeated optogenetic activation (9 session over 3 weeks) of the BLA-ACC pathway reduces the number of olig2 positive cells in the ACC. Impairment in oligodendrocyte function was further supported by MRI analysis showing decreased fractional anisotropy along the BLA-ACC pathway in optogenetically stimulated animals. Finally, they identified Sema4a as one of the top upregulated targets in the optogenetic stimulation paradigm potentially driving these effects in oligodendrocytes. They showed that the KD of Sema4a in the ACC was sufficient to block the expression of depressive-like behaviors induced by 3 weeks of optogenetic stimulations of the BLA to ACC pathway. Overall, this is a very interesting study that combined several approaches to address a set of very complex questions. Results from this paper provide interesting insights into the circuitry involved in mediating the expression of depressive-like behaviors induced by chronic pain and into the molecular mechanisms that may be involved in mediating these effects. Very interestingly, this study also provides an interesting translational perspective on the capacity of this model to reproduce human conditions. Furthermore, the authors also provide very compelling evidence supporting the role of oligodendrocytes in the expression of depressive-like behaviors.

Here are a few comments and suggestions:

1. I noticed that the whole study starts on the premise that pain-induced depressive-like behaviors results from hyperactivity of the BLA-ACC pathway. This is well supported by results shown in Figure 1 and 2, with for instance, c-fos induction in BLA neurons projecting to the ACC and optogenetic inhibition of the BLA-ACC pathway rescuing depressive-like behaviors induced by chronic pain. Then, in Figure 3, repeated optogenetic stimulations over 3 weeks (3 days per week) induce depressive-like behaviors but with no link to chronic pain. The rest of the study is then based on this stimulation paradigm. How do the authors link back this stimulation paradigm to the expression of depressive-like behaviors in the

context of pain if pain is not induced anymore?

I feel there is a missing link here that is maintained until the end of the manuscript. In fact, I feel that the study from Figure 3 to 7 could be related to the role of this pathway in mediating depressive like responses alone but not in the context of pain.

One way to solve this could be to combine this stimulation paradigm with cuff surgeries. Figure 1 shows that the expression of depressive-like behaviors appears globally at week 7 after cuff surgeries. One could assume to see the expression of this phenotype sooner after cuff surgeries if this stimulation paradigm was applied soon after surgeries. Has this been tested?

2. I was surprised to see that the optogenetic stimulation of the BLA-ACC pathway induced depressive-like but not anxiety like behaviors. More precisely, this pathway was previously defined as a main driver of anxiety (Padilla-Coreano et al., 2016, Lowery-Gionta et al., 2018, Felix-Ortiz et al., 2016; Marcus et al., 2020) in male mice. Here, I see that the novelty suppressed feeding (NSF) test was used in Figure 1 but not the other figures. I also see that the LD test was used (Figure 2 but not in other figures) as another measure of anxiety. Did the author consider using other tests such as the elevated plus maze or others to measure more precisely anxiety to confirm these effects on specific behavioral domains?

3. The authors use 3 days of stimulation every week for three weeks to induce the expression of depressive-like behaviors in mice. I would love the authors to explain what this is modelling in human conditions. For instance, is it a gradual but constant increase in the activity of the BLA-ACC pathway? How does this fit in the establishment of the phenotype of pain and depressive-like behaviors in human conditions?

4. The idea of combining mouse and human data in this study is very interesting and adds a very nice translational value to it. Still with the idea of testing pain-induced depressive-like behaviors, I was wondering if any of the samples included in the transcriptional analysis did suffer from chronic pain? If yes, it would be great to have this information in the text and provided in supplemental Table 2. Additionally, it would be great to see the incidence in the human cohort of other important confounding variables such as age, PH and postmortem interval, other environmental and life event variables such as early-life events, and comorbid conditions and medication status and know which one of them were controlled for in the analysis.

5. Figure 5h shows a nice level of correlation between expression values in the RNAseq and qPCR dataset. I assume these genes were selected because they were identified as differentially expressed in the RNAseq data. Did qPCR confirm this differential expression? Or did only the directionality of the effect was similar and highly correlated between techniques?

6. The last series of experiments aims at showing that the optogenetic stimulation paradigm inducing the expression of depressive-like behaviors does so by impairing myelination. This is nicely supported by the MRI fractional anisotropy analysis performed. Then, they show that the KD of Sema4a in the ACC is sufficient to rescue the behavioral effects of this stimulation paradigm. However, this KD affects every pyramidal (CaMKIIa) expressing neurons and not only the ones receiving direct inputs from the BLA.

How do the authors interpret this finding and how can these effects be associated mainly to the cells receiving inputs from the BLA?

7. I noticed that all over the different experiments, distinct behavioral tests are used to validate stress responses. For instance, ST and NSF in Figure 1, LD, ST and FST in Figure 2, LD, NSF, ST, Nesting and FST in Figure 3, ST in Figure 4, ST in Figure 5, Nesting in Figure 6 and ST and FST in Figure 7. For a matter of concordance and consistency, why not always using the same test or battery of test? Also, extended Figure 1H-I in the text refers to functional connectivity result but in fact refer to behavioral data. This should be corrected accordingly.

8. This work was performed in males only. Without asking to do all of this in females, I would like to see a discussion on how this pathway may contribute to driving similar effects in females. I would also be curious to see the authors' interpretation regarding the transcriptional signatures associated with these effects in females.

Reviewer #1 (Remarks to the Author):

Summary: In this manuscript, the authors explore connections between the BLA in the amygdala and the ACC in a mouse model of chronic pain concomitant with despair-like symptoms. First the authors characterize BLA activation through cFos 8 weeks after nerve injury and find some interesting changes ipsilateral to the injury. The connection between BLA and ACC is corroborated with MRI. Next, the authors utilize an inhibitory opto vector to evaluate the potential of the BLA to ACC projection to modify behavior in the model. The authors bother to actually determine if the Arch is inhibitory in slices, which is appreciated. Inhibition of BLA projections in the ACC did not modify von frey or CPP behavior or LD behavior but did reduce FST and splash test responses (in nerve injured animals). Z score analysis substantiated this effect although an argument could be made based on the original Guilloux paper that the correct analysis here should include LD pref (see below). Next, optogenetic excitation in naïve animals is used to actually induce depressive behaviors across successive weeks. This is an exciting result demonstrating that the pathway here is not just responsible for acute modification of a mouse-specific behavior (which may or may not be “depression”) but leads to more persistent changes with repeated stimulation. There are a couple of issues. Issues here are a lack of emotionality scoring of the LD, ST, NSF, and FST and missing data for FST at the 3 stimulation time points. In addition, the interpretation of this phenotype seems incomplete (see below comment). Next, RNA sequencing was used to evaluate differential gene expression in the ACC of the (male) mice and cohorts from human subjects. This is one of the most powerful parts of this manuscript especially the demonstration of correlations between the induced “depressed” mice and human suicides. The results of these data highlight oligodendrocytes and myelin states in the findings. Although there seem to be some issues between the mouse finding between sequencing/cell counting (ie decreased myelin/oligodendrocytes) and the behavior (quick return to normal 1 week after stimulation paradigm ended), these are still interesting results. Finally, one of the targets identified in the sequencing (Sema4a) was evaluated using a knockdown approach although low n’s for the experimental group reduce the confidence of the data. Overall, the paper uses a nice logical process to go from a functional anatomical connection to a specific gene identified across species to evaluate the connection between pain and major depression. The results have the potential to generate a significant number of new questions to be explored in this important clinical situation. Remaining major issues below reduce enthusiasm but many of these can be addressed.

Major:

1) Masking/Blinding – It appears from the methods that only one assay was completed with blinding/masking (Nest test – Line 597). If this is a mistake, please carefully clarify where masking and blinding was used and where it was not used. This is critical for transparency and reproducibility. There are circumstances for behavior assessment where it seems to have been difficult/impossible to do blinded to group. In the optogenetic repeat stimulation experiment, experimental animals are stimulated throughout the splash, NSF, and von Frey test but for controls “the light remained switched off”. It seems difficult to remain blinded if one group has green light running through a fiber optic but the controls did not. For the CPP/RTA, the same issue exists where the light is turned on for the experimental groups but not for the controls. Although there are no behavioral differences seen in the CPP experiments, this is still a concern.

Answer: We thank the reviewer for highlighting this point. Some of the experiments could not be double blinded. This is the case for the RTA, NSF and splash tests in the paradigm involving acute activation of the BLA-ACC pathway (see Extended Data Fig.3b-e), for which the single optogenetic stimulation occurred during behavioral testing. In contrast, during chronic activation of the pathway (Fig.4), optogenetic stimulations were conducted during the first 3 days of the week, followed by behavioral testing on next days, which was therefore fully blinded. In addition, we would like to emphasize that: (i) most tests were performed in the presence of two experimenters in the testing room, for parallel scoring (Splash Test, FST, NSF); (ii) during the characterization of the chronic BLA-ACC activation paradigm, which involved the processing of multiple cohorts (Fig.4), each test was scored across different cohorts by at least 2 experimenters, who obtained consistent results; (iii) for CPP, light was

delivered during conditioning, while no light was administered on the test day, and the time spent in each chamber was measured in an unbiased, automated fashion (infrared beams); (iv) for inhibition experiments in the neuropathic pain-induced depression model (NPID), experimenters were blind to neuropathic pain condition (cuff/sham). These points have now been clarified in the material and methods section (see page 19) and in Fig.2f, 4a. We also added a table detailing experimental designs (Extended Data Table 8).

2) Methods – Sample size justifications. There are a number of circumstances in which the sample size seemed to change over the course of an experiment. A striking example is in Figure 3. Here, the implication of the text and Fig 3i is that a single mouse went through all stimulations and all behavioral tests across the 4 weeks. The methods line 571-572 partially clarify how many tests mice went through but looking at the n, there are still concerns. For example, ST testing, the n seems to get larger across week 1, 2 and 3 (n=9 then 19 then 28 counting the control mice only). And those n's are significantly bigger than say the NSF or LD tests (n=7 at 9 stimulations). In other words there are two issues here. (1) Minor - The illustration is misleading (there is some clarification in the methods) and (2) Major - there is a serious question related to power here and the potential for type II errors. In other words, if an n=28 was used for all experiments might behavioral effects have been seen earlier in testing. If they were seen earlier in testing then the interpretation that this is not an acute effect would be off. This then impacts the remaining interpretation of the results.

Answer: Thirteen different cohorts were used to characterize behavioral impact of repeated optogenetic activation of the BLA-ACC pathway (Fig.4). A table is now included in the revised manuscript to summarize the tests performed in each cohort (see Extended Data Table 8). The distribution of tests across those cohorts was guided by the 3 following principles: First, the splash test was used as an internal control, as it can be repeated several times (unlike tests based on novelty-induced anxiety, such as LD and NSF). This is why its sample size is larger compared to NSF and LD; Second, the FST being the most stressful, it was always performed as a final test; Third, animals never went through more than 3 tests a week.

To address the reviewer's concern on potential type II error, we recently generated a fourteenth mouse cohort to increase sample size for LD and splash tests, and perform FST, at week 1, after 3 stimulations (see Fig.4b). These results confirm the notion that no behavioral deficits can be observed in individual tests (LD, NSF, ST, Nest and FST) at this time-point. In addition, we also computed emotional z-scores by combining LD, NSF, ST, Nest and FST results for each time point (Fig.4c; 100 control, 90 stimulated mice in total; after 3 stimulations; ctrl= 67, stim= 64; after 6 stimulations; ctrl= 40, stim= 35; after 9 stimulations; ctrl= 59, stim= 52). Even this global analysis failed to detect significant effects after 3 or 6 stimulations. This therefore reinforces the conclusion that 9 stimulations over 3 weeks are necessary for the emergence of global emotionality deficits across tests (the FST being the only individual test that already detects a significant effect after 6 stimulations, at 2 weeks).

3) Another example where the n is concerning is in the final figure showing normalization of the depressive phenotype. The true experimental group here (sema4a knockdown plus stimulation) has an n of 4. Given the very large n's used in the earlier FST experiments with the 9 stimulation paradigm (fig 3k), it is surprising that such a low n would be planned for such a challenging knockdown experiment. Please justify.

Answer: These experiments require a series of stereotaxic surgeries: first, bilateral injections are performed in the BLA (ChR2), and in the ACC (sh-Sema4a), followed 2 weeks later by implantation of a cannula in the ACC. Animals who lost their cannula during the course of

these manipulations, or had misplaced injections in the BLA or ACC (as assessed for all mice after final behavioral testing), were excluded from final analyses. Therefore, of the 39 mice initially used, only 27 were kept. We nevertheless agree with the reviewer's concern and, to address it, we recently generated 2 additional cohorts. Final results, presented in Fig.8 (initially Fig.7), now correspond to a total of 62 mice (N=11-18 mice/group). Importantly, they confirm our initial findings obtained with a smaller sample size, whereby Sema4a knock-down prevents behavioral deficits induced by repeated optogenetic stimulation of the BLA-ACC pathway.

4) Please justify focusing on ipsilateral changes (e.g. Fig 1I) for the primary figures of the paper rather than contralateral which would be more typical for a nerve injury model. Although the contralateral data is included in the study, it is odd that it is in the supplemental. Likely the practical reason was the lack of some significant effects on the left contralateral side perhaps due to potential for right amygdala lateralization as you cited. For transparency to readers of the main document, at a minimum, please rephrase Lines 120-122 to indicate that significant changes in FG/cfos was not seen in the contralateral left side (Supp Fig 1e-g). Older data from Goncalves (DOI: 10.1111/j.1460-9568.2012.08235.x) may also be informative in the interpretation of your data. Finally, these differences have even more importance given the random selection of the left or right ACC during cannula implantation (Lines 518-520).

Answer: We thank the reviewer for pointing this out. We have now included contralateral data in the main text (see Fig.1), and rephrased the results section accordingly (see page 4-5). We also cited the work of Goncalves et al., 2012, as suggested.

5) Please re-do z-score analysis from Figure S2c to include the LD preference test to truly get at "emotionality". As is, it's kind of unnecessary to do the analysis when the two assays you include are already significant on their own. As argued in Guilloux, the power of the technique is to evaluate overall emotionality when not all tests agree. A similar argument can be made for figure 7h if other tests were completed (if only splash and FST were done then it is fine in figure 7, as is). Note – Z-score method is not in the method section.

Answer: We recomputed z-scores to include results from the LD test, as suggested (see **Extended Data Fig.2**). This does not affect our conclusions, as the significant impact of neuropathic pain on emotionality score remained significant, and fully blocked by optogenetic inhibition of the BLA-ACC pathway. The main text has been modified to reflect this new analysis (see page 6). The reviewer is correct that only FST and Splash tests were performed in Fig.7 after Sema4a knock-down (corresponding to Fig.8 in the revised manuscript), since chronic BLA-ACC stimulation had no effect in LD and NSF tests (see Fig.4). Finally, how z-scores were computed is now indicated in the methods section (pages 31-32).

6) Given the inclusion of the emotionality z score analysis in figure 2, there is a missed opportunity for this analysis in figure 3. Here, the authors are arguing about a gradual development of a despair phenotype. This is an ideal situation to use a z-score to determine if there is an overall change in behavior given that some assays changed and some did not. Likely it is impossible to complete to the large variation in n and different animals completing different tests BUT if it is possible to evaluate a cohort that received all of the despair tests in the same order, it would be interesting to see what a z-score analysis shows.

Answer: We agree that inclusion of the z-score would strengthen results in Fig.3 (Fig.4 in revised manuscript). However, as pointed out by the reviewer, computing the z-score following the method initially proposed by Guilloux and colleagues, 2011 was not possible, since all behavioral tests were not systematically performed in each and every animal cohort (see above, and Extended Data Table 8). Instead, as an alternative, we computed z-scores for each test, within each cohort, and then averaged those scores across tests performed at each time-point in different cohorts (3, 6 and 9 stimulations). Results are included in Fig.4c, and the main text has been modified accordingly (page 7). Considering the high sample size achieved across our large number of cohorts (see revised Fig.4c), we believe these results convincingly demonstrate that, overall, enhanced emotionality did not emerge after 3 or 6 stimulations, and only became significant at the 3-week time-point.

7) Lines 373-381 Discussion – This paragraph discusses the repeat stim effect on depression behavior etc, but the interpretation feels incomplete. Although the gradual change in behavior is demonstrated from 1 to 2 to 3 weeks in Figure 3 suggesting plasticity, the fact that it immediately reverses at 4 weeks suggests the potential of a cumulative acute effect (analogous to wind-up) especially since it appears that stimulation occurred immediately prior to or during the actual behavior (Methods lines 535-537). The return to baseline at week 4 is particularly curious given the connection in this paradigm to oligodendrocytes and myelin (sequencing data). One might expect such changes to be more long term if they indeed contribute to the depression seen in the repeated stimulation paradigm. This idea is highlighted even more by the reduction in the number of Olig2 positive cells seen after 9 stimulation sessions line 296).

Answer: We recognize that our initial description was unclear. In our chronic BLA-ACC activation paradigm, optogenetic stimulations did not occur immediately prior or during behavioral testing, but instead were applied during the first 3 days of each week, followed, at least 24 hours later, by behavioral testing. We agree with the reviewer that, considering the proposed myelin-related mechanism, the kinetics we observed might seem surprising at first glance. Prompted by this comment, we have now conducted additional experiments to analyze the number of Olig2 positive cells 1 week after the end of optogenetic stimulations, when their behavioral impact has waned. Interestingly, these new results indicate that this delay is sufficient for the number of Olig2-positive cells to return to control levels (see Extended Data Fig.7). This kinetic therefore appears to support our behavioral results, and argues for a model with fast, adaptive changes in the oligodendrocyte lineage. We also note that similarly rapid changes have been observed by others in different experimental contexts. For example, McKenzie reported that motor learning (running on a complex wheel) was sufficient to trigger, within 4 to 11 days, significant increases in the number of oligodendrocyte progenitors and myelin-forming oligodendrocytes (McKenzie et al, 2014).

8) Please include sample images of expression in uScramble and sh-Sema4a ACC. Ideally this would show knockdown of Sema4a (with RNAScope or IHC) but at a minimum targeting with mCherry would be good to see.

Answer: As suggested by the reviewer, images illustrating the expression of the mCherry reporter following injections of uScramble and sh-Sema4a viral vectors in the ACC have now been included (see Fig.8f in revised manuscript).

9) Given the differences in cFos expression seen along the Anterior to posterior axis (fig S1), please include targeting illustrations for all animals used in these experiments that received stereotaxic injections to the BLA. In other words, overlay Ant to Post

illustrations from an atlas with each individual animal for the different experiments indicating where the center of the injection was for each animal (bilaterally). This is fine as extended data.

Answer: As suggested by the reviewer, injection sites have now been included in Extended Data Fig. 2, 3, and 9.

10) Lines 104/105 – Please clarify the details of the well-characterized CPID model that you are referring to here. In other words, there are multiple CPID models that exist. The sentence implies that your cuff model is the model of CPID. While it is a strong model, it is suggested that you switch out the acronym here and throughout for an acronym specific to your model (or just say “cuff model”) rather than use a general acronym to describe the model.

Answer: To avoid any misunderstanding, we have replaced CPID by neuropathic pain-induced depression (NPID) throughout the whole manuscript.

Minor:

11) Line 69 – Phrase “In extension to other afferences” is confusing without additional context. Please re-word for clarification. Are you referring to other afferent projections from higher level cortical structures, from the ACC, from the BLA? The part that is particularly confounding is the “in extension” part of the phrase.

Answer: This has been rephrased (page 3).

12) Line 105 – Indicate side of injury in sentence “...peripheral nerve injury on the right side...”

Answer: The side of lesion has been added (page 4).

13) Figures – Inclusion of individual data points is appreciated but it is suggested that the error bars be placed in front of data points as there are a couple of circumstances in which the error bars are occluded by the data points.

Answer: This has been done.

14)• Figure 3J – Add weeks above the three horizontal columns for easy interpretation.

Answer: This has been done (see Fig.4 in revised manuscript).

15) Supp Table 2 – Please list sex for mouse cohort too, not just human.

Answer: This has been done.

16) Extended data figures – In the main text sometimes these figure numbers have an ‘s’ for supplemental and sometimes they do not.

Answer: We made the necessary changes throughout.

17) The authors may consider reviewing and citing a recent meta-analysis Zheng et al (<https://doi.org/10.1038/s41398-022-01949-3>) which shows that pain then depression is most commonly associated with amygdala activation further supporting the translation of the cuff model (e.g. data in Fig 1K).

Answer: We thank the reviewer for pointing out this interesting meta-analysis, which is now cited in the result section (page 5).

18) Figure 3 – Please include FST data for 3 week stimulation. The 6 and 9 week data are included in 3K but the 3 week data is absent for some reason.

Answer: As suggested, a fourteenth mouse cohort was generated to conduct the FST after 3 stimulations (1-week time-point) and to increase the number of animals for the LD and NSF tests. These new data appear in revised Fig.4.

19) Methods – Optogenetic stimulation procedure (lines 522-540). This should be separated into two sub-sections, one for inhibition and one for excitation. As is, the two are confounded as the paragraph implies the 3 day stimulation paradigm was used for both but it seems from the data that no experiments involved 3 days of optogenetic inhibition. Related to excitation, please indicate when behavior was completed after the three days of stimulation during a week. Was it completed on day 4 or immediately after the 3 day of stimulation? Where did stimulation occur (in home cage?)? Finally, please describe and explain the use of green light stimulation right before behavior. Is this just an issue of confounding the excitation and inhibition experiments? Was the light truly continuous (not pulsed)? Why was green light used instead of blue OR was blue light used for the excitation experiments? Was the goal to show that generic light stimulation did not have an effect? Did control animals have fiber optics in their cannula even though the light was off? There is no mention in this section for the timing done for the cFos IHC or RNAScope experiments. Please add to section so that the timeline is very clear. Same issue for RNA seq experiments and MRI/DTI experiments. For all of these, when after the 9 stimulations were the tissues collected and/or animals scanned?

Answer: We have revised the methods section following the reviewer's suggestion, with 2 separate paragraphs for optogenetic activation and inhibition experiments (see page 18). Figures illustrating experimental designs have also been revised for clarity. During activation experiments, behavioral testing was performed 1 to 4 days after the last stimulation. As a systematic rule, only one test was performed each day, with a maximum of 3 tests a week. Animals were stimulated in their home-cages. For inhibition experiments, the stimulation was applied before LD or FST testing (since these paradigms were not compatible with tethered animals), and during Splash or von Frey testing. Pulsed blue light was used for excitation (control animals being also implanted with a cannula), while green light was applied for neuronal inhibition, in a continuous fashion. We apologize for the confusion and revised the methods section to clarify these points (see page 18).

For RNAseq, RNAScope, Fluidigm as well as DTI experiments, tissues were collected or animals were scanned 48 hours after the 9th optogenetic stimulation, matching the time-point when behavioral deficits were observed. For the c-Fos immunoperoxidase experiment, the goal was merely to determine if the stimulation of incoming BLA fibers was sufficient to induce neuronal activation in the ACC. To do so, animals were stimulated a 10th time, 5 days

after the 9th stimulation, and perfused 90 minutes later (with no stimulation in control mice). We now included all these information in each corresponding methods section.

20) Results Lines 199-207/Extended Fig s3 – In results paragraph, indicate time point for cFos quantification. Was this done at the 4 week time point when the behavior was no longer significant? In Figure S3 legend, please add info for I-L making it clear that this is mRNA from in situ hybridization not IHC.

Answer: We quantified c-Fos at 2 time-points. First, in Extended Data Fig. 3f, the goal was to control, in vivo, for the activation of ACC neurons following stimulation of afferents BLA fibers. This was done in some of the behavioral cohorts initially used for characterizing the kinetics defined in Fig.4. After completion of the 9 sessions of stimulation, followed by behavioral testing, mice were stimulated a 10th time (at the 4-week time-point) and perfused 90 minutes later. Second, to detect a latent increase in ACC neurons activation following BLA-ACC repeated activations (corresponding to the moment when behavioral testing is conducted), c-Fos was also quantified 48 hours after the 9th stimulation, using RNAscope. Those results are presented in Fig.4d-e and in Extended Data Fig.4a-e. This is now clarified in the revised manuscript (see page 7) and in Figure legends.

21) Line 229 – May want to rephrase “men” as “biological males” to account for the possibility that some of your human samples may have included biological (XY) males who identified as women or otherwise.

Answer: We had indeed investigated that possibility. An analysis of a few genes showing well-known sex-specific expression patterns was conducted, and confirmed that biological sex and self-identified gender were concordant for all samples:

This quality control is now mentioned in the material and methods.

22) Please speculate on change in FST behavior in control animals. Earlier in the paper you are seeing immobility of 100+ sec for controls but later (fig 7) the controls in the unscramble group are at 50.

Answer:

This variability is also present in Fig.2 and Fig.4, but less apparent since a higher number of animal cohorts was used. Indeed, cohorts in the latter experiments (4 batches in Fig.2, and 2 to 3 batches per time point in Fig.4) display average immobility time ranging from 50 to 150s, corresponding to an average of 100s. In our original submission, only 1 cohort was used for Sema4a knockdown, with an averaged immobility around 50s. When 2 additional Sema4a knockdown cohorts were generated for the present revision, this overall baseline difference disappeared (average immobility time for the 3 batches = 103s). Overall, we believe these baseline differences reflect batch variability, which we and others have routinely observed along the years. Behavioral data combining all 3 cohorts are now presented as % of controls (non-stimulated scrambled vector; see Fig. 8 in revised manuscript).

23) Methods – HEK-293 knockdown culture validation methods (other than the plasmid) are not described in methods. Please add.

Answer: Detailed information has been added concerning the HEK-293 knockdown (see page 31).

24) Stats – Please double check and add all stats used in paper to this section. As an example, chi-square is not included but is used for nest behavior. This might be a good place for the z-score analysis. Consider the use of non-parametric tests for von Frey. Based on the method used, this will not result in truly parametric data (although most people use parametric stats).

Answer: The statistics section has been revised as suggested (see page 31).

Reviewer #2 (Remarks to the Author):

Depression and chronic pain are frequently comorbid. Yet, our understanding of the mechanisms leading to the emotional consequences of chronic pain is poor and limits the design of rationale-based and effective therapeutic strategies. The study by Becker and colleagues aims at disentangling the neuronal circuitries and the molecular/cellular substrates underlying the emergence of depression as a consequence of chronic pain. First, by using retrograde tracing and neuronal activity markers (cFos), the authors show that basolateral amygdala (BLA) neurons projecting to the anterior cingulate cortex (ACC) are hyperactive in a mouse model of chronic pain-induced depression (CPID). Increased functional connectivity between the ACC and the amygdala is also detected by fMRI in this CPID model. Then, by acute optogenetic inhibition of BLA inputs to ACC, they show that the activation of the BLA-ACC circuit is required to induce CPID. In a complementary and opposite way, the repeated (3days x consecutive 3weeks) optogenetic activation of this circuit is sufficient to trigger a reversible depressive (but not anxiety) -like phenotype similar to CPID in absence of chronic pain. Mechanistically, such depression-inducing BLA-ACC circuit activation paradigm is accompanied by ACC transcriptomic changes that partly overlap with the transcriptomic signature of ACC in major depression patients. An extensive RNAseq data analysis shows that some of the most conserved and dysregulated gene modules are related to myelination and oligodendrocyte cell biology. qRT-PCR validation confirms the reduction of mature oligodendrocyte genes, and the upregulation of Sema4a, which is an interesting target as formerly shown to be cytotoxic to oligodendrocytes. Finally, the authors show that the in vivo knock-down of Sema4a before BLA-ACC circuit optogenetic activation reduces the emergence of depressive-like behaviors.

The study is well conducted, and the results are important. Yet, in the present form, the mechanistic part of the study is relatively weak, and there are concerns that the authors should address to provide more compelling evidence for their conclusions:

1) In Sema4a knock-down experiment (Figure 7e-h), the sample size of the stimulated sh-Sema4a group is too small (n=4) compared to the standard sample size for behavioral analysis (as also testified by other experiments in this present study). Since this experiment should validate the role of Sema4a in the emergence of depressive behaviors following BLA-ACC circuit activation, the sample size of this experimental group (and perhaps also of the other groups, given the general high variability of the data) should be increased. Moreover, in the forced swim test, UScramble ctrl mice seem to behave differently compared to ctrl mice at the same stage (i.e. most UScramble ctrl mice show about/less than 50s of immobility, while 2/3 of ctrl mice show more than 100s of immobility; compare Fig.7g with Fig. 2k right panel). How this discrepancy can be explained? This is again critical to assess the effect of Sema4a manipulation.

Answer: Based on a similar comment from Reviewer 1, we now added 2 animal cohorts to increase sample size in each group. Results confirmed that the deletion of Sema4a prevents the development of depressive-like behaviors induced by chronic activation of the BLA-ACC pathway. Concerning the FST, as mentioned in response to a similar comment from Reviewer 1 (comment #22), baseline differences likely reflected batch variability. In our original submission, only one cohort was used for the Sema4a knockdown experiment, with an averaged immobility around 50s. When 2 additional Sema4a knockdown cohorts were generated for the present revised manuscript, this overall baseline difference disappeared (average immobility time for the 3 batches = 103s). Behavioral data combining all 3 cohorts are now presented as % of controls (non-stimulated scrambled vector; Fig. 8).

2) The hypothesis that myelin/oligodendroglia loss or dysfunction mediate - or at least accompany - the emergence of depressive behaviors in the BLA-ACC circuit optogenetic activation model is not directly supported by any data, beyond reduced

numbers of Olig2+ cells (RNAseq data and reduced fractional anisotropy are not direct evidence). Histological analyses of the density/distribution of mature oligodendrocytes, OPCs and myelin should be done in the ACC of stimulated vs ctrl animals at week 3 and 1 week after the ending of the stimulation paradigm (to assess plastic changes/rescue). Sema4a immunostainings (or protein quantification) in ACC should be also performed to corroborate its upregulation and involvement in the process.

Answer: To address the reviewer's comment, we included 2 sets of additional experiments. First, we performed double immunostaining against Olig2 and Platelet-derived growth factor receptor alpha (PDGFRa), to quantify immature oligodendrocyte progenitor cells (OPCs) at the 3-week time-point (9 stimulations). Results showed no alterations in the number of PDGFRa+ cells. Second, we evaluated the number of Olig2+ cells 1 week after the 9th stimulation, when behavioral effects of BLA-ACC stimulations are no longer detectable. No difference was found between control and stimulated groups. Overall, these results reveal a histological kinetic that appears consistent with, and reinforces, our behavioral results (see Fig.7/Extended Data Fig.7). Accordingly, repeated BLA-ACC activation induced a significant decrease in the number of mature oligodendrocytes in the ACC, in the absence of any change in OPC proliferation, suggesting decreased survival. Strikingly, this effect was no longer significant 1 week later, indicating rapid recovery upon cessation of optogenetic stimulations. These new results are now discussed in the revised manuscript (see Discussion, page 14).

3) In line with this, since Sema4a is proposed as a possible oligodendro-toxic factor following repeated BLA-ACC circuit stimulation, histological analyses of the density/distribution of mature oligodendrocytes, OPCs and myelin should be done also in the ACC of stimulated vs ctrl animals upon Sema4a manipulation.

Answer: The goal of our study was to better understand neuronal and molecular substrates that may mediate the interplay between chronic pain and depression. While our results provide new knowledge documenting the important role of the BLA-ACC pathway in mood regulation, and causally implicate Sema4a in these processes, we agree with the reviewer that they also raise the questions of the molecular mechanisms mediating the impact of Sema4a on oligodendrocytes and emotional responses. Compared to previous work that characterized a role for this gene in neurological disorders, including multiple sclerosis, we believe our findings unravel a new facet of Sema4a functions. Although beyond the scope of the present manuscript, studies to explore these new avenues are currently initiated by our group. To acknowledge this limitation, and prospects for further work, we now added at the end of the discussion:

“Therefore, we document a new role for Sema4a, whereby changes in its expression may play a pivotal role in mood regulation, and represent a putative therapeutic target. Among others, avenues for future work will include characterizing the mechanisms by which Sema4a modulates oligodendrocytes, identifying in which ACC cell-types such processes are recruited by BLA afferences, and describing putative sex differences in these aspects of pain and emotional processing.”

4) Are Sema4a and oligodendrocyte genes/markers dysregulated also in the ACC of CPID mice? This would strengthen the link with chronic pain.

Answer: We thank the reviewer for this important suggestion, which we addressed using data recently published by Dai and collaborators, 2022. The authors used the Spared Nerve Injury (SNI) model to induce neuropathic pain, and characterized gene expression changes occurring 2 weeks later (when pain-induced depressive-like behaviors were observed in their

model), in the ACC and another cortical region, the medial prefrontal cortex. Raw data, available from NCBI's Gene Expression Omnibus (GSE197233), were reprocessed using our in-house pipeline and directly compared to those triggered in our paradigm by repeated optogenetic BLA-ACC activations, using RRHO2 (Rank-rank hypergeometric overlay; see Methods). Results revealed a striking pattern of genome-wide convergence. Importantly, this included strongly significant overlaps among genes that were commonly downregulated (both by SNI and after optogenetic BLA-ACC activations) and significantly enriched for myelin-related Gene Ontology terms. These results, now presented in Extended Data 8 of the revised manuscript, therefore suggest that, during chronic pain, impaired transcriptional activity of the myelination program may also contribute to mood dysregulation. Future work will be necessary to characterize the role of *Sema4a* in this particular context.

Minor:

5) In the abstract and in the main text, *Sema4a* is indicated as “a hub gene”. What do you mean?

Answer: In the context of our gene network analyses, we agree that the term “hub” was misleading, as it was primarily used to describe the fact that *Sema4a* was significantly upregulated in both mice and men in the context of altered mood. For clarity, this term is no longer used in the revised manuscript.

6) BLA and ACC abbreviations appear in the abstract without being previously introduced.

Answer: We thank the reviewer for pointing this out. The abstract has been revised accordingly.

7) Line 87: what do you mean with “synergistic”?

Answer: To tone down this phrasing, the word “synergistic” has been replaced by “coherent”.

8) Line 187: “at the end of week 1, 2 or 3”. Could you specify at which exact day?

Answer: The experimental design is now detailed in the method section, as well as in Extended Data Table 8. All behavioral tests were performed between 1 to 4 days after the last stimulation, with the exact day depending on the number and type of tests used. Animals never went through more than 3 tests a week, and we always started with anxiety tests (LD/NSF). When three tests were conducted during the same week, they were usually performed on the 1st, 2nd and 3rd day after stimulation. When NSF was used, other tests were performed on the 2nd, 3rd and 4th day after the last stimulation, because the food restriction required for this test was started on the first day following the last stimulation.

9) Line 248: “reveal significant dysregulation of myelination”. Since, these are transcript data, I would tone down this statement in “dysregulation of myelination gene program”.

Answer: As suggested by the reviewer, this sentence has been rephrased (page 9).

10) Line 279: “myelination deficiency,..., was indeed modelled by our optogenetic

paradigm". In absence of a more direct evidence (see above), myelination deficiency is only suggested by the presented data.

Answer: This has been rephrased as suggested (page 10).

11) Line 305: "repeated activation of the BLA-ACC pathway disrupts... cell proliferation within the ACC oligodendrocyte lineage". In the present manuscript, no data are included about OPC proliferation.

Answer: Since our new results indicate that alterations observed following BLA-ACC activations selectively affected mature oligodendrocytes (decreased number of Olig2+ cells, with no change in the number of immature OPCs), this sentence has been rephrased as follows (see pages 10-11):

"Since myelination is an important determinant of the structural connectivity assessed by DTI^{43,44}, these results reinforce the notion that repeated activation of the BLA-ACC pathway disrupts the transcriptional program and the survival of mature oligodendrocyte within the ACC, leading to altered connectivity between the 2 structures."

12) Line 314: "a potent inhibitor of myelination" and line 343: "inhibitor of myelin function". Indeed, available literature is more in line with a cytotoxic role of Sema4a on oligodendrocytes (at difference with Lingo, that is a classical inhibitor of myelination). I would change these definitions, as a cytotoxic activity is also more relevant for the proposed role of Sema4a in this study in adult mice (where myelination is already set up).

Answer: As suggested by the reviewer, this has been rephrased (page 11).

13) Line 379-380: "requires long-term molecular and neuronal plasticity". I would replace "neuronal plasticity" with "circuit plasticity", to also include myelin changes.

Answer: We now wrote "circuit plasticity", as suggested (page 13).

14) Figure 1g: Please, include a high magnification inset to better show FG/cFos colocalization.

Answer: A high magnification inset of FG/cFos colocalization now appears in Fig.1n.

15) Suppl. Fig3 i-m and text: at which time point has this analysis been performed? In which layers of the ACC?

Answer: This experiment was performed 48 hours after the last stimulation. Since previous preliminary analyses focusing specifically on layer II/III and V had not provided evidence for significant differences, here we opted to analyze the whole ACC (+0.5 to +1.2 mm in AP).

Reviewer #3 (Remarks to the Author):

In this manuscript Becker and colleagues tested the role of the BLA-ACC projections in mediating the depressive-like behaviors in a model of chronic pain. They first validated their model of pain induced depressive-like behaviors by showing that peripheral nerve injury, through cuff surgeries, induces depressive-like behaviors 7 weeks after surgeries. They showed that this was associated with an increase in the activity of BLA neurons projecting to the mPFC as measured by c-fos staining combined with retrograde fluorogold beads. The hyperactivity of this pathway was further confirmed with rsfMRI showing elevated connectivity between the ACC and BLA regions 8 weeks after the cuff surgeries. Next, they used optogenetics to test whether light-induced inhibition of this pathway could reverse pain-induced depressive-like behaviors. They first showed that the acute optogenetic inhibition of the BLA-ACC pathway has no effect on pain sensitivity. However, optogenetic inhibition of the BLA-ACC pathway before or during behavioral assessment reversed pain induced behavioral deficits in the splash test (ST) and forced swim test (FST) with no effect on light/dark box (LD) test. Conversely, the optogenetic activation of this pathway gradually triggered the expression of depressive but not anxiety-like behaviors with a stronger effect 3 weeks after the beginning of light stimulations. Importantly, this was associated with an increase in the expression of c-fos in pyramidal cells most importantly but also in GABAergic interneurons in the ACC.

They then used RNAseq to map changes in gene expression and gene networks in the ACC after the repeated optogenetic stimulation of the BLA to ACC pathway for 3 weeks. Interestingly, they took advantage of human data from a cohort of MDD samples for an equivalent brain region to overlap and compare transcriptional signatures in both species. Their results show this optogenetic stimulation protocol is sufficient to induce a large proportion of transcriptional changes equivalent to the signatures observed in humans with MDD. From this analysis, they identified a significant enrichment for genes related to the function of oligodendrocytes in both human MDD and optogenetically stimulated mice. Next, they showed that repeated optogenetic activation (9 sessions over 3 weeks) of the BLA-ACC pathway reduces the number of olig2 positive cells in the ACC. Impairment in oligodendrocyte function was further supported by MRI analysis showing decreased fractional anisotropy along the BLA-ACC pathway in optogenetically stimulated animals. Finally, they identified Sema4a as one of the top upregulated targets in the optogenetic stimulation paradigm potentially driving these effects in oligodendrocytes. They showed that the KD of Sema4a in the ACC was sufficient to block the expression of depressive-like behaviors induced by 3 weeks of optogenetic stimulations of the BLA to ACC pathway. Overall, this is a very interesting study that combined several approaches to address a set of very complex questions. Results from this paper provide interesting insights into the circuitry involved in mediating the expression of depressive-like behaviors induced by chronic pain and into the molecular mechanisms that may be involved in mediating these effects. Very interestingly, this study also provides an interesting translational perspective on the capacity of this model to reproduce human conditions. Furthermore, the authors also provide very compelling evidence supporting the role of oligodendrocytes in the expression of depressive-like behaviors.

Here are a few comments and suggestions:

1. I noticed that the whole study starts on the premise that pain-induced depressive-like behaviors results from hyperactivity of the BLA-ACC pathway. This is well supported by results shown in Figure 1 and 2, with for instance, c-fos induction in BLA neurons projecting to the ACC and optogenetic inhibition of the BLA-ACC pathway rescuing depressive-like behaviors induced by chronic pain. Then, in Figure 3, repeated optogenetic stimulations over 3 weeks (3 days per week) induce depressive-like behaviors but with no link to chronic pain. The rest of the study is then based on this stimulation paradigm. How do the authors link back this stimulation

paradigm to the expression of depressive-like behaviors in the context of pain if pain is not induced anymore?

I feel there is a missing link here that is maintained until the end of the manuscript. In fact, I feel that the study from Figure 3 to 7 could be related to the role of this pathway in mediating depressive-like responses alone but not in the context of pain. One way to solve this could be to combine this stimulation paradigm with cuff surgeries. Figure 1 shows that the expression of depressive-like behaviors appears globally at week 7 after cuff surgeries. One could assume to see the expression of this phenotype sooner after cuff surgeries if this stimulation paradigm was applied soon after surgeries. Has this been tested?

Answer: We thank the reviewer for the positive assessment of the manuscript, and for the thoughtful question. We believe our data support the hypothesis of an involvement of the BLA-ACC pathway in both naive and chronic pain conditions, as its optogenetic inhibition was sufficient to block pain-induced emotional dysfunction (Fig.1-2) while, conversely, its stimulation was sufficient to trigger anxiodepressive-like responses in naive mice (Fig.4). We nevertheless agree that, as mentioned also by Reviewer 2, our initial submission fell short of demonstrating whether molecular mechanisms recruited in our optogenetic paradigm (Fig.4-7) might be also relevant for the understanding of chronic pain. To address this possibility, we have now reanalyzed transcriptomic data recently published by Dai and collaborators, 2022 in a chronic neuropathic pain model, and compared them to changes observed in our optogenetic paradigm (using RRHO2). Results uncover striking similarity across the 2 datasets, including the downregulation of myelin- and oligodendrocyte-related genes. Therefore, it is possible to hypothesize that the latter changes may also contribute to mood dysregulation in the chronic pain context, which will have to be tested in future studies (see also our Response to comment #4 from Reviewer 2).

2. I was surprised to see that the optogenetic stimulation of the BLA-ACC pathway induced depressive-like but not anxiety like behaviors. More precisely, this pathway was previously defined as a main driver of anxiety (Padilla-Coreano et al., 2016, Lowery-Gionta et al., 2018, Felix-Ortiz et al., 2016; Marcus et al., 2020) in male mice. Here, I see that the novelty suppressed feeding (NSF) test was used in Figure 1 but not the other figures. I also see that the LD test was used (Figure 2 but not in other figures) as another measure of anxiety. Did the author consider using other tests such as the elevated plus maze or others to measure more precisely anxiety to confirm these effects on specific behavioral domains?

Answer: We thank the reviewer for this comment. In contrast with these studies from Padilla-Coreano et al (2016), Lowery-Gionta et al (2018), Felix-Ortiz et al (2016) and Marcus et al (2020), which all focused on different subparts of the PFC (see table below), our own study encompassed other cortical areas, areas 24a/24b (which we, and others, refer to as the anterior cingulate cortex, ACC; see Fillinger et al., 2017, 2018, Vogt and Paxinos, 2014). To the best of our knowledge, this is the first study that specifically determined the role of the BLA-ACC pathway in the regulation of anxiety-like behaviors. By complementing aforementioned BLA-PFC studies, these results indicate that separate pathways that originate in the BLA, but show divergent cortical projections, may differentially contribute to the control of anxiety-like responses.

Paper	Coordinates		
	Antero-posteriority	Medio-laterality	Dorso-ventrality
Padilla-Coreano et al. 2016 (PFC)	From +1.54 to +2.1	±0.4	-1.25
Lowery-Gionta et al., 2018 (PFC)	+1.9 to +2.1	±1	-2
Felix-Ortiz et al., 2016 (PFC)	+1.7	±0.3	-1.9
Marcus et al., 2020 (PFC)	+2.42	±0.35	-2.09
Becker et al. (ACC)	+0.7	±0.2	-1.0

Also, since we had previously fully characterized anxiodepressive-like behaviors induced in our neuropathic model using a battery of tests (including the elevated plus-maze, light-dark, marble burying, NSF, splash, tail suspension and forced swimming tests; see Yalcin et al., 2011; Sellmeijer et al., 2018), in the present work we only used one test to confirm the expected phenotype for depressive- (Splash test) and anxiety-like (NSF) behaviors. When our goal was to characterize the BLA-ACC optogenetic model for the first time, we again used a battery of 5 different tests. Finally, we agree that including other tests assessing anxiety-like behavior in future work (using tests such as the EPM or open field) may deepen our understanding of this specific behavioral domain.

3. The authors use 3 days of stimulation every week for three weeks to induce the expression of depressive-like behaviors in mice. I would love the authors to explain what this is modelling in human conditions. For instance, is it a gradual but constant increase in the activity of the BLA-ACC pathway? How does this fit in the establishment of the phenotype of pain and depressive-like behaviors in human conditions?

Answer: The steps that led to our paradigm using repeated stimulations were empirical. Since initial experiments had shown that an acute, single activation of the BLA-ACC pathway had no detectable impact on emotional responses (Extended Data Fig.3), we then switched to 3 daily sessions. This was driven by: i) previous results from our group, which had shown that subchronic stimulation (over 4 days) of excitatory neurons of the ACC (regardless of their afferents) was sufficient to induce anxiodepressive-behaviours (see Barthas et al, Biol Psych), and ii) was compatible, practically, with the realization of behavioral tests each week. Results showed that 9 sessions of stimulation of the pathway over 3 weeks were necessary for the progressive emergence of deficits, which were absent at 1 week, selectively detectable at 2 weeks in the FST, and spread at 3 weeks across tests and global emotionality scores. We believe this kinetic bears an analogy with our results in the chronic

pain model, in which emotional deficits also progressively emerged, and required tonic hyperactivity of the BLA-ACC pathway. Compared to the human condition, we speculate that this transitory and short-lasting phenotype models the proximal causes of depression (such as environmental stressors, or recent life events), rather than the endophenotypes associated with the disorder. In other words, our paradigm models clinical variations in thymic states, rather than the biological traits that act as distal risk factor, and are likely not reversible.

To better understand whether this phenotype relies on a gradual increase or more abrupt changes in neuronal activity of the pathway, additional studies will be necessary. These may notably include electrophysiological analyses at the end of each round of 3 stimulations, transcriptional analysis after the first and second rounds (when behavioral deficits have not yet emerged), as well as optogenetic manipulations using different protocols.

4. The idea of combining mouse and human data in this study is very interesting and adds a very nice translational value to it. Still with the idea of testing pain-induced depressive-like behaviors, I was wondering if any of the samples included in the transcriptional analysis did suffer from chronic pain? If yes, it would be great to have this information in the text and provided in supplemental Table 2. Additionally, it would be great to see the incidence in the human cohort of other important confounding variables such as age, PH and postmortem interval, other environmental and life event variables such as early-life events, and comorbid conditions and medication status and know which one of them were controlled for in the analysis.

Answer: This is a great suggestion. Clinical, socio-demographic and life events data regarding our cohort of depressed individuals were collected through validated psychological autopsy procedures (Dumais et al. 2005; McGirr et al., 2006, 2009), which consist of structured proxy-based interviews with a best informant, typically next-of-kin. Unfortunately, these interviews focus on psychiatric disorders and did not include structured evaluation of non-psychosomatic pain (although sparse data might be available from medical records). From our discussions with scientific directors from the Douglas Brain Canada Brain Bank, it is expected that such information may be more frequently available for more recently recruited subjects.

Regarding confounding variables, as mentioned in the Methods section (and similar to our original study, Lutz PE et al., 2017), RNA integrity number (RIN) and age were included as covariates in the differential expression analysis model, when human data were reprocessed to conduct a new analysis in males only. PMI and pH were not included since they are largely collinear with RIN, and we (Nagy et al., 2015) and others (Gallego Romero et al., 2014) have shown that the latter is a more reliable estimate of tissue quality.

Regarding comorbidities, all control subjects corresponded to healthy individuals with no psychiatric history, while depressed individuals had a history of early-life adversity (ELA), but did not suffer from psychosis or bipolar disorder (exclusion criteria). This information is now included in the material and methods section (see page 28). Our original study had reported that impaired myelination in the group of depressed individuals with a history of ELA was absent in another group of depressed individuals with no such history. Combined with our newer results in the mouse, this may support the speculative interpretation that optogenetic activation of the BLA-ACC pathway models the molecular mechanisms by which ELA acts in humans as a susceptibility factor for depression.

5. Figure 5h shows a nice level of correlation between expression values in the RNAseq and qPCR dataset. I assume these genes were selected because they were identified as differentially expressed in the RNAseq data. Did qPCR confirm this

differential expression? Or did only the directionality of the effect was similar and highly correlated between techniques?

Answer: These genes were indeed selected because they were differentially expressed in RNAseq data. While qPCR results showed significant consistency with RNAseq data in terms of directionality, providing technical validation, some of those genes were individually not significantly differentially expressed. To clarify this point, p-values from qPCR experiments for individual genes are now depicted as a color gradient in revised Fig.6h.

6. The last series of experiments aims at showing that the optogenetic stimulation paradigm inducing the expression of depressive-like behaviors does so by impairing myelination. This is nicely supported by the MRI fractional anisotropy analysis performed. Then, they show that the KD of Sema4a in the ACC is sufficient to rescue the behavioral effects of this stimulation paradigm. However, this KD affects every pyramidal (CaMKIIa) expressing neurons and not only the ones receiving direct inputs from the BLA. How do the authors interpret this finding and how can these effects be associated mainly to the cells receiving inputs from the BLA?

Answer: Available data in the literature indicate that Sema4a is expressed by neurons as well as oligodendrocytes and microglia in the mouse brain (McDermott et al., 2018) while, to our knowledge, how this expression pattern may correlate with cellular or synaptic interactions with afferents from structures such as the BLA is unknown. As a first step, we therefore opted for a strategy that non-specifically counteracted optogenetically-induced Sema4a over-expression. We agree with the reviewer that we cannot conclude on the cell-type in which this over-expression occurs and is responsible for emotional deficits, nor can we infer whether these cells are directly innervated by incoming BLA fibers. We now acknowledge these perspectives at the end of the discussion:

“Among others, avenues for future work will include characterizing the mechanisms by which Sema4a modulates oligodendrocytes, identifying in which ACC cell-types such processes are recruited by BLA afferences, and describing putative sex differences in these aspects of pain and emotional processing” (see page 14).

These open up 2 avenues for next studies, in order to either knockdown Sema4a in specific ACC cell-types, or in neurons specifically targeted by BLA afferents. Both raise technical questions, corresponding respectively to the identification of strategies or promoters able to drive the cell-type specific, or anterograde pathway-specific, expression of interfering RNAs. Strategies have been proposed to overcome these questions, which we plan to test in future experiments.

7. I noticed that all over the different experiments, distinct behavioral tests are used to validate stress responses. For instance, ST and NSF in Figure 1, LD, ST and FST in Figure 2, LD, NSF, ST, Nesting and FST in Figure 3, ST in Figure 4, ST in Figure 5, Nesting in Figure 6 and ST and FST in Figure 7. For a matter of concordance and consistency, why not always using the same test or battery of test? Also, extended Figure 1H-I in the text refers to functional connectivity result but in fact refer to behavioral data. This should be corrected accordingly.

Answer: The tests assessing anxiety-like behaviors were selected depending on the purpose of the experiment and according to technical limitations. In Fig.1, we only used one test to assess depressive-like (Splash test) and another one to assess anxiety-like behaviors (NSF), since we had previously fully characterized anxiodepressive-like behaviors induced in our cuff model (Yalcin et al., 2011; Sellmeijer et al., 2018). In Fig.2, we were only able to demonstrate the results of the LD test since we had technical issues with the NSF test. Indeed, mice either lost their cannula during the testing or broke the optic fiber. We also

couldn't include the Nest test in this experimental paradigm as the duration of the test (5h) was not compatible with our stimulation protocol (5min). However, in Fig.3 (Fig.4 in revised manuscript), as we aimed at characterizing our optogenetic model for the first time, we used a battery of 5 behavioral tests including LD, NSF, ST, Nest and FST. For molecular experiments (RNA-Sequencing, qPCR) only one behavioral test was conducted for validation, before tissue dissection 48h after the 9th stimulation (the Splash test was systematically used for those transcriptional experiments). To reduce the number of animals used in the study, the first quantification of Olig2+ cells in the ACC after repeated activation of the BLA-ACC pathway was performed on slices obtained from a cohort used for the behavioral characterization (Fig.4). The selected cohort only did the Nest test after 9 stimulations. Finally in Fig.8 (revised manuscript) we aimed at testing if Sema4a could prevent the development of depressive-like behaviors. We therefore only used tests assessing these types of behaviors (ST and FST). Extended Fig1H-I (Extended Fig 1c-d, in revised version) corresponds to the behavioral validation of the rs-fMRI cohort. Indeed, we clearly showed the development of mechanical hypersensitivity and depressive-like consequences at 8 weeks after the surgery, as expected, before animals were submitted to rs-fMRI.

8. This work was performed in males only. Without asking to do all of this in females, I would like to see a discussion on how this pathway may contribute to driving similar effects in females. I would also be curious to see the authors' interpretation regarding the transcriptional signatures associated with these effects in females.

Answer: We agree with the reviewer. Unfortunately, one of the limitations of the post mortem brain study that we leveraged in the present work is a significant sex bias, with around 24 and 76% of females and males, respectively. While depression is more frequent in females, this ratio reflects the fact that subjects recruited at the Douglas Bell Canada Brain Bank primarily died by suicide, which affects men approximately 3 times more frequently than women. While we therefore have no direct evidence available at the moment, we did recently initiate large-scale mouse projects that will be dedicated to the investigation of sex as a biological variable. In particular, on-going experiments will provide an extensive characterization, in female mice, of the kinetic and behavioral profiles of chronic neuropathic pain-induced consequences (over a protracted period of up to 6 months, similar to our original work in males; Sellmeijer et al., 2018). We also plan to characterize the transcriptomic adaptations that occur in the female ACC as a function of chronic pain, or of BLA-ACC activation, with an emphasis on underlying epigenetic sex-specific mechanisms (in particular DNA methylation, building on our recent studies; Lutz et al., 2021). This perspective is now mentioned at the end of the discussion (see page 14):

"Among others, avenues for future work will include the characterization of the mechanisms by which Sema4a modulates oligodendrocytes, identifying in which ACC cell-types such processes are recruited by BLA afferences, and describing putative sex differences in these aspects of pain and emotional processing."

Bibliography

Dai., W. et al. Sex-Specific Transcriptomic Signatures in Brain Regions Critical for Neuropathic Pain-Induced Depression. *Front Mol Neurosci* **15**, 886916 (2022).

Dumais, A. et al. Risk factors for suicide completion in major depression: a case-control study of impulsive and aggressive behaviors in men. *Am J Psychiatry* **162**, 2116–2124 (2005).

- Fillinger, C., Yalcin, I., Barrot, M., Veinante, P. Efferents of anterior cingulate areas 24a and 24b and midcingulate areas 24a' and 24b' in the mouse. *Brain Struct Funct* **223**, 1747-1778. (2018)
- Fillinger, C., Yalcin, I., Barrot, M., Veinante, P. Afferents to anterior cingulate areas 24a and 24b and midcingulate areas 24a` and 24b` in the mouse. *Brain Struct Funct* **222**,1509-1532. (2017)
- Gallego Romero, I., Pai, A. A., Tung, J. & Gilad, Y. RNA-seq: impact of RNA degradation on transcript quantification. *BMC Biol* **12**, 42 (2014).
- Gonçalves, L. and Dickenson, AH. Asymmetric time-dependent activation of right central amygdala neurons in rats with peripheral neuropathy and pregabalin modulation. *Eur J Neurosci* **36**, 3204-13 (2012).
- Guilloux, JP., Seney, M., Edgar, N., Sibille, E. Integrated behavioral z-scoring increases the sensitivity and reliability of behavioral phenotyping in mice: relevance to emotionality and sex. *J Neurosci Methods* **97**, 21-31 (2011).
- Lutz, P.-E. *et al.* Association of a History of Child Abuse With Impaired Myelination in the Anterior Cingulate Cortex: Convergent Epigenetic, Transcriptional, and Morphological Evidence. *Am J Psychiatry* **174**, 1185–1194 (2017).
- Lutz, P.-E. *et al.* Non-CG methylation and multiple histone profiles associate child abuse with immune and small GTPase dysregulation. *Nat Commun* **12**, 1132 (2021).
- McDermott, J.E., Goldblatt, D., Paradis, S. Class 4 Semaphorins and Plexin-B receptors regulate GABAergic and glutamatergic synapse development in the mammalian hippocampus. *Mol Cell Neurosci* **92**, 50-66 (2018).
- McGirr, A. *et al.* Risk factors for completed suicide in schizophrenia and other chronic psychotic disorders: a case-control study. *Schizophr Res* **84**, 132–143 (2006).
- McGirr, A. *et al.* Familial aggregation of suicide explained by cluster B traits: a three-group family study of suicide controlling for major depressive disorder. *Am J Psychiatry* **166**, 1124–1134 (2009).
- McKenzie, I. A. *et al.* Motor skill learning requires active central myelination. *Science* **346**, 318–322 (2014).
- Nagy, C. *et al.* Effects of postmortem interval on biomolecule integrity in the brain. *J Neuropathol Exp Neurol* **74**, 459–469 (2015).
- Sellmeijer, J. *et al.* Hyperactivity of Anterior Cingulate Cortex Areas 24a/24b Drives Chronic Pain-Induced Anxiodepressive-like Consequences. *J Neurosci* **38**, 3102–3115 (2018).
- Shukla, R. *et al.* Molecular characterization of depression trait and state. *Mol Psychiatry* **27**, 1083–1094 (2022).
- Vogt, B.A., Paxinos, G. Cytoarchitecture of mouse and rat cingulate cortex with human homologies. *Brain Struct Funct* **219**,185–192 (2014).
- Yalcin, I. *et al.* A time-dependent history of mood disorders in a murine model of neuropathic pain. *Biol Psychiatry* **70**, 946–953 (2011).

REVIEWERS' COMMENTS

Reviewer #1 (Remarks to the Author):

The authors have addressed this reviewers concerns. This paper remains a nice example of using data across species to investigate an important clinical problem. Looking forward to seeing it in Print.

Reviewer #2 (Remarks to the Author):

The authors have adequately addressed all of my concerns. The revised paper by Becker and colleagues has been improved since its original submission. This well-executed and thorough study expands our understanding of the neuronal circuitries and the molecular/cellular substrates underlying the emergence of depression as a consequence of chronic pain.

Reviewer #3 (Remarks to the Author):

I would like to thank the authors for their thoughtful reply. In my case, most of my initial comments have been properly addressed. I would still request two things.

1. The answers provided for my comments on points 2 (anxiety vs pain pathways) and 3 (what is the optogenetic paradigm modeling in human) are interesting and would deserve to be added to the discussion.

2. The answer provided for my comment on point 8 (how this pathway may contribute to the observed effects in females) is in my mind insufficient. By our days, it is now almost necessary to study both sexes and when studying only one sex, strong justifications are required. I am not asking to justify anything or redo the experiments. I am just asking for a thorough and thoughtful discussion on what could be happening in females.

Besides, all is good for me. Congrats on a great study.

We thank all reviewers for their positive evaluations. Here are some points that raised by the 3rd reviewer.

Reponses to Reviewer 3:

1. The answers provided for my comments on points 2 (anxiety vs pain pathways) and 3 (what is the optogenetic paradigm modeling in human) are interesting and would deserve to be added to the discussion.

We have now added the following sections to the revised manuscript:

Point 2: « The lack of detectable impact of this optogenetic manipulation on anxiety-like responses also strengthens the aforementioned notion that ACC inputs coming from the BLA selectively modulate mood states. Interestingly, activation of BLA terminals in other subparts of the PFC (i.e. areas 25/32, located more rostrally), different from those recruited in the present work (ACC areas 24a/24b), induced anxiety-like behavior⁶²⁻⁶⁵. These results therefore suggest differential control of emotional responses by multiple pathways that originate in the BLA but target separate cortical regions. » (Page 14)

Point 3: « Overall, it is possible to speculate that this kinetic bears an analogy with our results in the NPID model, in which emotional deficits also progressively emerged, and required tonic hyperactivity of the BLA-ACC pathway. Compared to the human disorder, this short-lasting phenotype might be considered to model proximal causes of depression (such as environmental stressors or recent life events), which trigger acute variations in thymic states, rather than underlying endophenotypes (constitutive biological traits thought to act as distal risk factor). » (Page 8)

2. The answer provided for my comment on point 8 (how this pathway may contribute to the observed effects in females) is in my mind insufficient. By our days, it is now almost necessary to study both sexes and when studying only one sex, strong justifications are required. I am not asking to justify anything or redo the experiments. I am just asking for a thorough and thoughtful discussion on what could be happening in females.

We now further discussed this point in the discussion (see Page 16):

« Accordingly, a limitation of the present study is that we focused on males only. Since sexual differences have been described in relation to chronic pain and associated emotional dysregulation⁷⁹⁻⁸¹, future studies will be necessary to investigate sex as a biological variable in behavioural and transcriptomic consequences of various models (including NPID), as well as in interactions among structures such as the BLA and the ACC (with no concrete evidence available to date). »